# Transfer Learning in Infinite Width Feature Learning Networks

**Clarissa Lauditi**[1]**, Blake Bordelon**[2]**, Cengiz Pehlevan**[1,3]

John A. Paulson School of Engineering and Applied Sciences[1]
Center for Mathematical Sciences and Applications[2]
Kempner Institute for the Study of Natural and Artificial Intelligence[3]
Harvard University
Cambridge, MA 02138, USA
{clauditi,cpehlevan}@seas.harvard.edu, blake@cmsa.fas.harvard.edu

## Abstract

We develop a theory of transfer learning in infinitely wide neural networks under gradient flow that quantifies when pretraining on a source task improves generalization on a target task. We analyze both (i) fine-tuning, when the downstream predictor is trained on top of source-induced features and (ii) a jointly rich setting, where both pretraining and downstream tasks can operate in a feature learning regime, but the downstream model is initialized with the features obtained after pre-training. In this setup, the summary statistics of randomly initialized networks after a rich pre-training are adaptive kernels which depend on both source data and labels. For (i), we analyze the performance of a readout for different pretraining data regimes. For (ii), the summary statistics after learning the target task are still adaptive kernels with features from both source and target tasks. We test our theory on linear and polynomial regression tasks as well as real datasets. Our theory allows interpretable conclusions on performance, which depend on the amount of data on both tasks, the alignment between tasks, and the feature learning strength.

## 1 Introduction

Modern deep-learning models achieve remarkable accuracy by scaling parameters, computation, and data (Hestness et al., 2017; Kaplan et al., 2020; Hoffmann et al., 2022). Yet collecting such large volumes of data is prohibitively expensive or outright impossible in many settings. Transfer learning offers a principled escape from this data bottleneck: by repurposing representations learned on data-rich source tasks, it reduces sample complexity while improving generalization (Tan et al., 2018; Brown et al., 2020; Li et al., 2020; Isik et al., 2025). Therefore, understanding which properties of the pretraining and downstream data distributions enable effective transfer is critical for modern deep learning. Despite its empirical success, transfer learning still lacks a principled theory that predicts when it will succeed. In this paper, we present a novel theory of transfer learning in multi-layer neural networks that elucidate the rich phenomenology of transfer learning.

Mathematically analyzing transfer learning is challenging, in part because representation learning in generic neural networks remains poorly understood. To overcome this difficulty, we focus on transfer after **representation learning in infinite-width neural networks** in the $\mu$P/mean-field parameterization (Song et al., 2018a; Chizat & Bach, 2018; Yang & Hu, 2021; Bordelon & Pehlevan, 2023). In this parameterization, feature learning is preserved even as the width of the network goes to infinity. We focus on supervised learning for both source and target tasks and derive results for the network performance after each phase of transfer learning. In particular, we analyze (1) linear toy models of fine-tuning with adaptive kernels after feature learning on source task and (2) non-linear models of transfer learning when both source and target tasks can operate in a feature learning regime. Our theory enables accurate predictions of the resulting network models for wide but finite neural networks.

Concretely, the contributions of this work are the following:

- We develop a theory of transfer learning for randomly initialized infinite width multilayer perceptrons (MLPs). This theory, in its most general form, allows for arbitrary laziness on task-1 (pre) or task-2 (post) training. In general (for models with more than one hidden layer), this theory is quite complex and involves non-markovian history dependence during both phases of optimization.

- To gain more analytical tractability we specialize our theory to two layer neural networks and investigate transfer learning in this setting. We analyze both fine tuning, where training on the second task is lazy, and rich learning where training on the second task can cause large changes in the hidden features. In the regime of finetuning, we can utilize results for the final feature kernels to characterize the predictors on the second task.

- We develop linear toy models of finetuning where we can explicitly compute typical test losses on the second task when sampling random pre and post training sets. These linear toy models reveal many aspects of the phase diagram of (un)successful transfer learning. If the pretraining (source) task is data rich, fine-tuning strictly improves over a two-layer linear model trained from random initialization. With limited data during pretraining, noise due to finite sample-size effects can cause negative transfer. For *very* rich pre-training, fine-tuning is sample efficient if and only if the target has significant projection on the pre-training source feature.

- We extend this investigation beyond linear tasks to polynomial source/target tasks and on real computer vision datasets. Consistently with our theoretical predictions, when the pre-training task is data-rich, fine-tuning on the second task after rich pretraining improves performance and sample-efficiency. With limited source data, rich pretraining can induce representation overfitting by causing negative transfer. In this setting, rich learning on the second task is often favorable.

## 1.1 RELATED WORKS

**Theory of Transfer Learning in Linear Models.** Several works have studied how properties of a representation support generalization from few examples on a downstream task (Bordelon et al., 2020; Canatar et al., 2021a; Sorscher et al., 2022; Dhifallah & Lu, 2021; Gerace et al., 2022). A general result is that the geometry of the neural representation (kernel-task alignment) controls the ability to learn a new supervised task from limited data (Canatar et al., 2021b). However, these theories at infinite width would predict a fixed representation at initialization, not allowing for features to adapt during learning, for either the source or the downstream tasks.

**Training Dynamics in Wide Networks.** Recent years have seen significant research on the learning dynamics of wide, randomly initialized neural networks. In standard / neural tangent parameterization, wide neural networks are described by kernel methods (Jacot et al., 2020; Arora et al., 2019; Lee et al., 2020). In this same parameterization, corrections to this limit at large but finite width reveal weak (perturbative) feature learning corrections to this limit, linearizing the dynamics of hidden representations around their static infinite width value (Roberts et al., 2022; Zavatone-Veth et al., 2021). Alternatively, other works have explored parameterizations that allow infinite width networks to learn features, known as mean-field or $\mu$P scaling, resulting in fundamentally nonlinear predictor dynamics. These works developed tools to study the representation learning dynamics during gradient descent training in infinite width neural networks, which require adoption of the mean-field/$\mu$P scaling of network width (Song et al., 2018b; Chizat & Bach, 2018; Yang & Hu, 2021; Bordelon & Pehlevan, 2023; Bordelon et al., 2024c; Bordelon & Pehlevan, 2022). In this infinite limit, the dynamics for kernels cannot be linearized around the lazy learning solution.

**Learning in Wide Bayesian Networks.** In contrast to gradient descent training, some works have pursued theory of networks sampled from a Bayesian posterior (Welling & Teh, 2011). In the infinite width $N \to \infty$ limit with neural tangent kernel (NTK) parameterization and dataset size $P$ held constant, networks converge to neural network Gaussian process (NNGP) models, which lacks representation learning (Lee et al., 2018). Beyond this kernel limit, extensions of deep Bayesian MLPs in NTK parameterization under the proportional limit $P, N \to \infty$ with $P/N = \alpha$ reveal scale–renormalized kernels after training (Li & Sompolinsky, 2021; Pacelli et al., 2023; Baglioni et al., 2024), with extensions to convolutional architectures (Aiudi et al., 2023; Bassetti et al., 2024). Large-deviation analyses in NTK parameterization further show kernel adaptation in finite-

width/proportional limits (Fischer et al., 2024; Rubin et al., 2024b; Seroussi et al., 2023; Andreis et al., 2025). An alternative strategy is to *adopt a mean-field/μP-like* parameterization where even the $N \to \infty$ limit at fixed $P$ give rise to significant changes in the kernels and predictor statistics compared to NNGP regression (Aitchison, 2020; Lauditi et al., 2025). Proportional limits in deep Bayesian networks have also been analyzed under the mean field scaling (Rubin et al., 2024a; van Meegen & Sompolinsky, 2024).

**Transfer Learning in Wide Networks.** Bayesian networks have been studied in a general multi-task framework in NTK parameterization in both lazy and proportional ($P/N = \alpha$) limits (Ingrosso et al., 2025; Shan et al., 2025). The works (Ingrosso et al., 2025; Shan et al., 2025) first introduced a Bayesian transfer-learning framework in which the target model is regularized to remain in the vicinity of the pre-trained source weights (which are treated as fixed realizations of the source posterior). In (Tahir et al., 2024) the authors analyze deep linear models of fine-tuning on synthetic data, in the special case when the source task has infinite data and the kernel is low rank, by showing that positive transfer learning depends on feature similarity between source and target tasks. A recent work analyzes fine-tuning for two-layer mean-field models under KL-regularized empirical risk minimization (Aminian et al., 2024). Here, we develop a theory for fine-tuning using adaptive kernels from source task, and in a finite-data regime where sample fluctuations can hurt generalization. Plus, we extend the theory for non-linear networks and in the jointly rich setting where feature learning can also happen on target task.

**Continual Learning Dynamics.** Gradient descent training under continual learning in large-width networks under mean-field scaling has been studied in Graldi et al. (2024). This analysis revealed that richer training dynamics could lead to more catastrophic forgetting in a sequential multi-task learning, where the task distribution shifts over training time. Average accuracy across tasks was often maximized at an intermediate feature learning strength. However, these results have not yet been studied within a theoretical framework.

## 2 MODEL AND TRANSFER LEARNING DEFINITIONS

Before specializing to specific transfer learning settings (such as fine tuning or linear networks), we first provide a general framework where we subsumes all of our analysis. Our width $N$ and depth $L$ MLP architecture has the form

$$f(\boldsymbol{x}) = \frac{1}{N}\boldsymbol{w}^L \cdot \boldsymbol{\phi}(\boldsymbol{h}^L(\boldsymbol{x})) \,, \; \boldsymbol{h}^{\ell+1} = \frac{1}{\sqrt{N}}\boldsymbol{W}^\ell \boldsymbol{\phi}(\boldsymbol{h}^\ell(\boldsymbol{x})) \,, \; \boldsymbol{h}^1 = \frac{1}{\sqrt{D}}\boldsymbol{W}^0\boldsymbol{x} \qquad (1)$$

where $\boldsymbol{x} \in \mathbb{R}^D$ is an input to the model and the variables $\boldsymbol{h}^\ell \in \mathbb{R}^N$ represent the hidden preactivation features in the forward pass. During pretraining, the model parameters $\{\boldsymbol{W}^\ell\}$ are optimized with (S)GD on the *source* or task-1 dataset $\mathcal{T}_1 = \{(\boldsymbol{x}_\mu^{(1)}, y_\mu^{(1)})\}_{\mu=1}^{P_1}$ where the loss function on the $P_1$ training points in $\mathcal{T}_1$ takes the form

$$\mathcal{L}_{\mathcal{T}_1}(\boldsymbol{\theta}) = \mathbb{E}_{\boldsymbol{x},y \in \mathcal{T}_1} \, \ell\left(\gamma_1^{-1} f(\boldsymbol{x}, \boldsymbol{\theta}), y\right), \qquad (2)$$

where $\ell$ is the per-data-point loss function (e.g. MSE or cross-entropy). The parameter $\gamma_1$ represents the *richness/non-linearity of optimization* for task-1 pretraining with $\gamma_1 \to 0$ corresponding to lazy / kernel learning (Chizat et al., 2020; Geiger et al., 2020; Bordelon & Pehlevan, 2022). This generates a final set of parameters $\boldsymbol{\theta}_1$. Using the final parameters from pretraining $\boldsymbol{\theta}_1$ as a starting point for transfer, we then run (S)GD on a second task $\mathcal{T}_2 = \{(\boldsymbol{x}_\mu^{(2)}, y_\mu^{(2)})\}_{\mu=1}^{P_2}$ on a loss function using a second richness parameter $\gamma_2$.

$$\mathcal{L}_{\mathcal{T}_2}(\boldsymbol{\theta}) = \mathbb{E}_{\boldsymbol{x},y \in \mathcal{T}_2} \, \ell(\gamma_2^{-1} f(\boldsymbol{x}, \boldsymbol{\theta}), y) \qquad (3)$$

We are ultimately interested in the solutions (and generalization performance) of the model that was post-trained on task-2. We will refer to the case where lazy learning on task-2 is performed $\gamma_2 \to 0$ as **fine-tuning** [1].

This general setting can be extended for Bayesian networks (see Appendix F), by considering the source task weights as quenched disordered variables for the target task $\mathcal{T}_2$. Here, an elastic weight coupling controls the reuse of features during transfer learning.

---

[1] Technically, to control initialization variance, we take $N \to \infty$ first before taking $\gamma_2 \to 0$.

## 2.1 Utilizing Infinite Width Feature Learning Limits

To make analytical progress on this problem, we focus our attention on *infinite width neural networks* $N \to \infty$ trained with gradient flow. Because the networks are in the mean-field/$\mu$P parameterization, this infinite limit preserves feature learning for $\gamma_1, \gamma_2 > 0$ Bordelon & Pehlevan (2022). If the weights are initialized i.i.d. with unit variance, and the model is trained with SGD with learning rate $\eta_i = \eta_0 N \gamma_i^2$ for $i \in \{1, 2\}$, then the final predictor $f(\boldsymbol{x})$ after post-training on task 2 can be expressed in terms of a collection of kernels that include

$$\Phi^\ell(\boldsymbol{x}, \boldsymbol{x}', t, t') = \frac{1}{N} \phi(\boldsymbol{h}^\ell(\boldsymbol{x}, t)) \cdot \phi(\boldsymbol{h}^\ell(\boldsymbol{x}', t')) \tag{4}$$

where $(t, t')$ are distinct time values for training across both gradient flow time in task-1 and task-2 (Yang & Hu, 2022; Bordelon & Pehlevan, 2022; Lauditi et al., 2025; Graldi et al., 2024). In the infinite width $N \to \infty$ limit, these functions become deterministic in their evolution and the neurons become statistically independent over the random initialization of weights. While this (in principle) provides a closed set of equations for the evolution of the network predictions $f(\boldsymbol{x})$, the resulting dynamics are quite complex (see Appendix B). To gain more insight into the mechanisms of transfer learning we will next specialize to simpler settings.

## 2.2 Two Stage Gradient Flow Dynamics for Two Layer Networks

First, we will examine the training dynamics for two layer networks where the dynamics in feature space are Markovian.

**Result 1 (In data-poor downstream regimes, feature learning on target task helps)** *Consider a two-layer ($L = 1$) MLP trained with gradient flow on $\mathcal{T}_1$ for times $t \in (0, t_1)$ with $\gamma_1$ and then subsequently trained on task $\mathcal{T}_2$ for times $t \in (t_1, t_2)$ with richness parameter $\gamma_2$. The infinite width $N \to \infty$ dynamics of the second model under gradient flow and with weight decay converges after a training time $t > t_1$ to a predictor $f(\boldsymbol{x}, t)$ on a test point $\boldsymbol{x}$*

$$f_1(\boldsymbol{x}, t) = \gamma_1^{-1} \langle z(t) \phi(h(\boldsymbol{x}, t)) \rangle \tag{5}$$

*where the average $\langle \cdot \rangle$ represents an average over the measure of hidden neuron activations. The preactivations $\boldsymbol{h}_\mu(t) = \frac{1}{\sqrt{D}} \boldsymbol{W}^0(t) \boldsymbol{x}_\mu$ and the readout variables $\boldsymbol{z}(t) = \boldsymbol{w}^1(t)$ evolve as single-site stochastic processes (neuron - decoupled) under the dynamical mean field theory (DMFT) equations*

$$h(\boldsymbol{x}, t) = \chi(\boldsymbol{x}) + \gamma_1 \int_0^{t_1} ds \sum_{\mu \in \mathcal{T}_1} \Delta_\mu(s) g_\mu(s) K_x(\boldsymbol{x}, \boldsymbol{x}_\mu) + \gamma_2 \int_{t_1}^t ds \sum_{\nu \in \mathcal{T}_2} \Delta_\nu(s) g_\nu(s) K_x(\boldsymbol{x}, \boldsymbol{x}_\nu)$$

$$z(t) = \psi + \gamma_1 \int_0^{t_1} ds \sum_{\mu \in \mathcal{T}_1} \Delta_\mu(s) \phi(h_\mu(s)) + \gamma_2 \int_{t_1}^t ds \sum_{\mu \in \mathcal{T}_2} \Delta_\mu(s) \phi(h_\mu(s))$$

$$g_\mu(t) = \dot{\phi}(h_\mu(t)) z(t). \tag{6}$$

*and the average $\langle \cdot \rangle$ is over both $\psi \sim \mathcal{N}(0, 1)$ and $\chi(\boldsymbol{x}) \sim \mathcal{GP}(0, \boldsymbol{K}_x)$ where $K_x(\boldsymbol{x}, \boldsymbol{x}') = \frac{1}{D} \boldsymbol{x} \cdot \boldsymbol{x}'$, while $\Delta_\mu(t) = -\partial_{f_\mu} \ell(f_\mu, y_\mu)$ represents error signals for the training points in $\mathcal{T}_1$ and $\mathcal{T}_2$. The predictor on the second task can be computed as $f_2(\boldsymbol{x}, t) = \gamma_2^{-1} \langle z(t) \phi(\boldsymbol{h}(\boldsymbol{x}, t)) \rangle$ for any $t > t_1$.*

At a high level, this mean-field theory states that interactions between neurons asymptotically decouple and all macroscopic properties of the network, including the predictors $f_1, f_2$ obey deterministic equations (Sompolinsky & Zippelius, 1981). Intuitively, averages over the neural population $\frac{1}{N} \sum_{i=1}^N g(h_i)$ should converge to population averages $\langle g(h) \rangle$ over a limiting density by a law of large numbers. This two-stage learning result indicates that there is a history dependence of the dynamics on the downstream task $\mathcal{T}_2$ that is inherited from the dynamics of pretraining on task $\mathcal{T}_1$, consistent with prior works on mean field continual/transfer learning (Graldi et al., 2024; Aminian et al., 2024). In this two layer setting, this dependence only enters through the random variables $\{h(t_1), z(t_1)\}$ which set the initial condition for the downstream task $\mathcal{T}_2$ due to the above Markov structure. This property does not hold in deeper models (see Appendix B). To validate the DMFT predictions, we numerically simulate the single-site stochastic processes using a Monte-Carlo approximation of the population averages (see Appendix D). At each time step we evolve the fields via

Euler discretization and compute the induced feature kernel and losses on $\mathcal{T}_1$ and $\mathcal{T}_2$. We provide simulations of transfer learning using the above stochastic processes in Figures 3, 4 revealing that $(\gamma_1, \gamma_2)$ can both impact the impact of pretraining on transfer learning.

One finding that we consistently see is that if the amount of data $P_2$ on $\mathcal{T}_2$ is small, that transfer learning confers greater benefits. Since the above model implicitly depends on the dataset size, but does not explicitly quantify how transfer learning depends on $P_1, P_2, \gamma_1, \gamma_2$, we next investigate the even simpler setting of linear networks.

## 2.3 TOY MODELS OF FINE-TUNING IN TWO-LAYER LINEAR NETWORKS

While the previous section described nonlinear two layer networks on arbitrary data, they did not admit an average-case analysis of generalization through averaging over the random datasets $\mathcal{T}_1, \mathcal{T}_2$. In this section, we analyze a tractable model which enables average case analysis of generalization for transfer, specifically deep linear models where $\phi(h) = h$. The source task $\mathcal{T}_1$ is generated by a linear target function $y_{s,\mu} = \frac{1}{\sqrt{D}} \boldsymbol{\beta}_s \cdot \boldsymbol{x}_{s,\mu}$ on random isotropic data $\boldsymbol{x}_{s,\mu} \sim \mathcal{N}(0, \boldsymbol{I}_D)$. The same is valid for the target task $\mathcal{T}_2$ with $y_{t,\mu} = \frac{1}{\sqrt{D}} \boldsymbol{\beta}_t \cdot \boldsymbol{x}_{t,\mu}$ and $\boldsymbol{x}_{t,\mu} \sim \mathcal{N}(0, \boldsymbol{I}_D)$. We pre-train on $\mathcal{T}_1$ with gradient flow on a squared loss. This induces an adaptive feature kernel $\boldsymbol{K}(t) = \langle \boldsymbol{h}(t)\boldsymbol{h}(t)^\top \rangle$ where the average is over the distribution of hidden neurons under Eq. 5. At the end of pre-training $(t_1 \to \infty)$, we freeze this kernel and treat it as a fixed feature map for the downstream task $\mathcal{T}_2$. Fine-tuning then reduces to kernel regression via gradient flow. The predictor on the downstream task follows the dynamics $\frac{d}{dt} f_2(\boldsymbol{x}) = \boldsymbol{k}(\boldsymbol{x})^\top \boldsymbol{K}(\boldsymbol{y} - \boldsymbol{f}_2(t))$ where $[\boldsymbol{k}(\boldsymbol{x})]_\mu = K(\boldsymbol{x}, \boldsymbol{x}_\mu)$ for $\mu \in [P_2]$ and $\boldsymbol{K} \in \mathbb{R}^{P_2 \times P_2}$ is the kernel on the target task data and $\mathbf{f}_2(t) \in \mathbb{R}^{P_2}$ is the predictor on task 2.

*Our goal is to quantify how the amount of data and the strength of feature learning $\gamma_1$ during pre-training affect performance on a downstream task.*

To do so, we study three regimes.

1. First, we look at the population limit $(P_1 \to \infty)$ in Result 2. In this regime, feature learning during pre-training on $\mathcal{T}_1$ is always beneficial for transfer.

2. Then we consider how finite data on $\mathcal{T}_1$ influences fine-tuning generalization in Result 3.

3. Finally, we look at the ultra-rich regime $\gamma_1 \to \infty$ where the adaptive NTK kernel becomes low rank. In this regime, finite data effects severely limit performance.

In the following, we give a sketch of the results for each case, clarifying both $(i)$ the adaptive feature kernel after pre-training on $\mathcal{T}_1$ and $(ii)$ the final test loss performance on $\mathcal{T}_2$ at convergence. This has to be compared with the test loss of an uninformed linear predictor, trained directly on $\mathcal{T}_2$.

**Result 2 (Data-rich pre-training consistently improves transfer)** *Consider the deep linear MLP of Eq. 1 with $\phi(x) \equiv x$. Train by gradient flow on $\mathcal{T}_1$ and feature-learning strength $\gamma_1 > 0$. In the infinite-width limit $N \to \infty$, and then in the population limit $P_1 \to \infty$ at fixed $D$, the adaptive feature kernel after pre-training converges to (see for instance Bordelon & Pehlevan (2022))*

$$\boldsymbol{K}^\ell(\boldsymbol{X}, \boldsymbol{X}') = \boldsymbol{X} \left[ \boldsymbol{I} + \frac{\chi^\ell}{D} \boldsymbol{\beta}_s \boldsymbol{\beta}_s^\top \right] \boldsymbol{X}'^\top, \tag{7}$$

*i.e., a rank-one spike along $\boldsymbol{\beta}_s \boldsymbol{\beta}_s^\top$. Moreover, $\chi^\ell$ increases strictly with $\gamma_1$.*

*With this adaptive kernel from $\mathcal{T}_1$, freeze the features and fine-tune the readout on $\mathcal{T}_2$. In the proportional limit $P_2, D \to \infty$ with $P_2 = \nu_2 D$ and for a fixed source/target alignment $\alpha_s = \frac{1}{D} \boldsymbol{\beta}_s \cdot \boldsymbol{\beta}_t$, the downstream test loss at convergence is*

$$\mathcal{L}(\nu_2, \alpha_s, \chi^\ell) = (1 - \nu_2) \left[ 1 - \frac{2\chi^\ell \alpha_s^2 \nu_2}{1 + \chi^\ell \nu_2} + \frac{(\chi^\ell)^2 \alpha_s^2 \nu_2^2}{(1 + \chi^\ell \nu_2)^2} \right] \leq (1 - \nu_2). \tag{8}$$

*Thus fine-tuning with the adaptive kernel is always better than the baseline $\mathcal{L} = 1 - \nu_2$, which one would obtain from random initialization, whenever $\chi^\ell > 0$ and $\alpha \neq 0$.*

For gradient flow from random initialization in a $L = 1$ linear network (Bordelon & Pehlevan, 2022) showed that the kernel has this spiked form with $\chi = \sqrt{1 + \gamma_1^2} - 1$ (see Appendix C.1). Similarly,

for Bayesian networks in the feature learning regime at infinite width, the $\chi^\ell$ at the end of training can be solved for exactly in terms of $\gamma_1$ as the solution degree $L$ polynomial Lauditi et al. (2025).

Under gradient flow finetuning on $\mathcal{T}_2$, the error vector which defines the generalization error is $\boldsymbol{v}_0(t) = \boldsymbol{\beta}_t - (\boldsymbol{K}^\ell)^{1/2}\hat{\boldsymbol{\beta}}(t)$, while the instantaneous training errors are $\boldsymbol{\Delta}(t) = D^{-1/2}\boldsymbol{X}_t^\top \boldsymbol{v}_0(t)$. The key quantities which determine the generalization dynamics on $\mathcal{T}_2$ are the correlation functions

$$C_\Delta(t,t') = \frac{1}{P_2}\boldsymbol{\Delta}(t)\cdot\boldsymbol{\Delta}(t'), \quad C_{v_0}(t,t') = \frac{1}{D}\boldsymbol{v}_0(t)\cdot\boldsymbol{v}_0(t'), \quad C_{sv_1}(t) = \frac{1}{D}\boldsymbol{\beta}_s\cdot\boldsymbol{v}_1(t), \quad (9)$$

with $\boldsymbol{v}_1(t) = \frac{\sqrt{D}}{P_2}\boldsymbol{X}\boldsymbol{\Delta}(t)$. From these, train loss $\hat{\mathcal{L}}(t)$ and test losses $\mathcal{L}(t)$ correspond respectively to

$$\hat{\mathcal{L}}(t) = C_\Delta(t,t), \quad \mathcal{L}(t) = C_{v_0}(t,t). \quad (10)$$

In the join limit $P_2, D \to \infty$ with $\nu_2 = P_2/D$, each entry of the vectors $\{\boldsymbol{\Delta}(t), \boldsymbol{v}_0(t), \boldsymbol{v}_1(t)\}$ becomes i.i.d. and described by a stochastic process known as single-site process. This is summarized by the DMFT fixed point equations, from which one can derive correlation functions as in Eq. 10, which are deterministic in the limit.

In the results that follow, the derivation for $\mathcal{T}_2$ test loss is similar in spirit, with the addition of correlation and response functions that depend specifically on the adaptive kernel after $\mathcal{T}_1$. We restrict to two-layer setting, even though we believe that the adaptive kernels after feature learning on $\mathcal{T}_1$ in the deep case have the same functional form as the one we study here.

**Result 3 (Finite-sample size effects can harm fine-tuning gains)** *Consider the two-layer MLP of Eq. 1 with $L = 1$ and $\phi(x) \equiv x$ at infinite width. In the proportional limit where $P_1, D \to \infty$ with $P_1 = \nu_1 D$, rescale $\gamma_1 = \tilde{\gamma}_1/\sqrt{D}$ for feature learning to happen at infinite width. After pre-training on $\mathcal{T}_1$, the adaptive feature kernel at convergence is*

$$\boldsymbol{K}(\boldsymbol{X}, \boldsymbol{X}') = \boldsymbol{X}\left[\boldsymbol{I} + \frac{c_1}{D}\left(\boldsymbol{g}\boldsymbol{\beta}_s^\top + \boldsymbol{\beta}_s\boldsymbol{g}^\top\right) + \frac{c_2}{D}\boldsymbol{\beta}_s\boldsymbol{\beta}_s^\top + \frac{c_3}{D}\boldsymbol{g}\boldsymbol{g}^\top\right]\boldsymbol{X}'^\top, \quad (11)$$

*i.e., a low-rank deformation of the isotropic baseline: a signal spike $\boldsymbol{\beta}_s\boldsymbol{\beta}_s^\top$, a noise spike $\boldsymbol{g}\boldsymbol{g}^\top$, and a crosstalk term $\boldsymbol{g}\boldsymbol{\beta}_s^\top + \boldsymbol{\beta}_s\boldsymbol{g}^\top$. We establish this novel result for the trained kernel in Appendix C.2 which incorporates finite data effects. The Gaussian vector $\boldsymbol{g}$ captures these finite-sample fluctuations of the $\mathcal{T}_1$ dataset and it is uncorrelated with $\boldsymbol{\beta}_s$. Its covariance $\text{Cov}(\boldsymbol{g}) = \frac{1}{\nu_1}C_\Delta^\infty$ is set by the train loss at convergence on $\mathcal{T}_1$, given by*

$$C_\Delta^\infty = \lim_{t\to\infty}\frac{1}{P_1}\boldsymbol{\Delta}(t)\cdot\boldsymbol{\Delta}(t), \quad \boldsymbol{\Delta}(t) = \frac{1}{\sqrt{D}}\boldsymbol{X}(\boldsymbol{\beta}_s - \frac{\sqrt{D}}{\gamma_1 N}\boldsymbol{W}(t)^\top\boldsymbol{w}(t)). \quad (12)$$

*The coefficients $c_1, c_2, c_3$ are deterministic functions of $(\tilde{\gamma}_1, \nu_1)$ given by the DMFT saddle point equations.*

*With this adaptive feature kernel from $\mathcal{T}_1$, freeze the features and fine-tune the readout on $\mathcal{T}_2$. Call $\alpha_s = \frac{1}{D}\boldsymbol{\beta}_s\cdot\boldsymbol{\beta}_t, \alpha_g = \frac{1}{D}\boldsymbol{g}\cdot\boldsymbol{\beta}_t$ the alignments of the target direction with the source and noise respectively. The downstream test loss at convergence (for $\alpha_s = 1, \alpha_g = 0$) is*

$$\mathcal{L}(c_1, c_2, c_3, \nu_2) = (1 - \nu_2)\frac{(1 + c_3\nu_2)^2 + c_1^2\nu_2^2}{((1 + c_2\nu_2)(1 + c_3\nu_2) - c_1^2\nu_2^2)^2} \quad (13)$$

*which might can be higher than the baseline $\mathcal{L} = (1 - \nu_2)$ depending on $c_1, c_2, c_3$ respective values.*

With finite data, pre-training on $\mathcal{T}_1$ leads to an adaptive feature kernel as in Eq. 11 after a short path integral derivation (see Appendix C.2). Computing the constants $c_1, c_2, c_3$ is in principle hard, because it requires solving for correlations and response functions from DMFT at limiting time. We leave them as constants and derive conclusions for some interpretable cases. We do not expect, in general, transfer learning to have a positive effect when crosstalk and noise components $c_1, c_3$ grow large compared to $c_2$. In the population limit where $\nu_1 \to \infty$, we expect instead $\text{Cov}(\boldsymbol{g}) \to 0$, thus recovering the pure signal spike when there are no sample size fluctuations.

With this kernel, similarly to the sketch of Result 2, we study the limiting dynamics of the error field $\boldsymbol{v}_0(t) = \boldsymbol{\beta}_t - \boldsymbol{K}^{1/2}\hat{\boldsymbol{\beta}}(t)$. This time, together with the correlation functions $C_\Delta(t,t'), C_{v_0}(t,t')$ that

define train and test losses, we get contributions from $C_{sv}(t) = \frac{1}{D}\boldsymbol{\beta}_s \cdot \boldsymbol{v}_1(t)$ and $C_{gv}(t) = \frac{1}{D}\boldsymbol{g} \cdot \boldsymbol{v}_1(t)$ which we need to study at limiting time.

Because of the dependency on many variables (i.e., $\nu_2, \alpha_s, \alpha_g, c_1, c_2, c_3$), in Eq. 13 we report the loss in the special case where $\alpha_s = 1$ and $\alpha_g = 0$ (see general expression in the Appendix C.2). Notice that this reduces to the linear-probe baseline $\mathcal{L} = 1 - \nu_2$ for $c_1 = c_2 = c_3 = 0$; improves monotonically with $c_2$; and worsens with increasing crosstalk $c_1$ in this special case.

**Result 4 (Ultra-rich pretraining undermines fine-tuning performance)** *Consider the two-layer MLP of Eq. 1 with $L = 1$, $\phi(x) \equiv x$ and $\gamma_1 = \tilde{\gamma}_1/\sqrt{D}$ at infinite width. On $\mathcal{T}_1$, consider the balance condition $\partial_t(\boldsymbol{W}\boldsymbol{W}^\top - \boldsymbol{w}\boldsymbol{w}^\top) = 0$. When $\tilde{\gamma}_1 \to \infty$, or equivalently for small weight initialization $\boldsymbol{W}_0\boldsymbol{W}_0^\top \approx \boldsymbol{w}_0\boldsymbol{w}_0^\top$, then $\boldsymbol{W} = \boldsymbol{w}\boldsymbol{v}^\top$ is low-rank with $\boldsymbol{v} \in \mathbb{R}^D$. In the proportional regime $P_1 = \nu_1 D$, solve for $\boldsymbol{v}$ at limiting time through DMFT. The adaptive feature kernel after pre-training on $\mathcal{T}_1$ is $\boldsymbol{K}(\boldsymbol{X}, \boldsymbol{X}') \propto \boldsymbol{X}(\frac{1}{D}\boldsymbol{v}\boldsymbol{v}^\top)\boldsymbol{X}'^\top$, i.e.*

$$\boldsymbol{K}(\boldsymbol{X}, \boldsymbol{X}') = \boldsymbol{X}\left[\frac{\nu_1^2}{D}\boldsymbol{\beta}_s\boldsymbol{\beta}_s^\top + \frac{\nu_1(1-\nu_1)}{D}\boldsymbol{g}\boldsymbol{g}^\top + \frac{\nu_1\sqrt{\nu_1(1-\nu_1)}}{D}\left(\boldsymbol{\beta}_s\boldsymbol{g}^\top + \boldsymbol{g}\boldsymbol{\beta}_s^\top\right)\right]\boldsymbol{X}'^\top, \quad (14)$$

*which is a rank-one kernel with signal $\boldsymbol{\beta}_s$ and noise $\boldsymbol{g} \sim \mathcal{N}(0, \boldsymbol{I})$, such that $\boldsymbol{g} \perp \boldsymbol{\beta}_s$. A noiseless linear target $\boldsymbol{y}_t = \frac{1}{\sqrt{D}}\boldsymbol{X}_t^\top\boldsymbol{\beta}_t$ is exactly solvable iff $\boldsymbol{\beta}_t \in \text{span}\{\boldsymbol{v}\}$. Ultra-rich pre-training therefore collapses the features, and only the projection of $\boldsymbol{\beta}_t$ onto this collapsed subspace is learnable. The asymptotic test loss for $\nu_1 \in [0, 1]$ is*

$$\mathcal{L}(\nu_1, \alpha_s, \alpha_g) = 1 - (\sqrt{\nu_1}\alpha_s + \sqrt{1-\nu_1}\alpha_g)^2, \quad (15)$$

*with $\alpha_s = \frac{1}{D}\boldsymbol{\beta}_s \cdot \boldsymbol{\beta}_t, \alpha_g = \frac{1}{D}\boldsymbol{g} \cdot \boldsymbol{\beta}_t$ the alignments with the source and noise respectively. In the data-rich limit $\nu_1 \to 1$, the learned feature collapses to the signal ($\boldsymbol{v} \to \boldsymbol{\beta}_s$) and the downstream loss to $\mathcal{L} = 1 - \alpha_s^2$, which is the residual (unexplained) variance of $\boldsymbol{y}_t$.*

This result can be considered as a special case of Result 3, when there is no bulk component in the adaptive NTK after learning $\mathcal{T}_1$ (see Eq. 14 and Appendix C.3 for details). The loss of Eq. 15 does not depend on the amount of data $\nu_2$ in $\mathcal{T}_2$, since any dependency on $P_2$ comes from how well it is possible to estimate a single scalar coefficient in this rank-1 feature, which vanishes as $P_2 \to \infty$.

## 3 TRANSFER LEARNING PHENOMENOLOGY

In the following, we illustrate the interplay between transfer learning, feature learning strength, sample size and task similarity leveraging our theoretical results in Section 2. We start with the fine-tuning setting, where data on both $\mathcal{T}_1$ and $\mathcal{T}_2$ tasks are generated by linear target functions, and then proceed to the jointly rich setting, allowing feature learning on both tasks. By increasing the task complexity, we derive conclusions on the benefit of transfer learning from polynomial to real datasets.

### 3.1 FINE-TUNING

**Infinite data on $\mathcal{T}_1$** In the population risk limit from Result 2, when $\nu_1 \to \infty$, the test loss is a monotonically decreasing function of source/task alignment $\alpha$ (see Fig. 1(a)) and thus fine-tuning has always a positive gain from feature learning on $\mathcal{T}_1$.

**Finite data on $\mathcal{T}_1$** By contrast, when $\nu_1$ is finite the features learned on $\mathcal{T}_1$ are noisy because of finite sample size fluctuations: the adaptive NTK (see Eq. 11) acquires, in addition to the useful signal spike (controlled by $c_2$), both a noise spike (controlled by $c_3$) and a crosstalk term (proportional to $c_1$), as shown in Result 3. As a consequence, the test loss is no longer a decreasing function of source/task alignment $\alpha_s$ (see Fig 1(b)). If we suppose the target task having a non-zero alignment with the noise $\alpha_g \neq 0$, then transfer is most helpful in the low $\nu_2$ regime and when source/target similarity $\alpha_s$ is high; although, with enough data on $\mathcal{T}_2$, both noise and crosstalk terms can corrupt the signal direction, making it convenient to learn from scratch instead of using transfer learning.

The simple alignment case ($\alpha_s=1, \alpha_g=0$) of Eq. 13 shows that there $(i)$ larger $c_2$ always helps, while $(ii)$ $c_1$ always hurt, since it rotates the high-gain direction towards the noise. Instead $(iii)$ $c_3$ when the noise is uncorrelated with the target ($\alpha_g = 0$) act as a ridge (regularization effect) in high dimension (see Appendix C.2).

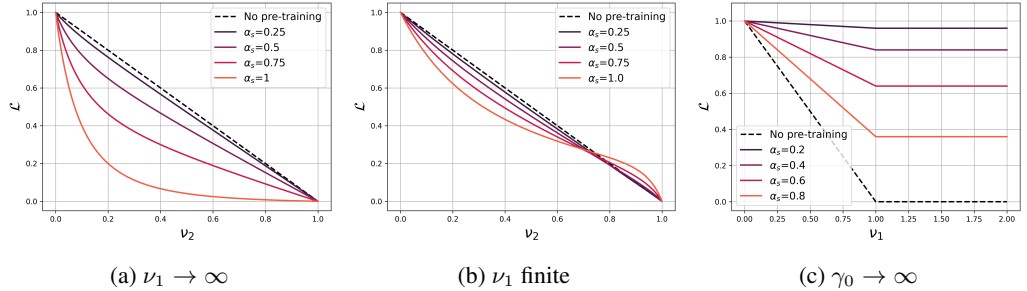

Figure 1: Fine-tuning from an adaptive kernel from $\mathcal{T}_1$. Dashed black: no pre-training (linear probe). (a) Loss is strictly decreasing with source/target alignment $\alpha_s$ (*Result 2*). (b) Non-zero alignment with the noise direction ($\alpha_g = 0.1$) can cause negative transfer at high $\nu_2 = P_2/D$ (*Result 3*). (c) Test loss on $\mathcal{T}_2$ depends only on source data $\nu_1 = P_1/D$ and the alignments $\{\alpha_g, \alpha_s\}$ (*Result 4*. In the panel, $\alpha_g = 0$).

**Large $\gamma_0$ on $\mathcal{T}_1$** Consistent with Eq. 15 of Result 4, when $\alpha_g = 0$ (Fig 1(c)), since $\alpha_s \in [-1, 1]$, then with this rank-one feature one can only learn up to $\mathcal{L} = 1 - \alpha_s^2$, and the perfect interpolation happens only when target task is perfectly aligned with the source task (i.e., $\alpha_s = 1$). This suggests that it is in principle harmful to have an infinitely rich pre-training. We show in Appendix C.4 that this is consistent with what happens when fine-tuning a non-linear model on polynomial tasks.

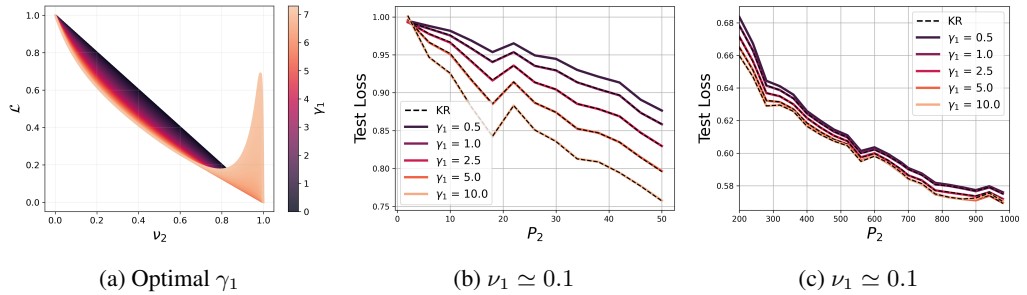

Figure 2: Fine-tuning with adaptive kernels from $\mathcal{T}_1$. Losses vs $\nu_2$ and for different $\gamma_1$ values on $\mathcal{T}_1$. (a) Linear model from Result 3 when $c_1 = \nu_1 \sqrt{\nu_1(1-\nu_1)}\chi, c_2 = \nu_1^2\chi, c_3 = \nu_1(1-\nu_1)\chi$ with $\chi = \sqrt{1 - \gamma_1^2} - 1$ has optimal $\gamma_1$ at large $\nu_2$. (b)/(c) Two-layer ReLU MLP on CIFAR10: source task is regression on $\{0, 1\}$ classes; target task is regression on $\{0, 9\}$ classes. Theory is obtained by performing kernel regression on $\mathcal{T}_2$ from the adaptive kernel after $\mathcal{T}_1$.

**Real datasets** To concretely show that most of the conclusions one can derive from our theoretical models of fine-tuning are still applicable to non-linear models, we make some phenomenological comparisons. As anticipated for finite $\nu_1$, our theory from Result 3 predicts that the constants $c_1, c_2, c_3$ are functions of feature strength $\gamma_1$ and $\nu_1$. We make an ansatz for these functions at large $\gamma_1$ inspired by model in Result 4. The test loss of Eq. 13 will be then a function $\mathcal{L}(\gamma_1, \nu_1, \nu_2)$. When $\nu_1$ is finite and so the alignment between noise and target tasks is non-zero (i.e., $\alpha_g \neq 0$), our theory in Fig. 2(a) predicts that the optimal feature-learning strength $\gamma_1^\star(\nu_2)$ is large when $\nu_2$ is small (variance reduction dominates), and it decrease as $\nu_2$ grows (bias from feature drift starts to hurt). At large $\nu_2$, there exists an optimal value of feature learning strength $\gamma_1$ that lowers the loss with respect to the baseline (see Fig. 2(a)). Similarly, after training a non-linear model on CIFAR-10 with different $\gamma_1$ on $\mathcal{T}_1$, Figs. 2(b)/(c) (which refers to performing kernel regression on $\mathcal{T}_2$ with the fixed kernel from $\mathcal{T}_1$, or equivalently to the *lazy* $\gamma_2 \to 0$ dynamics of Result 1) show that larger $\gamma_1$ yields lower test loss at small $P_2$ ($\propto \nu_2$), but the advantage shrinks and the curves collapse as $P_2$ increases; with enough target data, pre-training feature strength matters less. Again, consistently with our theory (*lazy* fine-tuning $\gamma_2 \to 0$ from Result 1), we also show in Fig. 9 that on polynomial task high $\gamma_2$ can be detrimental when $P_2$ is large.

## 3.2 TRANSFER LEARNING OF POLYNOMIAL TASKS WITH NONLINEAR ACTIVATIONS

**Low to High Degree Polynomials**  Kernel limits of neural networks are strongly biased to fit their data with low degree polynomials when data is high dimensional and isotropic. This spectral bias (Rahaman et al., 2019; Bordelon et al., 2020; Canatar et al., 2021b) reflects the fact that kernel methods learn eigenfunctions in order of decreasing eigenvalue (Novak et al., 2018; Belkin et al., 2019; Zhi-Qin John Xu et al., 2020). By contrast, networks trained in the feature-learning regime can learn sparse polynomials from much fewer data and training steps (Mei et al., 2018; Dandi et al., 2023b; Troiani et al., 2024; Dandi et al., 2024). The staircase property (Abbe et al., 2021; 2023; 2024; Yang et al., 2025) explored by Dandi et al. (2023b) makes this hierarchy explicit in multi-index polynomial settings.

Inspired by the utility of feature learning on sparse polynomials of Gaussian data $\boldsymbol{x} \sim \mathcal{N}(0, \boldsymbol{I})$, we study transfer from a linear source task to a quadratic target by employing the two-layer MLP model of Result 1 in the jointly rich setting. Figure 3(a) shows that pretraining on the linear task (right panel) lowers the test loss on the quadratic target compared to training from scratch (left panel). The feature-learning strength $\gamma_2$ on $\mathcal{T}_2$ here accelerates early gains but it also induces stronger forgetting of the source features during transfer learning, as pointed out in (Graldi et al., 2024). Eventually, there is an intermediate value of $\gamma_2$ that minimizes both target loss on $\mathcal{T}_2$ and catastrophic forgetting on $\mathcal{T}_1$.

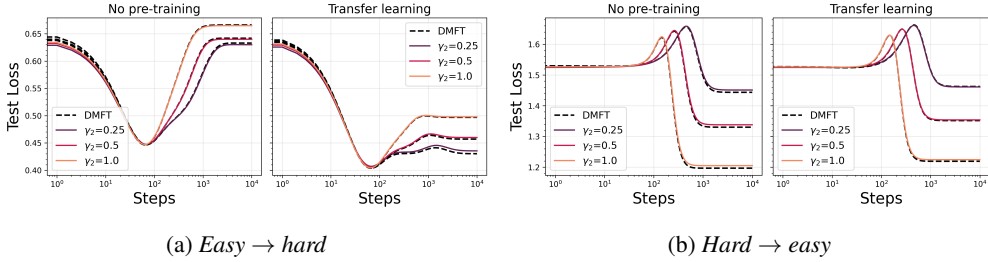

(a) *Easy → hard*                    (b) *Hard → easy*

Figure 3: Test losses of a two-layer ReLU MLP vs steps for different feature learning strength $\gamma_2$ on $\mathcal{T}_2$. (a) Low degree polynomial source task $y_1(\boldsymbol{x}) = D^{-1/2}\boldsymbol{\beta} \cdot \boldsymbol{x}$ with $P_1 = 1000$, $D = 100$ and $\gamma_1 = 1.0$. Target task is $y_2(\boldsymbol{x}) = (D^{-1/2}\boldsymbol{\beta} \cdot \boldsymbol{x})^2$ with $P_2 = 100$. (b) Source task $\mathrm{He}_5(\boldsymbol{\beta}_1 \cdot \boldsymbol{x})$ with $P_1 = 1000$ and $\gamma_1 = 1.0$. Target task: $\mathrm{He}_2(\boldsymbol{\beta}_2 \cdot \boldsymbol{x})$ with $P_2 = 600$ and $\boldsymbol{\beta}_1 \cdot \boldsymbol{\beta}_2 = 0.8$. Solid lines: gradient-descent on an $N = 20000$ two-layer ReLU network. Dashed lines: DMFT theory from 1.

**High to Low Degree Polynomials**  In Figure 3(b), we compare the model performances when learning a low degree Hermite polynomial target function from either a random initial condition or the features learned from a high degree Hermite source task. In both cases, learning the target is speeded up by feature learning strength $\gamma_2$. Similarly to a grokking phenomena (Power et al., 2022; Liu et al., 2022; Kumar et al., 2024; Fan et al., 2024), we conjecture that in this initial training phase the network begins memorizing its training set and slightly overfits, then after adapts features to the data, leading to improved test loss at late times. This adaptations of features happens faster when training with higher $\gamma_2$ (rich feature learning from Result 1). However, in this setting, because the pre-training on $\mathcal{T}_1$ makes the target model at initialization to rely on spurious high-frequency features components that are not needed by the simpler task $\mathcal{T}_2$, transfer learning has no benefit in this scenario compared to no pre-training performance.

## 3.3 ROLE OF TRANSFER LEARNING ON REAL DATASETS

Moving beyond synthetic tasks, we consider simple image regression problems. We start with CIFAR-10, where a model pre-trained on two source classes is then fine-tuned on two disjoint target classes. We compare the performance of a target model trained on this second task $\mathcal{T}_2$ from random initialization (Fig. 4(a)) with the performance of the same model when using features learned from a data-rich source $\mathcal{T}_1$ (Fig. 4(b)). Here, transfer learning leads to a lower test loss compared to no-pretraining for each value of feature learning strength $\gamma_2$. In both cases, there exists an optimal early stopping time which minimizes the loss before slightly overfitting. We show that our DMFT theory from Result 1 is well-predictive of this jointly rich setting. In Fig. 4(c) the distribution preactivations $p(h)$ of the target model shows that, as $\gamma_2$ grows large, feature learning makes $p(h)$ highly

non-Gaussian. In Appendix B we also show that, similarly to fine-tuning setting (i.e., linear probe) on real datasets (Fig. 2(b)/(c)), feature learning on $\mathcal{T}_1$ is crucial when downstream task is data-poor (small $P_2$); with large $P_2$ the model is able to rely more on supervision signals from the data itself and transfer learning offers little additional improvement.

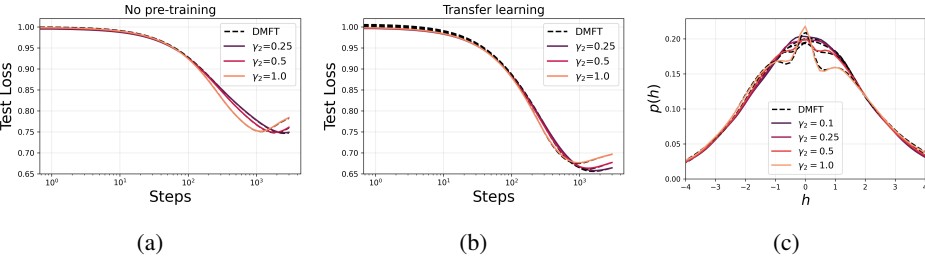

Figure 4: (a)/(b) Transfer learning is beneficial for real tasks at any feature learning strength $\gamma_2$. Source task: classes $1/2$ of CIFAR-10 with $P_1 = 10K$ and $\gamma_1 = 1.0$. Target task: classes $8/9$ of CIFAR-10 with $P_2 = 200$. (c) Preactivation distribution of the target model for different $\gamma_2$. Solid lines: GD at convergence ($N = 20000$, two-layer ReLU MLP); black dashed lines: DMFT from 1.

## 4  DISCUSSION AND CONCLUSION

In this work, we develop a theory of transfer learning in infinitely wide neural networks under gradient flow. First, we provide the theory for non-linear MLPs, in the general setting which enables feature learning on both pre-training and downstream tasks. Here, transfer learning on polynomial tasks outperforms no pre-training when moving from easy (low degree) to hard (high degree) benchmarks. No such gain is observed from hard to easy objectives, since the pre-trained model eventually biases the representation toward high-degree components that are misaligned with the low-degree task. On real vision tasks, transfer learning speeds up performance, showing a consistent improvement in test loss. Consistently throughout these benchmarks, feature learning on downstream tasks enhances performance with a data-limited target. Second, we study fine-tuning with fixed features from a pre-trained rich source. Our results illustrate how the source/target similarity, the amount on data and feature learning strength control the relative benefits of transfer learning compared to learning from scratch. Here, different pre-training regimes lead to different conclusions on fine-tuning benefits. (i) If source task is data-rich, fine-tuning is always beneficial; (ii) for finite source data, noise from finite samples can corrupt fine-tuning gains; (iii) when source task is infinitely rich, the target task is exactly solvable if and only if it is perfectly aligned with the source.

This work is limited for many reasons. Our data-average case of linear toy models rely on simplifying assumptions, such as isotropic data, that enable closed-form analysis but limit the scope of quantitative predictions. Relaxing these assumptions, for instance by incorporating data-averaged study of structured or heavy-tailed data, would help bridge the gap between our theoretical insights and the behavior of large-scale neural networks. Future works could also explore how representation learning in deeper networks enable transfer learning. Specifically, it could be interesting to study what number of hidden layers should be preserved during transfer learning (Bansal et al., 2021). Another possible future direction could be to connect our framework with curriculum learning, where tasks are organized in a structured sequence rather than treated independently; our theory could help clarify when and why such curricula improve generalization and feature reuse.

## 5  ACKNOWLEDGMENTS

The authors would like to thank Stefano Sarao Mannelli and Luca Saglietti, as well as the members of the Pehlevan Lab for insightful discussions. C.L. is supported by DARPA grants DIAL-FP-038 and AIQ-HR00112520041. B.B. acknowledges support from the Center of Mathematical Sciences and Applications (CMSA) of Harvard University. C.P. is supported by an NSF CAREER Award (IIS-2239780), DARPA grants DIAL-FP-038 and AIQ-HR00112520041, the Simons Collaboration on the Physics of Learning and Neural Computation, and the William F. Milton Fund from Harvard University. This work has been made possible in part by a gift from the Chan Zuckerberg Initiative Foundation to establish the Kempner Institute for the Study of Natural and Artificial Intelligence.

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

## A    PRIMER ON DMFT

Dynamical Mean Field Theory (DMFT) is a method from statistical physics for analyzing high-dimensional dynamical systems in the presence of a **quenched random disorder**. The disorder is "quenched" in the sense that it remains effectively *fixed* over the time scale of the dynamics. The method was originally introduced for classical spin-glass systems, where the disorder takes the form of random couplings between spins Sompolinsky & Zippelius (1981); in neural-network models it can arise from random connectivity Helias & Dahmen (2020), random data Mignacco et al. (2020); Gerbelot et al. (2024a), or random initial conditions for the weights Bordelon & Pehlevan (2022).

In our specific setting, the relevant source of disorder is the random initialization of the model parameters, which shapes the subsequent feature-learning dynamics. The key idea behind DMFT is that, in the limit of a large number of neurons (width $N \to \infty$), the high-dimensional coupled dynamics of all neurons becomes statistically equivalent to that of a single neuron evolving under an effective stochastic process. This reduction is possible because, at infinite width, each neuron decouple statistically and the interaction with the rest of the network appears only through macroscopic quantities (population averages), which become deterministic by the law of large numbers.

These macroscopic quantities are called **order parameters** in statistical physics jargon, and are respectively:

- **correlation functions**, which describe how the neuron's activity at different pairs of time $t, t'$ co-varies, capturing the temporal structure of the dynamics, and
- **response functions**, quantifying how sensitive the neuron's dynamics are to small perturbations in the input or the effective stochastic field. They measure how a tiny change introduced at time $t'$ influences the neuron activity at a later time $t$.

To make this paper self-contained, we report an explicit example on how to study gradient flow dynamics through DMFT for a linear regression problem with isotropic covariates, which is relevant for all the theoretical results we present in the main text.

### A.1 LINEAR REGRESSION WITH ISOTROPIC COVARIANCE

Let $\boldsymbol{X} \in \mathbb{R}^{N \times P}$ denote the data matrix, whose columns $\boldsymbol{x}_\mu \in \mathbb{R}^N$ are $P$ i.i.d. samples of an $N$-dimensional isotropic covariate, i.e. $\boldsymbol{x}_\mu \sim \mathcal{N}(0, \boldsymbol{I}_N)$ with $\mu = 1, \ldots, P$. We study the joint high dimensional limit $P, N \to \infty$ with $\nu = \frac{P}{N}$ fixed. A linear teacher generates the labels according to

$$\boldsymbol{y} = \frac{1}{\sqrt{N}} \boldsymbol{X}^\top \boldsymbol{\beta}_\star \tag{16}$$

where $\boldsymbol{\beta}_\star \in \mathbb{R}^N$ is the fixed target vector. A student with parameter vector $\boldsymbol{\beta}(t)$ is trained by gradient flow on the squared loss

$$\mathcal{L}(t) = \frac{1}{2P} ||\boldsymbol{X}^\top (\boldsymbol{\beta}_\star - \boldsymbol{\beta}(t))||^2 = \frac{1}{2P} ||\boldsymbol{\Delta}(t)||^2, \tag{17}$$

where we introduced the error vector $\boldsymbol{\Delta}(t)$ on the training set

$$\boldsymbol{\Delta}(t) = \frac{1}{\sqrt{N}} \boldsymbol{X}^\top \boldsymbol{v}_0(t), \quad \boldsymbol{v}_0(t) = \boldsymbol{\beta}_\star - \boldsymbol{\beta}(t). \tag{18}$$

Thus $\boldsymbol{v}_0(t)$ measures the current mismatch between student and teacher in parameter space. Gradient flow on $\boldsymbol{\beta}(t)$ reads

$$\frac{d}{dt} \boldsymbol{\beta}(t) = -\frac{1}{P} \boldsymbol{X} \boldsymbol{X}^\top (\boldsymbol{\beta}(t) - \boldsymbol{\beta}_\star). \tag{19}$$

In terms of the error vector $\boldsymbol{v}_0(t)$, this becomes

$$\frac{d}{dt} \boldsymbol{v}_0(t) = -\frac{\sqrt{N}}{P} \boldsymbol{X} \boldsymbol{\Delta}(t) + \delta(t) \boldsymbol{\beta}_\star \tag{20}$$
$$= -\boldsymbol{v}_1(t) + \delta(t) \boldsymbol{\beta}_\star.$$

where we introduced an initial condition $\delta(t)\boldsymbol{\beta}_\star$ and the field $\boldsymbol{v}_1(t)$. The key quantities which determine the generalization dynamics are the correlation functions

$$C_\Delta(t, t) = \frac{1}{P} \boldsymbol{\Delta}(t) \cdot \boldsymbol{\Delta}(t), \quad C_{v_0}(t, t) = \frac{1}{N} \boldsymbol{v}_0(t) \cdot \boldsymbol{v}_0(t) \tag{21}$$

which correspond to train and test losses respectively.

We aim to characterize the joint distribution of the fields $\{\boldsymbol{\Delta}(t), \boldsymbol{v}_0(t), \boldsymbol{v}_1(t)\}$ over draws of the random disorder $\mathcal{D} = \{\boldsymbol{X}\}$. To do so, we can start by defining the moment generating function of those fields with a Martin-Siggia-Rose integral over trajectories Martin et al. (1973)

$$\mathcal{Z}[\boldsymbol{j}_\Delta, \boldsymbol{j}_{\boldsymbol{v}_0}, \boldsymbol{j}_{\boldsymbol{v}_1}] = \left\langle \exp \left( \sum_t \left[ \boldsymbol{j}_\Delta(t) \cdot \boldsymbol{\Delta}(t) + \boldsymbol{j}_{\boldsymbol{v}_0} \cdot \boldsymbol{v}_0(t) + \boldsymbol{j}_{\boldsymbol{v}_1}(t) \cdot \boldsymbol{v}_1(t) \right] \right) \right\rangle_\mathcal{D}. \tag{22}$$

This object, once derived, enables computation of arbitrary moments from derivatives with respect to the $\boldsymbol{j}$ variables at $\boldsymbol{j} = 0$. For example, the two-point correlation between the $\boldsymbol{v}_0$ variables is given by

$$\langle v_{0,i}(t) v_{0,k}(t') \rangle_\mathcal{D} = \frac{\partial^2}{\partial j_{v_{0,i}}(t) \partial j_{v_{0,k}}(t')} \mathcal{Z}[\boldsymbol{j}_\Delta, \boldsymbol{j}_{\boldsymbol{v}_0}, \boldsymbol{j}_{\boldsymbol{v}_1}]. \tag{23}$$

Following a similar derivation to Bordelon & Pehlevan (2023), in the asymptotic limit, the Markovian (deterministic) system reduces to a low-dimensional stochastic non-Markovian system after the disorder average (neurons decouple statistically), i.e.

$$\partial_t v_0(t) = -u_1(t) - \int_0^t dt' R_\Delta(t, t') v_0(t') + \delta(t) \beta_\star, \quad u_1(t) \sim \mathcal{GP}\left(0, \frac{1}{\nu} C_\Delta\right) \tag{24}$$

$$\Delta(t) = u_\Delta(t) + \frac{1}{\nu} \int dt' R_{01}(t, t') \Delta(t'), \quad u_\Delta(t) \sim \mathcal{GP}(0, C_{v_0}) \tag{25}$$

$$C_\Delta(t, t') = \langle \Delta(t) \Delta(t') \rangle, \quad C_{v_0}(t, t') = \langle v_0(t) v_0(t') \rangle \tag{26}$$

$$R_\Delta(t, t') = \left\langle \frac{\partial \Delta(t)}{\partial u_\Delta(t')} \right\rangle, \quad R_{01}(t, t') = \left\langle \frac{\partial v_0(t)}{\partial u_1(t')} \right\rangle \tag{27}$$

where the Gaussian Process in Eq. 24 is disorder dependent (notice that this vanishes in the population limit $\nu \to \infty$), while the second term $\int dt' R_\Delta(t, t') v_0(t')$ is called **memory term**, since it depends on early stages of the system at times $t' < t$. The same is valid for the $\Delta(t)$ vector. The averages $\langle \cdot \rangle$ are over the random variables $\{u_1(t), u_\Delta(t)\}$.

### A.1.1 TEST LOSS AT LIMITING TIME

Since the system is linear, the response functions are T.T.I. By taking a Fourier transform, the DMFT equations becomes

$$i\omega v_0(\omega) = -u_1(\omega) - R_\Delta(\omega)v_0(\omega) + \beta_t \tag{28}$$

$$R_\Delta(\omega) = \left(1 - \nu^{-1}R_{01}\right)^{-1} \tag{29}$$

$$R_{01}(\omega) = -\mathcal{H}(\omega) \tag{30}$$

by defining $\mathcal{H}(\omega) = \frac{1}{i\omega + R_\Delta(\omega)}$. Solving for the error $v_0(\omega)$, this gives

$$v_0(\omega) = \frac{1}{i\omega + R_\Delta(\omega)}\Big[\beta_\star - u_1(\omega)\Big]. \tag{31}$$

Remembering the test loss definition of Eq. 232, we then get

$$C_{v_0}(\omega, \omega') = \langle v_0(\omega)v_0(\omega')\rangle$$
$$= \frac{\mathcal{H}(\omega)\mathcal{H}(\omega')}{1 - \nu^{-1}R_\Delta(\omega)R_\Delta(\omega')\mathcal{H}(\omega)\mathcal{H}(\omega')}. \tag{32}$$

At limiting time, we take the $\omega, \omega' \to 0$ limit of the loss $C_{v_0}(\omega, \omega')$

$$\lim_{t,t'\to\infty} C_{v_0}(t, t') = \lim_{\omega,\omega'\to 0}(i\omega)(i\omega')C_{v_0}(\omega, \omega'). \tag{33}$$

Using the equation

$$R_\Delta = 1 - \frac{1}{\nu}R_\Delta\mathcal{H} \implies \lim_{\omega\to 0} R_\Delta\mathcal{H} = \nu \tag{34}$$

and by noticing that

$$\lim_{\omega\to 0} i\omega\mathcal{H}(\omega) = \lim_{\omega\to 0} \frac{i\omega}{i\omega + \nu/(i\omega\mathcal{H})} = 1 - \nu \tag{35}$$

we can combine all the results to get the loss at convergence

$$\lim_{t\to\infty} C_{v_0,v_0}(t, t) = \frac{(i\omega\mathcal{H})(i\omega\mathcal{H})}{1 - \nu^{-1}R\mathcal{H}R\mathcal{H}} \tag{36}$$

$$= (1 - \nu). \tag{37}$$

## B  DEEP INFINITE WIDTH TRANSFER LEARNING DYNAMICS

Using the dynamical mean field theory techniques of Bordelon & Pehlevan (2022), we can track the dynamics of preactivations $h^\ell(x, t)$ and pre-gradients $z^\ell(x, t)$ which are defined as

$$h^{\ell+1}(x, t) = \frac{1}{\sqrt{N}}W^\ell(t)\phi(h^\ell(x, t))$$

$$g^\ell(x, t) = \dot{\phi}(h^\ell(x, t)) \odot z^\ell(x, t) , \ z^\ell(x, t) = \frac{1}{\sqrt{N}}W^\ell(t)^\top g^{\ell+1}(x, t). \tag{38}$$

We also introduce the variables $\Delta(x, t) = -\frac{\partial}{\partial f(x,t)}\ell(f(x, t), y(x))$, which for mean square error is simple $y(x) - f(x)$. On task one $\mathcal{T}_1$ and times $t \in (0, t_1)$ we have

$$h^\ell(x, t) = u^\ell(x, t) + \gamma_1\int dx'\int_0^t dt' \left[A^{\ell-1}(x, x', t, t') + p_1(x')\Delta(x', t')\Phi^{\ell-1}(x, x', t, t')\right]g^\ell(x', t')$$

$$z^\ell(x, t) = r^\ell(x, t) + \gamma_1\int dx'\int_0^t dt' \left[B^\ell(x, x', t, t') + p_1(x')\Delta(x', t')G^{\ell+1}(x, x', t, t')\right]\phi(h^\ell(x', t'))$$

$$p_1(x) = \frac{1}{P_1}\sum_{x'\in\mathcal{T}_1}\delta(x - x') , \ u^\ell \sim \mathcal{GP}(0, \Phi^{\ell-1}) , \ r^\ell \sim \mathcal{GP}(0, G^{\ell+1}) \tag{39}$$

where the correlation functions $\Phi^\ell, G^\ell$ are defined as

$$\Phi^\ell(\boldsymbol{x}, \boldsymbol{x}', t, t') = \left\langle \phi(h^\ell(\boldsymbol{x}, t))\phi(h^\ell(\boldsymbol{x}', t')) \right\rangle \ , \ G^\ell(\boldsymbol{x}, \boldsymbol{x}', t, t') = \left\langle g^\ell(\boldsymbol{x}, t)g^\ell(\boldsymbol{x}', t') \right\rangle \qquad (40)$$

and the response functions are

$$A^\ell(\boldsymbol{x}, \boldsymbol{x}', t, t') = \left\langle \frac{\delta\phi(h^\ell(\boldsymbol{x}, t))}{\delta r^\ell(\boldsymbol{x}', t')} \right\rangle \ , \ B^\ell(\boldsymbol{x}, \boldsymbol{x}', t, t') = \left\langle \frac{\delta g^\ell(\boldsymbol{x}, t)}{\delta u^\ell(\boldsymbol{x}', t')} \right\rangle . \qquad (41)$$

On task-2 where $t \in (t_1, t_2)$ we have the following dynamics

$$h^\ell(\boldsymbol{x}, t) = u^\ell(\boldsymbol{x}, t) + \gamma_1 \int d\boldsymbol{x}' \int_0^{t_1} dt' \left[ A^{\ell-1}(\boldsymbol{x}, \boldsymbol{x}', t, t') + p_1(\boldsymbol{x}')\Delta(\boldsymbol{x}', t')\Phi^{\ell-1}(\boldsymbol{x}, \boldsymbol{x}', t, t') \right] g^\ell(\boldsymbol{x}', t')$$

$$+ \gamma_2 \int d\boldsymbol{x}' \int_{t_1}^t dt' \left[ A^{\ell-1}(\boldsymbol{x}, \boldsymbol{x}', t, t') + p_2(\boldsymbol{x}')\Delta(\boldsymbol{x}', t')\Phi^{\ell-1}(\boldsymbol{x}, \boldsymbol{x}', t, t') \right] g^\ell(\boldsymbol{x}', t')$$

$$z^\ell(\boldsymbol{x}, t) = r^\ell(\boldsymbol{x}, t) + \gamma_1 \int d\boldsymbol{x}' \int_0^{t_1} dt' \left[ B^\ell(\boldsymbol{x}, \boldsymbol{x}', t, t') + p_2(\boldsymbol{x}')\Delta(\boldsymbol{x}', t')G^{\ell+1}(\boldsymbol{x}, \boldsymbol{x}', t, t') \right] \phi(h^\ell(\boldsymbol{x}', t'))$$

$$+ \gamma_2 \int d\boldsymbol{x}' \int_{t_1}^t dt' \left[ A^{\ell-1}(\boldsymbol{x}, \boldsymbol{x}', t, t') + p_2(\boldsymbol{x}')\Delta(\boldsymbol{x}', t')G^{\ell+1}(\boldsymbol{x}, \boldsymbol{x}', t, t') \right] \phi(h^\ell(\boldsymbol{x}', t')) \qquad (42)$$

where $p_2(\boldsymbol{x}) = \frac{1}{P_2} \sum_{\boldsymbol{x}' \in \mathcal{T}_2} \delta(\boldsymbol{x} - \boldsymbol{x}')$. The $\Delta(\boldsymbol{x}, t)$ features for $t \in (t_1, t_2)$ takes the form

$$\frac{d}{dt} f(\boldsymbol{x}, t) = \sum_\ell \mathbb{E}_{\boldsymbol{x}'} G^{\ell+1}(\boldsymbol{x}, \boldsymbol{x}', t, t)\Phi^\ell(\boldsymbol{x}, \boldsymbol{x}', t, t)\Delta(\boldsymbol{x}', t') \ , \ f(\boldsymbol{x}, t_1) = 0. \qquad (43)$$

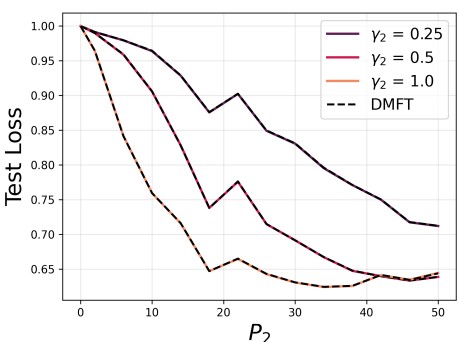

Figure 5: Test losses as a function of target data $P_2$ for different feature learning strength $\gamma_2$ on downstream task. Source task is a regression on two classes (0/1) of CIFAR with $P_1 = 1000$ labels $\bar{y} \in \{-1, 1\}^{P_1}$ and richness $\gamma_1 = 1.0$. Target task is a regression on two classes of CIFAR (0/9) with $P_2$ data points and labels $y \in \{-1, 1\}^{P_2}$.

## C  TOY MODELS OF FINE-TUNING IN THE PROPORTIONAL REGIME

In the current section, we will develop theories of transfer learning in the proportional regime, i.e. by allowing the data on both source and target tasks to grow arbitrarily large $P_1, P_2 \to \infty$, such that $\nu_2 = \frac{P_2}{D} = \Theta_D(1)$ is fixed, with $D$ input dimension. In the following, we will make three distinctions regarding the source task $\mathcal{T}_1$. In general, $\mathcal{T}_1$ is defined by a teacher model $\boldsymbol{\beta}_s \in \mathbb{R}^D$

$$y_{s,\mu} = \frac{1}{\sqrt{D}}\boldsymbol{\beta}_s \cdot \boldsymbol{x}_\mu \qquad (44)$$

for random isotropic data $\boldsymbol{x}_\mu \sim \mathcal{N}(0, \boldsymbol{I})$ and labels $|\boldsymbol{y}_s|^2 = 1$. The student is instead a two-layer model

$$f(\boldsymbol{x}_\mu) = \frac{\sqrt{D}}{N\gamma_0}\boldsymbol{a}^\top \left( \frac{1}{\sqrt{D}}\boldsymbol{W} \right)\boldsymbol{x}_\mu \qquad (45)$$

with $\boldsymbol{W} \in \mathbb{R}^{N \times D}, \boldsymbol{a} \in \mathbb{R}^N$ whose dynamics we study at limiting time $t \to \infty$ after learning with gradient flow (GF) and from random initial conditions $W_{ij}(0), a_j(0) \sim \mathcal{N}(0, 1)$. Depending on $P_1$, pretraining learns either (i) a single rank-one spike aligned with the signal direction $\boldsymbol{\beta}_s$ (population regime), or (ii) a finite rank deformation composed of the aligned spike plus several spikes correlated with a noise direction $\boldsymbol{g} \in \mathbb{R}^D$ and independent on the source direction $\boldsymbol{\beta}_s$. For this reason, we make distinctions in pretraining with the following scenarios: infinite data on $\mathcal{T}_1$ (i.e., $\nu_1 \to \infty$); limited data on $\mathcal{T}_1$ (i.e., finite $\nu_1$), and feature learning strength $\gamma_0 \to \infty$ on $\mathcal{T}_1$. In each of these settings, we wonder if the NTK kernels after feature learning on $\mathcal{T}_1$ have either a positive or a negative effect on transfer learning. For that, we consider a downstream task $\mathcal{T}_2$ defined by a target rule

$$y_{t,\mu} = \frac{1}{\sqrt{D}}\boldsymbol{\beta}_t \cdot \boldsymbol{x}_\mu \tag{46}$$

with $\boldsymbol{x}_\mu \in \mathcal{N}(0, 1), \boldsymbol{\beta}_t \in \mathbb{R}^D, |\boldsymbol{y}_t|^2 = 1$ and a fixed $\nu_2 = \frac{P_2}{D}$. We study gradient flow (GF) with the final NTK kernels from $\mathcal{T}_1$, and in each case the dependency of the loss of $\mathcal{T}_2$ on the amount of data $\{\nu_1, \nu_2\}$, the source/target alignment $\frac{1}{D}\boldsymbol{\beta}_t \cdot \boldsymbol{\beta}_s = \alpha$, and the feature learning strength $\gamma_0$.

## C.1 INFINITE DATA ON $\mathcal{T}_1$

As pointed out in (Bordelon & Pehlevan, 2022), by sending width $N \to \infty$ first at fixed $P_1$, the dynamics of a model such as Eq. 45 with $\boldsymbol{\theta} = \text{Vec}\{\boldsymbol{W}, \boldsymbol{a}\}$ can be studied through the lens of dynamical mean field theory (DMFT). If we choose a MSE loss on $\mathcal{T}_1$, i.e. $\mathcal{L} = \frac{1}{2P_1}\sum_{\mu=1}^{P_1}(y_{s,\mu} - f_\mu)^2$, and study gradient flow $\frac{d}{dt}\boldsymbol{\theta} = -\gamma^2 \nabla_\theta \mathcal{L}$ from random initial conditions $W_{ij}(0), a_j(0) \sim \mathcal{N}(0, 1)$, we get that one of the summary statistics we can track is the feature kernel $\boldsymbol{K}(t) = \left\langle \boldsymbol{h}(t)\boldsymbol{h}(t)^\top \right\rangle \in \mathbb{R}^{P_1 \times P_1}$, with $\boldsymbol{h}_\mu(t) \equiv \frac{1}{\sqrt{D}}\boldsymbol{W}(t)\boldsymbol{x}_\mu$ being the preactivation vector. With isotropic data $\boldsymbol{x}_\mu \in \mathcal{N}(0, 1)$, it is possible to show that the kernel $\boldsymbol{K}(t)$ only grows in the rank one source direction $\boldsymbol{y}_s \boldsymbol{y}_s^\top$ (see (Bordelon & Pehlevan, 2022) for a complete derivation). In particular, the limiting kernel has the form $\lim_{t \to \infty} \boldsymbol{K}(t) = \boldsymbol{I} + \chi \boldsymbol{y}_s \boldsymbol{y}_s^\top$, with $\chi = \sqrt{1 + \gamma_0^2} - 1$ which is an increasing function of the feature learning strength $\gamma_0$.

Now, if we allow the $\mathcal{T}_1$ dataset $P_1 \to \infty$ at fixed $D$, by averaging over the data distribution we get a kernel after feature learning on $\mathcal{T}_1$ which has the form

$$\boldsymbol{K}(\boldsymbol{x}, \boldsymbol{x}') = \boldsymbol{x}^\top \left[\boldsymbol{I} + \frac{\chi}{D}\boldsymbol{\beta}_s\boldsymbol{\beta}_s^\top\right]\boldsymbol{x}' \tag{47}$$

where we recall $\boldsymbol{\beta}_s$ being the source task vector. After pretraining on $\mathcal{T}_1$, the adaptive feature kernel $\boldsymbol{K}(\boldsymbol{x}, \boldsymbol{x}')$ is fixed. Thus, at infinite width, fine-tuning on $\mathcal{T}_2$ is equivalent to kernel regression with this frozen kernel (see Jacot et al. (2020)). The natural objective in this regime is the mean-squared error

$$\mathcal{L}(t) = \frac{1}{2P_2}||\boldsymbol{f}_2(\boldsymbol{X}_t) - \boldsymbol{y}_t||^2 \tag{48}$$

and using the feature representation from $\mathcal{T}_1$ this becomes

$$\mathcal{L}(t) = \frac{1}{2P_2}|\boldsymbol{X}_t^\top \left[\boldsymbol{I} + \frac{\chi}{D}\boldsymbol{\beta}_s\boldsymbol{\beta}_s^\top\right]^{1/2} \hat{\boldsymbol{\beta}}(t) - \boldsymbol{X}_t^\top \boldsymbol{\beta}_t|^2 \tag{49}$$

with $\boldsymbol{X}_t \in \mathbb{R}^{D \times P_2}$. This leads to

$$\frac{d}{dt}\hat{\boldsymbol{\beta}}(t) = -\frac{\partial \mathcal{L}}{\partial \hat{\boldsymbol{\beta}}} \tag{50}$$

from which, by defining $\boldsymbol{v}_0(t) = \boldsymbol{\beta}_t - \left[\boldsymbol{I} + \frac{\chi}{D}\boldsymbol{\beta}_s\boldsymbol{\beta}_s^\top\right]^{1/2}\hat{\boldsymbol{\beta}}(t)$, we get

$$\frac{d}{dt}\boldsymbol{v_0}(t) = -\left(\boldsymbol{I} + \frac{\chi}{D}\boldsymbol{\beta}_s\boldsymbol{\beta}_s^\top\right)\frac{\boldsymbol{X}\boldsymbol{X}^\top}{P}\boldsymbol{v}_0 + \delta(t)\boldsymbol{\beta}_t. \tag{51}$$

where $\delta(t)$ is a Dirac Delta function. We can introduce the following auxiliary fields

$$\boldsymbol{\Delta} = \frac{1}{\sqrt{D}}\boldsymbol{X}^\top \boldsymbol{v}_0 \in \mathbb{R}^P \tag{52}$$

$$\boldsymbol{v}_1 = \frac{\sqrt{D}}{P}\boldsymbol{X}\boldsymbol{\Delta} \in \mathbb{R}^D \tag{53}$$

$$C_{sv} = \frac{1}{D}\boldsymbol{\beta}_s \cdot \boldsymbol{v}_1 \tag{54}$$

which are self-averaging in the asymptotic limit $P, D \to \infty$ due to concentration of measure over the high-dimensional indices. The above dynamics becomes

$$\frac{d}{dt} \boldsymbol{v}_0 = -\boldsymbol{v}_1(t) - \chi \boldsymbol{\beta}_s C_{sv}(t) + \delta(t) \boldsymbol{\beta}_t. \tag{55}$$

with initial condition $\boldsymbol{v}_0(0) = \boldsymbol{\beta}_t$.

### C.1.1 DATA AVERAGE

Our goal is to track the statistics of the random field $\boldsymbol{v}_0$ at limiting time, from which we will be able to recover the loss function $\mathcal{L}$ at convergence. Once we average over the random $\mathcal{T}_2$ dataset, we expect this to depend on the finite sample fluctuations of $\mathcal{T}_2$ since $\nu_2 = \frac{P_2}{D}$ is fixed, and on the alignment with the pretraining source which we is controlled by a hyperparameter $\alpha = \frac{1}{D} \boldsymbol{\beta}_t \cdot \boldsymbol{\beta}_s$.

In order to do that, we develop a DMFT or path integral derivation (Agoritsas et al., 2018; Sarao Mannelli et al., 2020; Mignacco et al., 2020; Mignacco & Urbani, 2022; Gerbelot et al., 2024b; Dandi et al., 2023a; Bordelon & Pehlevan, 2022; Bordelon et al., 2024a).

First, we enforce the definitions of the fields and the $\boldsymbol{v}_0$ dynamics by functional $\delta$-constraints with conjugate fields $\{\hat{\boldsymbol{v}}_0, \hat{\boldsymbol{\Delta}}, \hat{\boldsymbol{v}}_1, \hat{C}_{sv}\}$. The resulting moment generating function (MGF) $\mathcal{Z}$ of DMFT depends linearly on the data matrix $\boldsymbol{X}$

$$\mathcal{Z} = \int \frac{dC_{sv} d\hat{C}_{sv}}{2\pi} \int \frac{d\boldsymbol{v}_0 d\hat{\boldsymbol{v}}_0}{2\pi} \int \frac{d\boldsymbol{\Delta} d\hat{\boldsymbol{\Delta}}}{2\pi} \int \frac{d\boldsymbol{v}_1 d\hat{\boldsymbol{v}}_1}{\pi} \exp\left[i \int dt \hat{\boldsymbol{v}}_0 \cdot \left(\partial_t \boldsymbol{v}_0 + \boldsymbol{v}_1 + \chi \boldsymbol{\beta}_s C_{sv}(t) - \delta(t) \boldsymbol{\beta}_t\right)\right]$$

$$\times \exp\left(-i \int dt \hat{\boldsymbol{\Delta}} \cdot \left(\frac{1}{\sqrt{D}} \boldsymbol{X}^\top \boldsymbol{v}_0\right) - i \int dt \hat{\boldsymbol{v}}_1 \cdot \left(\frac{\sqrt{D}}{P} \boldsymbol{X} \boldsymbol{\Delta}\right)\right)$$

$$\times \exp\left(i \int dt \left(\boldsymbol{\Delta} \hat{\boldsymbol{\Delta}} + \boldsymbol{v}_1 \hat{\boldsymbol{v}}_1\right) + i \int dt \hat{C}_{sv}(t) \left(C_{sv}(t) - \frac{1}{D} \boldsymbol{\beta}_s \cdot \boldsymbol{v}_1\right)\right). \tag{56}$$

Since the entries $x_{\mu,i} \sim \mathcal{N}(0,1)$ are i.i.d., we can average over the data distribution

$$\left\langle \exp\left[-i \int dt \mathrm{Tr} \boldsymbol{X}^\top \left(\frac{1}{\sqrt{D}} \boldsymbol{v}_0 \hat{\boldsymbol{\Delta}}^\top + \frac{\sqrt{D}}{P} \hat{\boldsymbol{v}}_1 \boldsymbol{\Delta}^\top\right)\right]\right\rangle_{\boldsymbol{X}}$$

$$= \exp\left(-\frac{1}{2} \int dt dt' \left[\frac{1}{D} \boldsymbol{v}_0(t) \cdot \boldsymbol{v}_0(t') \hat{\boldsymbol{\Delta}}(t) \cdot \hat{\boldsymbol{\Delta}}(t') + \frac{1}{\nu} \frac{1}{P} \boldsymbol{\Delta}(t) \cdot \boldsymbol{\Delta}(t') \hat{\boldsymbol{v}}_1(t) \cdot \hat{\boldsymbol{v}}_1(t')\right]\right) \tag{57}$$

$$\times \exp\left(\int dt dt' \frac{1}{P} \boldsymbol{\Delta}(t) \cdot \hat{\boldsymbol{\Delta}}(t') \boldsymbol{v}_0(t) \cdot \hat{\boldsymbol{v}}_1(t')\right).$$

By defining the correlation and response functions

$$C_{v_0,v_0}(t,t') \equiv \frac{1}{D} \boldsymbol{v}_0(t) \cdot \boldsymbol{v}_0(t') \tag{58}$$

$$C_{\Delta,\Delta}(t,t') \equiv \frac{1}{P} \boldsymbol{\Delta}(t) \cdot \boldsymbol{\Delta}(t') \tag{59}$$

$$R_{\Delta,\hat{\Delta}}(t,t') \equiv -\frac{i}{P} \boldsymbol{\Delta}(t) \cdot \hat{\boldsymbol{\Delta}}(t') \tag{60}$$

$$R_{v_0,\hat{v}_1}(t,t') \equiv -\frac{i}{D} \boldsymbol{v}_0(t) \cdot \hat{\boldsymbol{v}}_1(t') \tag{61}$$

we can enforce their definitions with the use of delta functions, for instance

$$1 \equiv \int \frac{dC_{v_0,v_0}(t,t') d\hat{C}_{v_0,v_0}(t,t')}{2\pi D^{-1}} \exp\left(\frac{D}{2} C_{v_0,v_0}(t,t') \hat{C}_{v_0,v_0}(t,t') - \frac{1}{2} \hat{C}_{v_0,v_0}(t,t') \boldsymbol{v}_0(t) \cdot \boldsymbol{v}_0(t')\right) \tag{62}$$

thus getting

$$
\begin{aligned}
\mathcal{Z} = \int \frac{dC_{sv}(t)d\hat{C}_{sv}(t)}{2\pi} \int \frac{dC_{v_0,v_0(t,t')}d\hat{C}_{v_0,v_0}(t,t')}{2\pi} \int \frac{dC_{\Delta,\Delta}(t,t')d\hat{C}_{\Delta,\Delta}(t,t')}{2\pi} \int \frac{dR_{\Delta,\hat{\Delta}}(t,t')d\hat{R}_{\Delta,\hat{\Delta}}(t,t')}{2\pi} \\
\times \int \frac{dR_{v_0,\hat{v}_1}(t,t')d\hat{R}_{v_0,\hat{v}_1}(t,t')}{2\pi} \exp\left[\frac{D}{2}\int dtC_{sv}(t)\hat{C}_{sv}(t) + \frac{D}{2}\int dtdt'C_{v_0,v_0}(t,t')\hat{C}_{v_0,v_0}(t,t')\right] \\
\times \exp\left[\frac{\nu_2 D}{2}\int dtdt'C_{\Delta,\Delta}(t,t')\hat{C}_{\Delta,\Delta}(t,t') - \nu_2 D\int dtdt'R_{\Delta,\hat{\Delta}}(t,t')\hat{R}_{\Delta,\hat{\Delta}}(t,t')\right] \\
\times \exp\left[-D\int dtdt'R_{v_0,\hat{v}_1}(t,t')\hat{R}_{v_0,\hat{v}_1}(t,t') + D\int dtdt'R_{\Delta,\hat{\Delta}}(t,t')R_{v_0,\hat{v}_1}(t,t')\right] \\
\times \exp\left[\sum_{i=1}^{D}\ln\mathcal{Z}_{01}\Big[C_{\Delta,\Delta}, C_{sv}, \hat{C}_{sv}, R_{\Delta,\hat{\Delta}}\Big] + \sum_{j=1}^{P}\ln\mathcal{Z}_\Delta\Big[C_{v_0,v_0}, R_{v_0,\hat{v}_1}\Big]\right]
\end{aligned}
\tag{63}
$$

where we collect every single site action (factorized respectively over input neurons and patterns)

$$
\begin{aligned}
\mathcal{Z}_{01}\Big[C_{\Delta,\Delta}, C_{sv}, \hat{C}_{sv}, \hat{C}_{v_0,v_0}, R_{\Delta,\hat{\Delta}}\Big] = \int \frac{dv_0 d\hat{v}_0}{2\pi} \int \frac{Dv_1 D\hat{v}_1}{2\pi} \exp\left[-\frac{1}{2}\int dt\hat{C}_{sv}(t)\beta_s v_1(t)\right] \\
\times \exp\left[-\frac{1}{2}\int dtdt'\hat{C}_{v_0,v_0}v^0(t)v^0(t') - \frac{1}{2\nu}\int dtdt'C_{\Delta,\Delta}\hat{v}_1(t)\hat{v}_1(t')\right] \\
\times \exp\left[-i\int dtdt'R_{\Delta,\hat{\Delta}}v_0(t)\hat{v}_1(t') + i\int dtv_1(t)\hat{v}_1(t)\right] \\
\times \exp\left[+i\int dt\hat{v}_0\Big(\partial_t v_0 + v_1 + \chi\beta_s C_{sv}(t) - \delta(t)\beta_t\Big)\right]
\end{aligned}
\tag{64}
$$

$$
\begin{aligned}
\mathcal{Z}_\Delta\Big[C_{v_0,v_0}, R_{v_0,\hat{v}_1}, \hat{C}_{\Delta,\Delta}\Big] = \int \frac{d\Delta d\hat{\Delta}}{2\pi} \exp\left[-\frac{1}{2}\int dtdt'\hat{C}_{\Delta,\Delta}(t,t')\Delta(t)\Delta(t')\right] \\
\times \exp\left[-\frac{1}{2}\int dtdt'C_{v_0,v_0}(t,t')\hat{\Delta}(t)\hat{\Delta}(t') - \frac{i}{\nu_2}\int dtdt'R_{v_0,\hat{v}_1}\Delta(t)\hat{\Delta}(t')\right] \\
\times \exp\left[+i\int dt\Delta(t)\hat{\Delta}(t)\right].
\end{aligned}
\tag{65}
$$

### C.1.2 DMFT ACTION

We now group all of the correlation and response functions, as well as their conjugate order parameters into a list named $\boldsymbol{q}$. The MGF can be written in the compact form

$$
\mathcal{Z} = \int d\boldsymbol{q}\exp\Big(-D\mathcal{S}(\boldsymbol{q})\Big)
\tag{66}
$$

where $\mathcal{S}$ is the $\mathcal{O}(1)$ DMFT action

$$
\begin{aligned}
\mathcal{S} = -\frac{1}{2}\int dtC_{sv}(t)\hat{C}_{sv}(t) - \frac{1}{2}\int dtdt'C_{v_0,v_0}(t,t')\hat{C}_{v_0,v_0}(t,t') - \frac{\nu_2}{2}\int dtdt'C_{\Delta,\Delta}(t,t')\hat{C}_{\Delta,\Delta}(t,t') \\
+ \int dtdt'R_{\Delta,\hat{\Delta}}R_{v_0,\hat{v}_1} - \frac{1}{D}\sum_{i=1}^{D}\ln\mathcal{Z}_{01}\Big[C_{\Delta,\Delta}, C_{sv}, \hat{C}_{sv}, R_{\Delta,\hat{\Delta}}\Big] - \frac{1}{D}\sum_{j=1}^{P}\ln\mathcal{Z}_\Delta\Big[C_{v_0,v_0}, R_{v_0,\hat{v}_1}\Big].
\end{aligned}
\tag{67}
$$

As $D \to \infty$, the moment-generating function $\mathcal{Z}$ is exponentially dominated by the saddle point of $\mathcal{S}$. The equations that define this saddle point also define our DMFT. First of all, we realize that at the saddle point

$$\hat{R}_{\Delta,\hat{\Delta}} = \frac{1}{\nu_2} R_{v_0,\hat{v}_1} \tag{68}$$

$$\hat{R}_{v_0,\hat{v}_1} = R_{\Delta,\hat{\Delta}}. \tag{69}$$

The resulting equations $\frac{\partial \mathcal{S}}{\partial \boldsymbol{q}} = 0$ give

$$-\frac{1}{2}C_{sv}(t) + \frac{1}{2D}\sum_{i=1}^{D}\left\langle \beta_s v_1(t) \right\rangle_i = 0 \tag{70}$$

$$-\frac{1}{2}C_{v_0,v_0}(t,t') + \frac{1}{2D}\sum_{i=1}^{D}\left\langle v^0(t)v^0(t') \right\rangle_i = 0 \tag{71}$$

$$-\frac{\nu_2}{2}C_{\Delta,\Delta}(t,t') + \frac{1}{2D}\sum_{j=1}^{P}\left\langle \Delta(t)\Delta(t') \right\rangle_j = 0. \tag{72}$$

Here, $\langle \rangle_i$ represents an average over the single site distribution defined by the moment generating function $\mathcal{Z}_{01}$. Similarly, $\langle \rangle_j$ is the average over the distribution defined by $\mathcal{Z}_\Delta$. Regarding the response functions we have

$$R_{\Delta,\hat{\Delta}} + \frac{i}{P}\sum_{j=1}^{P}\left\langle \Delta(t)\hat{\Delta}(t') \right\rangle_j = 0 \tag{73}$$

$$R_{v_0,\hat{v}_1} + \frac{i}{D}\sum_{i=1}^{D}\left\langle v_0(t)\hat{v}_1(t') \right\rangle_i = 0 \tag{74}$$

Lastly, we have a collection of saddle point equations that defines the conjugated order parameters, which must vanish at the saddle point (Crisanti & Sompolinsky, 2018; Bordelon & Pehlevan, 2022)

$$\hat{C}_{sv}(t) = \hat{C}_{v_0,v_0} = \hat{C}_{\Delta,\Delta}(t,t') = 0. \tag{75}$$

### C.1.3 HUBBARD TRANSFORMATION

Since we know that the correlation and response functions must take deterministic values in the limit $D \to \infty$, we can represent the quadratic terms in the log-density in $\hat{v}_1, \hat{\Delta}(t)$ as linear averages over Gaussian variables $u_1(t), u_\Delta(t)$

$$\exp\left(-\frac{1}{2\nu}\int dt dt' C_{\Delta,\Delta}\hat{v}_1(t)\hat{v}_1(t')\right) = \left\langle \exp\left(-i\int dt\hat{v}_1(t)u_1(t)\right) \right\rangle_{u_1 \sim \mathcal{N}(0,\frac{1}{\nu_2}C_{\Delta,\Delta})} \tag{76}$$

$$\exp\left(-\frac{1}{2}\int dt dt' C_{v_0,v_0}(t,t')\hat{\Delta}(t)\hat{\Delta}(t')\right) = \left\langle \exp\left(-i\int dt\hat{\Delta}(t)u_\Delta(t)\right) \right\rangle_{u_\Delta \sim \mathcal{N}(0,C_{v_0,v_0})}. \tag{77}$$

After introducing these Gaussian random variables, we can solve the integrals over the conjugated fields $\hat{v}_0, \hat{v}_1, \hat{\Delta}$, and obtain the defining equations for the random variables of interest

$$v_1(t) = u_1(t) + \int dt' R_{\Delta,\hat{\Delta}}(t,t')v_0(t') \tag{78}$$

$$\partial_t v_0 = -u_1(t) - \int dt' R_{\Delta,\hat{\Delta}}(t,t')v_0(t') - \chi\beta_s C_{sv}(t) + \delta(t)\beta_t \tag{79}$$

$$\Delta(t) = u_\Delta(t) + \frac{1}{\nu_2}\int dt' R_{v_0,\hat{v}_1}(t,t')\Delta(t'). \tag{80}$$

### C.1.4 SIMPLIFYING THE RESPONSE FUNCTIONS

From the saddle point equations, we notice that the response functions involve averages over the conjugated variables $\{\hat{\Delta}, \hat{v}_1\}$, which we now argue can be replaced as derivatives with respect to the Hubbard variables. For instance

$$
\begin{aligned}
R_{\Delta,\hat{\Delta}}(t,t') &= -i \int \prod_t \frac{d\Delta(t)d\hat{\Delta}(t)}{2\pi} \Delta(t)\hat{\Delta}(t') \Big\langle \exp\Big(i\int dt\hat{\Delta}(t)\Big[\Delta(t) - u_\Delta(t) - \frac{1}{\nu_2}\int dt' R_{v_0,\hat{v}_1}(t,t')\Delta(t')\Big]\Big)\Big\rangle_{u_\Delta} \\
&= \int \prod_t \frac{d\Delta(t)d\hat{\Delta}(t)}{2\pi} \Delta(t)\Big\langle \frac{\partial}{\partial u_\Delta(t')} \exp\Big(i\int dt\hat{\Delta}(t)\Big[\Delta(t) - u_\Delta(t) - \frac{1}{\nu_2}\int dt' R_{v_0,\hat{v}_1}(t,t')\Delta(t')\Big]\Big)\Big\rangle_{u_\Delta} \\
&= \int dt'' \Big\langle \Delta(t) [C_{v_0,v_0}]^{-1}(t',t'')u_\Delta(t'')\Big\rangle_{u_\Delta} \\
&= \Big\langle \frac{\partial\Delta(t)}{\partial u_\Delta(t')}\Big\rangle_{u_\Delta}
\end{aligned}
\tag{81}
$$

which holds via integration by parts and Stein's lemma. The same can be said for $R_{v_0,\hat{v}_1}(t,t')$

$$
R_{v_0,\hat{v}_1}(t,t') = \Big\langle \frac{\partial v_0(t)}{\partial u_1(t')}\Big\rangle_{u_1}. \tag{82}
$$

### C.1.5 LIMITING TIME DYNAMICS

We can recognize that the response functions in the above system will have time-translation invariant structure so that $R(t,t') = R(t-t')$. We can therefore take a Fourier transform of these equations, which gives

$$
R_{v_0,\hat{v}_1}(\omega) = -\frac{1}{i\omega + R_{\Delta,\hat{\Delta}}(\omega)} \tag{83}
$$

$$
R_{\Delta,\hat{\Delta}}(\omega) = \Big(1 - \nu_2^{-1}R_{v_0,\hat{v}_1}(\omega)\Big)^{-1} \tag{84}
$$

being $\nu_2 = \frac{P_2}{D}$. The same for the random variables which define the DMFT equations

$$
i\omega v_0(\omega) = -u_1(\omega) - R_{\Delta,\hat{\Delta}}(\omega)v_0(\omega) - \chi C_{sv}(\omega)\beta_s + \beta_t \tag{85}
$$

$$
v_0(\omega) = \frac{1}{i\omega + R_{\Delta,\hat{\Delta}}(\omega)}\Big[\beta_t - \chi C_{sv}(\omega)\beta_s - u_1(\omega)\Big]. \tag{86}
$$

For compactness, we introduce a shorthand $\mathcal{H}(\omega) = \frac{1}{i\omega + R_\Delta(\omega)}$. Similarly for $\Delta(\omega)$ we have

$$
\Delta(\omega) = R_{\Delta,\hat{\Delta}}(\omega)u_\Delta(\omega). \tag{87}
$$

The **loss** is governed by the two-frequency correlation function $C_{v_0,v_0}(\omega,\omega') \equiv \Big\langle v_0(\omega)v_0(\omega')\Big\rangle$. By calling $\frac{1}{D}\boldsymbol{\beta}_s \cdot \boldsymbol{\beta}_t = \alpha$ the alignment between source and target task, $C_{v_0,v_0}(\omega,\omega')$ can be derived as being

$$
C_{v_0,v_0}(\omega,\omega') = \mathcal{H}(\omega)\mathcal{H}(\omega')\Big[1 + \chi^2 C_{sv}(\omega)C_{sv}(\omega') - \alpha\chi\Big(C_{sv}(\omega) + C_{sv}(\omega')\Big) + \frac{1}{\nu_2}R_\Delta(\omega)R_\Delta(\omega')C_{0,0}(\omega,\omega')\Big]. \tag{88}
$$

By collecting $C_{0,0}(\omega,\omega')$, we get

$$
C_{v_0,v_0}(\omega,\omega') = \frac{\mathcal{H}(\omega)\mathcal{H}(\omega')}{1 - \nu_2^{-1}R_\Delta(\omega)R_\Delta(\omega')\mathcal{H}(\omega)\mathcal{H}(\omega')}\Big[1 + \chi^2 C_{sv}(\omega)C_{sv}(\omega') - \alpha\chi\Big(C_{sv}(\omega) + C_{sv}(\omega')\Big)\Big]. \tag{89}
$$

It is important to notice that, as soon as we send $\gamma_0 \to 0$, which is the feature strength on source task $\mathcal{T}_1$, then $\chi \to 0$ and we recover the test loss

$$
C_{v_0,v_0}(\omega,\omega') = \frac{\mathcal{H}(\omega)\mathcal{H}(\omega')}{1 - \nu_2^{-1}R_\Delta(\omega)R_\Delta(\omega')\mathcal{H}(\omega)\mathcal{H}(\omega')} \tag{90}
$$

which is the one we would expect in absence of any dependency on the source vector $\boldsymbol{\beta}_s$, meaning without any pretraining on $\mathcal{T}_1$. So, the interesting setting is the one for which $\chi > 0$ for a given alignment value $\alpha$. In particular, we would like to study the sign of the term in the brackets $[\cdot]$ of Eq. 89 when $t \to \infty$ or, equivalently, when $\omega, \omega' \to 0$.

First, we can compute what the correlation $C_{sv}(\omega)$ is

$$C_{sv}(\omega) = \langle v_1(\omega)\beta_s \rangle = \langle u_1(\omega)\beta_s \rangle + R_\Delta(\omega) \langle v_0(\omega)\beta_s \rangle = R_\Delta(\omega)\mathcal{H}(\omega) \left[ \alpha - \chi C_v(\omega) \right] \quad (91)$$

$$= \frac{\alpha R_\Delta \mathcal{H}}{1 + \chi R_\Delta \mathcal{H}}. \quad (92)$$

Now to get the final result, we take the $\omega, \omega' \to 0$ limit of the loss $C_{v_0,v_0}(\omega, \omega')$

$$\lim_{t,t' \to \infty} C_{v_0,v_0}(t,t') = \lim_{\omega,\omega' \to 0} (i\omega)(i\omega') C_{v_0,v_0}(\omega, \omega'). \quad (93)$$

Using the equation

$$R_\Delta = 1 - \frac{1}{\nu_2} R_\Delta \mathcal{H} \implies \lim_{\omega \to 0} R_\Delta \mathcal{H} = \nu_2 \quad (94)$$

and by noticing that

$$\lim_{\omega \to 0} i\omega \mathcal{H}(\omega) = \lim_{\omega \to 0} \frac{i\omega}{i\omega + \nu_2/(i\omega \mathcal{H})} = 1 - \nu_2 \quad (95)$$

we can combine all the results to get the loss at convergence

$$\lim_{t \to \infty} C_{v_0,v_0}(t,t) = \frac{(i\omega \mathcal{H})(i\omega \mathcal{H})}{1 - \nu_2^{-1} R\mathcal{H}R\mathcal{H}} \left[ 1 - 2\alpha \chi C_{sv} + \chi^2 C_{sv}^2 \right] \quad (96)$$

$$= (1 - \nu_2) \times \left[ 1 - \frac{2\chi \alpha^2 \nu_2}{1 + \chi \nu_2} + \frac{\chi^2 \alpha^2 \nu_2^2}{(1 + \chi \nu_2)^2} \right]. \quad (97)$$

Some key observations about this result:

- The loss only depends on $\alpha^2$ rather than $\alpha$ directly. This reflects the symmetry of the problem $\boldsymbol{\beta}_s \to -\boldsymbol{\beta}_s$.

- The loss is always lower than the original loss for any feature learning strength $\chi > 0$, since

$$\mathcal{L} \leq (1 - \nu_2) \left[ 1 - \frac{\chi \nu_2 \alpha}{1 + \chi \nu_2} \right]^2 \leq (1 - \nu_2) \quad (98)$$

which means that transfer learning has a positive effect in this setting, as soon as feature learning happens on $\mathcal{T}_1$. This is because during pre-training we minimized population risk by allowing $P_1 \to \infty$ on $\mathcal{T}_1$. As a consequence, the NTK kernel is a rank-one spiked kernel in the source direction $\boldsymbol{\beta}_s \boldsymbol{\beta}_s^\top$; there are no spurious noise spikes, and as soon as $\alpha > 0$ (nonzero source–target alignment), transfer learning cannot hurt.

- When $\alpha = 0$, meaning the target vector of the downstream task $\boldsymbol{\beta}_t$ lies in the orthogonal space w.r.t. $\boldsymbol{\beta}_s$, we recover the usual $\mathcal{L} = 1 - \nu_2$ learning curve for linear probes (Hastie et al., 2022). This happens also when $\chi = 0$, meaning if we choose a lazy pretraining on $\mathcal{T}_1$. In that case, indeed, the NTK at initialization would have just the bulk structure with no spike aligned with the source.

- If $\chi \to \infty$, which happens if the feature learning strength on the pretraining $\gamma_0 \to \infty$, then

$$\mathcal{L} = (1 - \nu_2)(1 - \alpha^2). \quad (99)$$

## C.2 Finite data on $\mathcal{T}_1$

In the proportional limit, i.e. when $\nu_1 = \frac{P_1}{D}$ is fixed, the pretraining on $\mathcal{T}_1$ learns a noisy version of the source vector $\boldsymbol{\beta}_s$ due to finite sample size fluctuations, and modulated by the feature learning strength $\gamma_0$ on $\mathcal{T}_1$. As a consequence, we expect an interplay between signal and noise components on the benefits of transfer learning on $\mathcal{T}_2$.

First, let's recall the network definition, which is

$$f(\boldsymbol{x}) = \frac{\sqrt{D}}{N\gamma_0}\boldsymbol{a}^\top\left(\frac{1}{\sqrt{D}}\boldsymbol{W}\right)\boldsymbol{x}. \tag{100}$$

This means that GD dynamics $\boldsymbol{\theta}_{t+1} = \boldsymbol{\theta}_t - \eta\gamma^2\nabla_{\boldsymbol{\theta}_t}\mathcal{L}$ for the parameters collection $\boldsymbol{\theta} = \text{Vec}\{\boldsymbol{W}, \boldsymbol{a}\}$ and on a loss function $\mathcal{L} = \frac{1}{2P_1}\sum_{\mu=1}^{P_1}(y_\mu - f_\mu)^2$ can be written layer-wise as

$$\boldsymbol{W}(t) = \boldsymbol{W}(0) + \frac{\eta\gamma_0}{\sqrt{D}}\sum_{t'<t}\boldsymbol{a}(t')\boldsymbol{h}(t')^\top \tag{101}$$

$$\boldsymbol{a}(t) = \boldsymbol{a}(0) + \frac{\eta\gamma_0}{\sqrt{D}}\sum_{t'<t}\boldsymbol{W}(t')\boldsymbol{h}(t') \tag{102}$$

having defined the fields

$$\boldsymbol{\Delta}(t) = \frac{1}{\sqrt{D}}\boldsymbol{X}\boldsymbol{v}(t) \in \mathbb{R}^{P_1} \tag{103}$$

$$\boldsymbol{v}(t) = \boldsymbol{\beta}_s - \frac{\sqrt{D}}{N\gamma_0}\boldsymbol{W}(t)^\top\boldsymbol{a}(t) = \boldsymbol{\beta}_s - \boldsymbol{\xi}(t) - \eta\sum_{s<t}C_a(t,s)\boldsymbol{h}(s) \tag{104}$$

$$\boldsymbol{h}(t) = \frac{\sqrt{D}}{P}\boldsymbol{X}^\top\boldsymbol{\Delta}(t) \in \mathbb{R}^D. \tag{105}$$

As a consequence, the feature matrix $\boldsymbol{H}(t) \in \mathbb{R}^{N\times P_1}$ is

$$\boldsymbol{H}(t) = \left(\boldsymbol{W}(0) + \frac{\eta\gamma_0}{\sqrt{D}}\sum_{t'<t}\boldsymbol{a}(t')\boldsymbol{h}(t')^\top\right)\boldsymbol{X}^\top \tag{106}$$

hence, the kernel

$$\boldsymbol{K}(t) = \frac{1}{N}\boldsymbol{H}(t)^\top\boldsymbol{H}(t)$$
$$= \boldsymbol{X}\left[\frac{\boldsymbol{W}^\top(0)\boldsymbol{W}(0)}{N} + \frac{\eta\gamma_0^2}{D}\sum_{s<t}\left(\boldsymbol{\xi}(s)\boldsymbol{h}(s)^\top + \boldsymbol{h}(s)\boldsymbol{\xi}(s)^\top\right) + \frac{\eta^2\gamma_0^2}{D}\sum_{s,s'<t}C_a(s,s')\boldsymbol{h}(s)\boldsymbol{h}(s')^\top\right]\boldsymbol{X}^\top \tag{107}$$

with

$$\boldsymbol{\xi}(s) = \frac{\sqrt{D}}{N\gamma_0}\boldsymbol{W}^\top(0)\boldsymbol{a}(s) \tag{108}$$

$$C_a(s,s') = \frac{1}{N}\boldsymbol{a}(s)^\top\boldsymbol{a}(s'). \tag{109}$$

If we proceed by substitution, we get

$$\boldsymbol{\xi}(t) = \frac{\sqrt{D}}{N\gamma_0}\boldsymbol{W}^\top(0)\boldsymbol{a}(0) + \frac{\eta}{N}\boldsymbol{W}^\top(0)\sum_{s<t}\boldsymbol{W}(s)\boldsymbol{h}(s)$$
$$= \frac{\sqrt{D}}{N\gamma_0}\boldsymbol{W}^\top(0)\boldsymbol{a}(0) + \frac{\eta}{N}\boldsymbol{W}^\top(0)\boldsymbol{W}(0)\sum_{s<t}\boldsymbol{h}(s) + \eta^2\gamma_0^2\frac{\sqrt{D}}{N}\boldsymbol{W}^\top(0)\sum_{s<t}\sum_{s'<s}\boldsymbol{a}(s')\frac{\boldsymbol{h}(s')^\top\boldsymbol{h}(s)}{D}$$
$$= \eta\sum_{s<t}\boldsymbol{h}(s) + \eta^2\gamma_0^2\sum_{s<t}\sum_{s'<s}C_h(s,s')\boldsymbol{\xi}(s') \tag{110}$$

where we realized that $\frac{\sqrt{D}}{N\gamma_0}\boldsymbol{W}^\top(0)\boldsymbol{a}(0) = \mathcal{O}(\sqrt{\frac{D}{N}})$ vanishes if we send $N \to \infty$ at fixed $D$, since $\boldsymbol{W}(0)$ and $\boldsymbol{a}(0)$ are uncorrelated at initialization, and that $\frac{1}{N}\boldsymbol{W}^\top(0)\boldsymbol{W}(0) \to \boldsymbol{I}_D$ for the same reason. Plus, we know that the correlations $C_h(s,s') = \frac{\boldsymbol{h}(s)^\top\boldsymbol{h}(s')}{D}$ concentrates in the limit $D \to \infty$; the same holds for $C_a(s,s') = \frac{1}{N}\boldsymbol{a}^\top(s)\boldsymbol{a}(s')$ in the $N \to \infty$ limit.

Now, we can collect the time indices as rows of matrix variables, for instance $\boldsymbol{\xi} \in \mathbb{R}^{T \times D}$ and solve for $\boldsymbol{\xi}$, thus getting

$$\boldsymbol{\xi} = \underbrace{\left(\boldsymbol{I} - \eta^2\gamma_0^2\boldsymbol{\Theta}C_h^{\downarrow}\right)^{-1}}_{\in\mathbb{R}^{T\times T}}\eta\boldsymbol{\Theta}\boldsymbol{h} \tag{111}$$

being $C_h^{\downarrow}(s,s') = C_h(s,s')\Theta(s-s')$ the lower-triangular matrix and $(\boldsymbol{\Theta})_{t,s} = \mathbf{1}(t > s)$. In the same way, for the $\boldsymbol{h}(t) \in \mathbb{R}^D$ field, which we can get from a short path integral derivation similarly to what we have done above (see (Bordelon & Pehlevan, 2025)), we have

$$\begin{aligned}\boldsymbol{h}(t) &= \boldsymbol{u}(t) + \sum_{s<t} R_\Delta(t,s)\boldsymbol{v}(s) \\ &= \boldsymbol{u}(t) + \sum_{s<t} R_\Delta(t,s)\left(\boldsymbol{\beta}_s - \boldsymbol{\xi}(s) - \eta\sum_{s'<s} C_a(s,s')\boldsymbol{h}(s')\right)\end{aligned} \tag{112}$$

with $u(t) \sim \mathcal{GP}(0, \frac{1}{\nu_1}C_\Delta)$ and $\nu_1 = \frac{P_1}{D}$. Again, by collecting the time indices we can solve for $\boldsymbol{h} \in \mathbb{R}^{T \times D}$

$$\boldsymbol{h} = \underbrace{\left(\boldsymbol{I} + \eta\boldsymbol{R}_\Delta^{\downarrow}\left(\boldsymbol{I} - \eta^2\gamma_0^2\boldsymbol{\Theta}C_h^{\downarrow}\right)^{-1}\boldsymbol{\Theta} + \eta\boldsymbol{R}_\Delta^{\downarrow}C_a^{\downarrow}\right)^{-1}}_{\in\mathbb{R}^{T\times T}}\left[\boldsymbol{u} + \boldsymbol{R}_\Delta^{\downarrow}\mathbf{1}\boldsymbol{\beta}_s^{\top}\right] \tag{113}$$

having defined

$$R_\Delta^{\downarrow}(t,s) = \Theta(t-s)R_\Delta(t,s) \tag{114}$$

$$C_a^{\downarrow}(s,s') = C_a(s,s')\Theta(s-s'). \tag{115}$$

By staring at Eqs. 111, 113 we realize that, since time operators do not create new spatial direction, both $\{\boldsymbol{\xi}(t), \boldsymbol{h}(t)\} \in \mathbb{R}^D$ fields can only grow in either the source direction $\boldsymbol{\beta}_s$ or in the uncorrelated noise direction $\boldsymbol{u}(t)$, which comes from finite sample fluctuations of $\boldsymbol{X}$. Consequently, $\{\boldsymbol{\xi}(t), \boldsymbol{h}(t)\}$ admit the causal decomposition

$$\boldsymbol{h}(t) = c(t)\boldsymbol{\beta}_s + \sum_{s<t} R_{hu}(t,s)\boldsymbol{u}(s) \tag{116}$$

$$\boldsymbol{\xi}(t) = d(t)\boldsymbol{\beta}_s + \sum_{s<t} R_{\xi u}(t,s)\boldsymbol{u}(s) \tag{117}$$

where we replaced time-dependent scalars $\{c(t), d(t)\}$, which are functions of $\{\eta, \gamma_0, \nu_1\}$. These represent the projection of the fields along the fixed teacher direction $\boldsymbol{\beta}_s$, while the $\{R_{hu}, R_{\xi u}\}$ are the usual casual-time response functions which map the drive $\boldsymbol{u}(\cdot)$ to the features $\boldsymbol{h}(\cdot)$ and $\boldsymbol{\xi}(\cdot)$. Precisely

$$\boldsymbol{R}_{hu} = \left(\boldsymbol{I} + \eta\boldsymbol{R}_\Delta^{\downarrow}\left(\boldsymbol{I} - \eta^2\gamma_0^2\boldsymbol{\Theta}C_h^{\downarrow}\right)^{-1}\boldsymbol{\Theta} + \eta\boldsymbol{R}_\Delta^{\downarrow}C_a^{\downarrow}\right)^{-1} \tag{118}$$

$$\boldsymbol{R}_{\xi u} = \eta\left(\boldsymbol{I} - \eta^2\gamma_0^2\boldsymbol{\Theta}C_h^{\downarrow}\right)^{-1}\boldsymbol{\Theta}\boldsymbol{R}_{hu}. \tag{119}$$

In general, deriving the limiting time of the fields $\{\boldsymbol{h}(t), \boldsymbol{\xi}(t)\}$ requires to study the $t \to \infty$ limit of correlation and response functions as they appear in Eqs. 111, 113, which is in principle hard. Because of that, in the following derivation we will assume the casual decomposition as in Eqs. 116, 117, and recover the feature kernel from that.

### C.2.1 ANSATZ ON THE KERNEL STRUCTURE

Given the above discussion, and going back to the kernel expression as in Eq. 107, we can now assume the kernel at convergence ($t \to \infty$) having the functional form

$$\boldsymbol{K}(\boldsymbol{X}, \boldsymbol{X}) = \boldsymbol{X}\underbrace{\left[\boldsymbol{I} + \frac{c_1}{D}\left(\boldsymbol{g}\boldsymbol{\beta}_s^{\top} + \boldsymbol{\beta}_s\boldsymbol{g}^{\top}\right) + \frac{c_2}{D}\boldsymbol{\beta}_s\boldsymbol{\beta}_s^{\top} + \frac{c_3}{D}\boldsymbol{g}\boldsymbol{g}^{\top}\right]}_{M}\boldsymbol{X}^{\top} \tag{120}$$

where $\boldsymbol{g} \in \mathbb{R}^D$ is a Gaussian vector $\boldsymbol{g} \perp \boldsymbol{\beta}_s$ such that $\text{Cov}(\boldsymbol{g}) = \frac{1}{\nu_1} C_\Delta^\infty$, and with $C_\Delta^\infty = \lim_{t \to \infty} \frac{1}{P_1} \boldsymbol{\Delta}(t) \cdot \boldsymbol{\Delta}(t')$ which concentrates as $P_1 \to \infty$. As $\nu_1 \to \infty$, we expect $C_\Delta^\infty \to 0$. Instead, $\{c_1, c_2, c_3\}$ are constants which are functions of $\{\eta, \gamma_0, \nu_1\}$.

Notice that, differently from before, the kernel depends now on the noise direction $\boldsymbol{g}$ tuned by the constants $\{c_1, c_3\}$. We do not expect, in general, transfer learning to have a positive effect as soon as the niose component $c_3$ grow large compared to the signal spike tuned by $c_2$.

Again, we do gradient flow with this final NTK and a loss function $\mathcal{L}(t) = \frac{1}{2P_2} |\boldsymbol{X}_t^\top \boldsymbol{M}^{1/2} \hat{\boldsymbol{\beta}}(t) - \boldsymbol{X}_t^\top \boldsymbol{\beta}_t|^2$ and $\boldsymbol{X} \in \mathbb{R}^{D \times P_2}$

$$\frac{d}{dt}\hat{\boldsymbol{\beta}}(t) = \boldsymbol{M}^{1/2} \frac{\boldsymbol{X}\boldsymbol{X}^\top}{P_2} \Big( \boldsymbol{\beta}_t - \boldsymbol{M}^{1/2} \hat{\boldsymbol{\beta}}(t) \Big) \tag{121}$$

from which, by defining $\boldsymbol{v}_0 = \boldsymbol{\beta}_t - \boldsymbol{M}^{1/2} \hat{\boldsymbol{\beta}}(t)$ as usual, we get

$$\frac{d}{dt}\boldsymbol{v_0} = -\Big( \boldsymbol{I} + \frac{c_1}{D}\Big( \boldsymbol{g}\boldsymbol{\beta}_s^\top + \boldsymbol{\beta}_s\boldsymbol{g}^\top \Big) + \frac{c_2}{D}\boldsymbol{\beta}_s\boldsymbol{\beta}_s^\top + \frac{c_3}{D}\boldsymbol{g}\boldsymbol{g}^\top \Big) \frac{\boldsymbol{X}\boldsymbol{X}^\top}{P_2}\boldsymbol{v_0} + \delta(t)\boldsymbol{\beta}_t. \tag{122}$$

We can introduce the following fields

$$\boldsymbol{\Delta} = \frac{1}{\sqrt{D}}\boldsymbol{X}^\top \boldsymbol{v}_0 \in \mathbb{R}^{P_2} \tag{123}$$

$$\boldsymbol{v}_1 = \frac{\sqrt{D}}{P_2}\boldsymbol{X}\boldsymbol{\Delta} \in \mathbb{R}^D \tag{124}$$

$$C_{sv} = \frac{1}{D}\boldsymbol{\beta}_s \cdot \boldsymbol{v}_1 \tag{125}$$

$$C_{gv} = \frac{1}{D}\boldsymbol{g} \cdot \boldsymbol{v}_1 \tag{126}$$

and getting the dynamics

$$\frac{d}{dt}\boldsymbol{v}_0 = -\boldsymbol{v}_1(t) - \Big( c_1\boldsymbol{g} + c_2\boldsymbol{\beta}_s \Big) C_{sv}(t) - \Big( c_1\boldsymbol{\beta}_s + c_3\boldsymbol{g} \Big) C_{gv}(t) + \delta(t)\boldsymbol{\beta}_t. \tag{127}$$

By enforcing the fields definitions, we can do a path integral derivation similar to the one in Sec. C.1, and so by averaging over the $\mathcal{T}_2$ dataset with $\nu_2 = \frac{P_2}{D}$ fixed, we get the usual MGF of DMFT $\mathcal{Z} = \int d\boldsymbol{q} \exp\Big( -D\mathcal{S}(\boldsymbol{q}) \Big)$ with $\boldsymbol{q}$ being the collection of correlation and response functions while $\mathcal{S}$ being the DMFT action.

### C.2.2 DMFT ACTION

In this setting, the action takes the form

$$\mathcal{S} = -\frac{1}{2}\int dt C_{sv}(t)\hat{C}_{sv}(t) - \frac{1}{2}\int dt C_{gv}(t)\hat{C}_{gv}(t) - \frac{1}{2}\int dtdt' C_{v_0,v_0}(t,t')\hat{C}_{v_0,v_0}(t,t')$$
$$- \frac{\nu_2}{2}\int dtdt' C_{\Delta,\Delta}(t,t')\hat{C}_{\Delta,\Delta}(t,t') + \int dtdt' R_{\Delta,\hat{\Delta}}(t,t')R_{v_0,\hat{v}_1}(t,t')$$
$$- \frac{1}{D}\sum_{i=1}^D \ln \mathcal{Z}_{01}\Big[ C_{sv}, C_{gv}, C_{\Delta,\Delta}, \hat{C}_{sv}, \hat{C}_{gv}, \hat{C}_{v_0,v_0}, R_{\Delta,\hat{\Delta}} \Big] - \frac{1}{D}\sum_{j=1}^{\nu_2 D} \ln \mathcal{Z}_\Delta\Big[ C_{v_0,v_0}, R_{v_0,\hat{v}_1}, \hat{C}_{\Delta,\Delta} \Big].$$
$$\tag{128}$$

with single site functions

$$
\mathcal{Z}_{01} = \int \frac{dv_0 d\hat{v}_0}{2\pi} \int \frac{dv_1 d\hat{v}_1}{2\pi} \exp\left[ -\frac{1}{2\nu} \int dt dt' C_{\Delta,\Delta} \hat{v}_1(t) \hat{v}_1(t') - \frac{1}{2} \int dt \hat{C}_{sv}(t) \beta_s v_1(t) \right]
$$

$$
\times \exp\left[ -\frac{1}{2} \int dt \hat{C}_{gv}(t) g v_1(t) - \frac{1}{2} \int dt dt' \hat{C}_{v_0,v_0} v_0(t) v_0(t') - i \int dt dt' R_{\Delta,\hat{\Delta}} v_0(t) \hat{v}_1(t') \right]
$$

$$
\times \exp\left[ i \int dt \hat{v}_0 \left( \partial_t v_0 + v_1 + \left( \frac{c_1}{\sqrt{\nu_1}} g + \frac{c_2}{\sqrt{\nu_1}} \beta_s \right) C_{sv}(t) + \left( \frac{c_1}{\sqrt{\nu_1}} \beta_s + \frac{c_3}{\nu_1} g \right) C_{gv}(t) - \delta(t) \beta_t \right) \right]
$$

$$
\times \exp\left[ i \int dt v_1(t) \hat{v}_1(t) \right]
$$

$$(129)$$

and

$$
\mathcal{Z}_{\Delta} = \int \frac{d\Delta d\hat{\Delta}}{2\pi} \exp\left[ -\frac{1}{2} \int dt dt' C_{v_0,v_0}(t,t') \hat{\Delta}(t) \hat{\Delta}(t') - \frac{1}{2} \int dt dt' \hat{C}_{\Delta,\Delta}(t,t') \Delta(t) \Delta(t') \right]
$$

$$
\times \exp\left[ -i\nu^{-1} \int dt dt' R_{v_0,\hat{v}_1}(t,t') \Delta(t) \hat{\Delta}(t') + i \int dt \Delta(t) \hat{\Delta}(t) \right].
$$

$$(130)$$

Again, in the $D \to \infty$ limit, the saddle point equations which make $\mathcal{S}$ locally stationary give

$$
-\frac{1}{2} C_{sv}(t) + \frac{1}{2D} \sum_{i=1}^{D} \left\langle \beta_s v_1(t) \right\rangle_i = 0 \tag{131}
$$

$$
-\frac{1}{2} C_{gv}(t) + \frac{1}{2D} \sum_{i=1}^{D} \left\langle g v_1(t) \right\rangle_i = 0 \tag{132}
$$

$$
-\frac{1}{2} C_{v_0,v_0}(t,t') + \frac{1}{2D} \sum_{i=1}^{D} \left\langle v_0(t) v_0(t') \right\rangle_i = 0 \tag{133}
$$

$$
-\frac{\nu}{2} C_{\Delta,\Delta}(t,t') + \frac{1}{2P_2} \sum_{j=1}^{P_2} \left\langle \Delta(t) \Delta(t') \right\rangle_j = 0 \tag{134}
$$

and the same for the response functions

$$
R_{\Delta,\hat{\Delta}} + \frac{i}{P_2} \sum_{j=1}^{P_2} \left\langle \Delta(t) \hat{\Delta}(t') \right\rangle_j = 0 \tag{135}
$$

$$
R_{v_0,\hat{v}_1} + \frac{i}{D} \sum_{i=1}^{D} \left\langle v_0(t) \hat{v}_1(t') \right\rangle_i = 0 \tag{136}
$$

being the averages $\langle \cdot \rangle_i$, $\langle \cdot \rangle_j$ over the single site distributions $\mathcal{Z}_{01}$ and $\mathcal{Z}_{\Delta}$ (factorized over $i \in \{D\}$ and $j \in \{P_2\}$ respectively). At the same time, as usual, the conjugated fields vanish

$$
\hat{C}_{sv}(t) = \hat{C}_{v_0,v_0} = \hat{C}_{\Delta,\Delta}(t,t') = 0. \tag{137}
$$

Since in the $P_2, D \to \infty$ limit with $\nu_2 = \frac{P_2}{D}$ fixed all the correlation and response functions concentrate, we can use Hubbard-Stratonovich transformations to linearize the quadratic terms in

$\mathcal{Z}_{01}$ and $\mathcal{Z}_{\Delta}$ by introducing some Gaussian fields

$$\exp\left(-\frac{1}{2\nu_2}\int dtdt' C_{\Delta,\Delta}\hat{v}_1(t)\hat{v}_1(t')\right) = \left\langle \exp\left(-i\int dt\hat{v}_1(t)u_1(t)\right)\right\rangle_{u_1\sim\mathcal{N}(0,\frac{1}{\nu_2}C_{\Delta,\Delta})}$$

(138)

$$\exp\left(-\frac{1}{2}\int dtdt' C_{v_0,v_0}(t,t')\hat{\Delta}(t)\hat{\Delta}(t')\right) = \left\langle \exp\left(-i\int dt\hat{\Delta}(t)u_{\Delta}(t)\right)\right\rangle_{u_{\Delta}\sim\mathcal{N}(0,C_{v_0,v_0})}.$$

(139)

As a consequence, the DMFT equations that describe the single site stochastic processes are

$$v_1(t) = u_1(t) + \int dt' R_{\Delta,\hat{\Delta}}(t')v_0(t'), \quad u_1(t) \sim \mathcal{GP}\left(0, \frac{1}{\nu_2}C_{\Delta,\Delta}\right)$$

(140)

$$\partial_t v_0 = -u_1(t) - \int dt' R_{\Delta,\hat{\Delta}}(t')v_0(t') - (c_1 g + c_2\beta_s)C_{sv}(t) - (c_1\beta_s + c_3 g)C_{gv}(t) + \delta(t)\beta_t$$

(141)

$$\Delta(t) = u_{\Delta}(t) + \frac{1}{\nu_2}\int dt' R_{v_0,\hat{v}_1}\Delta(t'), \quad u_{\Delta}(t) \sim \mathcal{GP}\left(0, C_{v_0,v_0}\right).$$

(142)

### C.2.3 Simplifying the Response Functions

As we did in Sec. C.1.4, via integration by parts and Stein's lemma we can simplify the saddle point equations for the correlation functions, which become

$$R_{v_0,\hat{v}_1} = \left\langle \frac{\partial v_0(t)}{\partial u_1(t')}\right\rangle_{u_1}$$

(143)

$$R_{\Delta,\hat{\Delta}} = \left\langle \frac{\partial \Delta(t)}{\partial u_{\Delta}(t')}\right\rangle_{u_{\Delta}}.$$

(144)

### C.2.4 Limiting Time Dynamics

We notice again that the loss can be obtained from the time-time diagonal of the correlation function $C_{v_0,v_0} = \langle v_0(t)v_0(t)\rangle$, which we would like to study at limiting time. Because of that, and by noticing that the system is time translational invariant, we can take a Fourier transform of Eq. 141, thus getting

$$i\omega v_0(\omega) = -u_1(\omega) - R_{\Delta}(\omega)v_0(\omega) - C_{sv}(\omega)\left(c_1 g + c_2\beta_s\right) - C_{gv}(\omega)\left(c_1\beta_s + c_3 g\right) + \beta_t$$

$$\Rightarrow v_0(\omega) = \frac{1}{i\omega + R_{\Delta}(\omega)}\left[\beta_t - u_1(\omega) - C_{sv}(\omega)\left(c_1 g + c_2\beta_s\right) - C_{gv}(\omega)\left(c_1\beta_s + c_3 g\right)\right].$$

(145)

where we call $\mathcal{H}(\omega) = \frac{1}{i\omega + R_{\Delta}(\omega)}$ as before. The same can be done for $\Delta(\omega)$

$$\Delta(\omega) = R_{\Delta}(\omega)u_{\Delta}(\omega)$$

(146)

and for both the correlations of $v_1$ with the signal $\beta_s$ and the noise $g$ directions of $\mathcal{T}_1$, once we define the alignments

$$\alpha_s = \frac{1}{D}\beta_t \cdot \beta_s$$

(147)

$$\alpha_g = \frac{1}{D}\beta_t \cdot g.$$

(148)

Recalling their definitions, we get

$$C_{gv}(\omega) = \left\langle gv_1(\omega)\right\rangle = gR_{\Delta}(\omega)\left\langle v_0(\omega)\right\rangle$$

$$= R_{\Delta}(\omega)\mathcal{H}(\omega)\left[\alpha_g - c_1 C_{sv}(\omega) - c_3 C_{gv}(\omega)\right]$$

$$= \frac{R_{\Delta}\mathcal{H}}{\left[1 + c_3 R_{\Delta}\mathcal{H}\right]}\left[\alpha_g - c_1 C_{sv}(\omega)\right]$$

(149)

and

$$
\begin{aligned}
C_{sv}(\omega) &= \Big\langle \beta_s v_1(\omega) \Big\rangle = \beta_s R_\Delta(\omega) \Big\langle v_0(\omega) \Big\rangle \\
&= R_\Delta(\omega)\mathcal{H}(\omega)\Big[\alpha_s - c_2 C_{sv}(\omega) - c_1 C_{gv}(\omega)\Big] \\
&= \frac{R_\Delta\mathcal{H}\Big[\left(1 + c_3 R_\Delta\mathcal{H}\right)\alpha_s - c_1 R_\Delta\mathcal{H}\alpha_g\Big]}{\left(1 + c_2 R_\Delta\mathcal{H}\right)\left(1 + c_3 R_\Delta\mathcal{H}\right) - c_1^2 R_\Delta^2\mathcal{H}^2}
\end{aligned}
\tag{150}
$$

which implies

$$
\begin{aligned}
C_{v_0,v_0}(\omega,\omega') &\equiv \Big\langle v_0(\omega) v_0(\omega') \Big\rangle \\
&= \frac{\mathcal{H}(\omega)\mathcal{H}(\omega')}{1 - \nu_2^{-1} R_\Delta(\omega) R_\Delta(\omega')\mathcal{H}(\omega)\mathcal{H}(\omega')}\bigg[1 - \Big(c_1\alpha_g + c_2\alpha_s\Big)\Big(C_{sv}(\omega) + C_{sv}(\omega')\Big) \\
&\quad - \Big(c_1\alpha_s + c_3\alpha_g\Big)\Big(C_{gv}(\omega) + C_{gv}(\omega')\Big) + (c_1^2 + c_2^2)C_{sv}(\omega)C_{sv}(\omega') \\
&\quad + \Big(c_1 c_3 + c_1 c_2\Big)\Big(C_{sv}(\omega)C_{gv}(\omega') + C_{sv}(\omega')C_{gv}(\omega)\Big) \\
&\quad + (c_1^2 + c_3^2)C_{gv}(\omega)C_{gv}(\omega')\bigg].
\end{aligned}
\tag{151}
$$

Now to get the final result, we take the $\omega, \omega' \to 0$ limits. Using the equation

$$
R_\Delta = 1 - \frac{1}{\nu_2} R_\Delta\mathcal{H} \Rightarrow \lim_{\omega\to 0} R_\Delta\mathcal{H} = \nu_2
\tag{152}
$$

which implies also

$$
\lim_{\omega\to 0}(i\omega)\mathcal{H} = 1 - \nu_2
\tag{153}
$$

we can derive the limiting time of correlation functions

$$
C_{gv}(0) = \frac{\nu_2}{1 + c_3\nu_2}\Big[\alpha_g - c_1 C_{sv}(0)\Big]
\tag{154}
$$

$$
C_{sv}(0) = \frac{\nu_2\left[\left(1 + c_3\nu_2\right)\alpha_s - c_1\nu_2\alpha_g\right]}{\left(1 + c_2\nu_2\right)\left(1 + c_3\nu_2\right) - c_1^2\nu_2^2}
\tag{155}
$$

Because of the dependency of many variables, let's study the loss in the special case where $\alpha_s = 1$ and $\alpha_g = 0$. In this case, one obtains the following loss

$$
\mathcal{L} = (1 - \nu_2)\frac{(1 + c_3\nu_2)^2 + c_1^2\nu_2^2}{D^2}
\tag{156}
$$

with

$$
D = \left(1 + c_2\nu_2\right)\left(1 + c_3\nu_2\right) - c_1^2\nu_2^2
\tag{157}
$$

$$
C_{sv}(0) = \frac{\nu_2\left(1 + c_3\nu_2\right)}{\left(1 + c_2\nu_2\right)\left(1 + c_3\nu_2\right) - c_1^2\nu_2^2}
\tag{158}
$$

$$
C_{gv}(0) = -\frac{c_1\nu_2^2}{\left(1 + c_2\nu_2\right)\left(1 + c_3\nu_2\right) - c_1^2\nu_2^2}.
\tag{159}
$$

It is now interesting to distinguish between some limiting cases in the overparameterized setting where $\nu_2 \in [0,1]$. First of all, for the kernel to be PSD it is sufficient to restrict to the span $\{\boldsymbol{\beta}_s, \boldsymbol{g}\}$, from which we get the conditions

$$
(1 + c_2)(1 + c_3) \geq c_1^2; \quad 1 + c_2 \geq 0; \quad 1 + c_3 \geq 0.
\tag{160}
$$

- Baseline ($c1 = c_2 = c_3 = 0$): we recover

$$
\mathcal{L} = 1 - \nu_2
\tag{161}
$$

as the reference loss of a linear probe with no pretraining on $\mathcal{T}_1$.

- If the signal term $c_2 = 0$, then

$$\mathcal{L} = (1 - \nu_2) \frac{(1 + c_3 \nu_2)^2 + c_1^2 \nu_2^2}{(1 + c_3 \nu_2 - c_1^2 \nu_2^2)^2}. \tag{162}$$

  – No crosstalk ($c_1 = 0$), then

$$\mathcal{L} = 1 - \nu_2, \quad \forall c_3 \tag{163}$$

  so the noise has no effect on the baseline loss in this aligned setting ($\alpha_g = 0, \alpha_s = 1$).

  – In this setting, crosstalk proportional to $c_1$ can never actually help because of PSD conditions on the kernel, which means that $c_1 \neq 0$ has always a negative effect on transfer learning. One would need $\alpha_g \neq 0$ to get a non empty range of values for which $c_1$ can actually help.

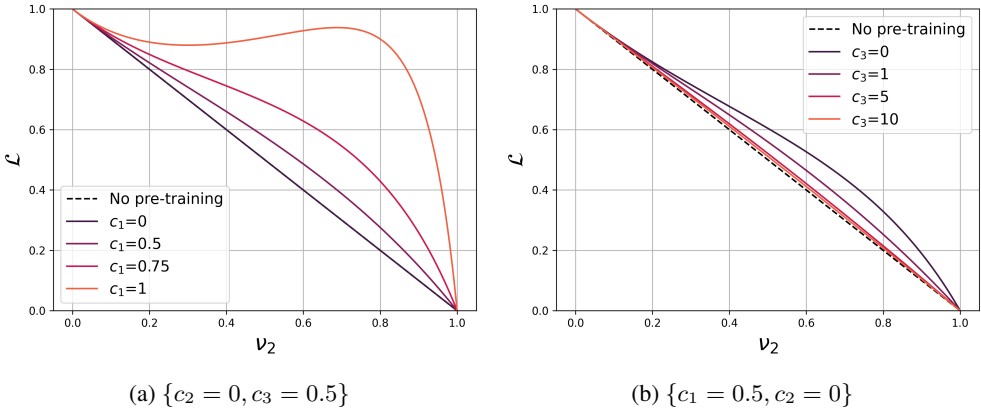

(a) $\{c_2 = 0, c_3 = 0.5\}$            (b) $\{c_1 = 0.5, c_2 = 0\}$

Figure 6: Fine-tuning from an adaptive kernel with limited data on source task ($\nu_1$ finite): loss vs downstream data $\nu_2 = P_2/D$. Dashed black: no pre-training (linear probe). In absence of signal from $\mathcal{T}_1$ (i.e., $c_2 = 0$) (a) crosstalk $c_1$ has a negative effect on transfer since $\alpha_g = 0$; (b) noise $c_3$ uncorrelated with the target acts has a regularization effect on the loss, pushing it towards the baseline $\mathcal{L} = 1 - \nu_2$.

- If the crosstalk term $c_1 = 0$, then

$$\mathcal{L} = (1 - \nu_2) \frac{1}{(1 + \nu_2 c_2)^2} \tag{164}$$

  and the loss is independent on the noise $c_3$, while the signal $c_2 > 0$ strictly helps.

- If the noise term $c_3 = 0$, then

$$\mathcal{L} = (1 - \nu_2) \frac{1 + c_1^2 \nu_2^2}{(1 + c_2 \nu_2 - c_1^2 \nu_2^2)^2} \tag{165}$$

  and the loss is a monotonically increasing function of the crosstalk term $c_1 \neq 0$.

## C.3   FEATURE LEARNING STRENGTH $\gamma_0 \to \infty$ ON $\mathcal{T}_1$

If, at initialization $\boldsymbol{W}_0, \boldsymbol{a}_0$ are small on $\mathcal{T}_1$, under gradient flow

$$\partial_t(\boldsymbol{W}\boldsymbol{W}^\top - \boldsymbol{a}\boldsymbol{a}^\top) = 0 \tag{166}$$

which, if we choose exactly $\boldsymbol{W}_0\boldsymbol{W}_0^\top = \boldsymbol{a}_0\boldsymbol{a}_0^\top$, implies that $\boldsymbol{W} = \boldsymbol{a}\boldsymbol{v}^\top$. Since $\boldsymbol{f} = \frac{1}{\sqrt{D}}\boldsymbol{X}\boldsymbol{\beta}_s$, with $\boldsymbol{X} \in \mathbb{R}^{P_1 \times D}$, then we can solve for $\boldsymbol{v} \in \mathbb{R}^D$ and studying the dynamics

$$\partial_t\boldsymbol{v}(t) = -\frac{1}{P}(\boldsymbol{X}^\top\boldsymbol{X})(\boldsymbol{v}(t) - \boldsymbol{\beta}_s) \tag{167}$$

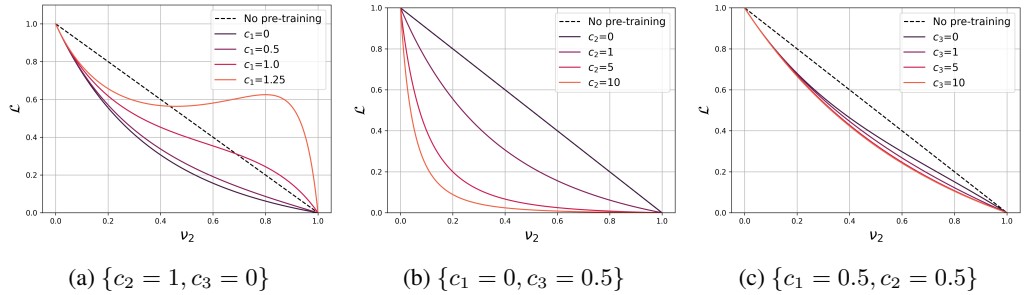

(a) $\{c_2 = 1, c_3 = 0\}$      (b) $\{c_1 = 0, c_3 = 0.5\}$      (c) $\{c_1 = 0.5, c_2 = 0.5\}$

Figure 7: Fine-tuning from an adaptive kernel with limited data on source task ($\nu_1$ finite): loss vs downstream data $\nu_2 = P_2/D$. Dashed black: no pre-training (linear probe). No crosstalk ($c_1 = 0$): (a) positive signal $c_2 > 0$ from $\mathcal{T}_1$ strictly lowers the loss compared to the baseline; (b) at fixed signal, curves collapse for any noise $c_3$, since it is uncorrelated with the target direction in this case ($\alpha_g = 0$).

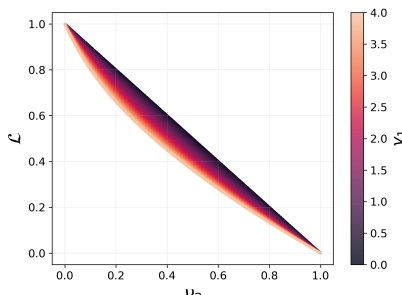

Figure 8: Linear model from Result 3 when $c_1 = \nu_1\sqrt{\nu_1(1-\nu_1)}\chi$, $c_2 = \nu_1^2\chi$, $c_3 = \nu_1(1-\nu_1)\chi$ with $\chi = \sqrt{1-\gamma_1^2} - 1$. When source and target tasks are sufficiently aligned ($\alpha_s = 0.7, \alpha_g = -0.01$), the test loss on $\mathcal{T}_2$ is monotone in $\gamma_1$.

from which the feature kernel can be derived as $M = \frac{1}{N}W^\top W = \frac{|a|^2}{N}vv^\top$. By calling $v_0(t) = \beta_s - v(t)$, we get

$$\partial_t v_0(t) = -v_1(t) \tag{168}$$

$$\Delta(t) = \frac{1}{\sqrt{D}}Xv_0(t) \in \mathbb{R}^{P_1} \tag{169}$$

$$v_1(t) = \frac{\sqrt{D}}{P}X^\top\Delta(t) \in \mathbb{R}^D. \tag{170}$$

With a short path integral (or cavity) derivation similar to what we did in previous sections, it is possible to exploit translational invariance of the model, thus getting the DMFT equations that describe the single site stochastic processes. In the current setting, those are

$$v_1(t) = u_1(t) + \int dt' R_\Delta(t,t')v_0(t'), \quad u_1(t) \sim \mathcal{GP}\left(0, \frac{1}{\nu_1}C_\Delta\right) \tag{171}$$

$$\partial_t v_0(t) = -u_1(t) - \int dt' R_\Delta(t,t')v_0(t') + \delta(t)\beta_s \tag{172}$$

$$\Delta(t) = u_\Delta(t) + \frac{1}{\nu_1}\int dt' R_{01}(t,t')\Delta(t'), \quad u_\Delta(t) \sim \mathcal{GP}\left(0, C_{0,0}\right) \tag{173}$$

where, as usual if $P_1 = \nu_1 D$, then

$$C_\Delta(t, t') = \frac{1}{P_1} \sum_{j=1}^{P_1} \left\langle \Delta(t)\Delta(t') \right\rangle_j \tag{174}$$

$$C_{0,0}(t, t') = \frac{1}{D} \sum_{i=1}^{D} \left\langle v_0(t)v_0(t') \right\rangle_i \tag{175}$$

$$R_\Delta(t, t') = \left\langle \frac{\partial \Delta(t)}{\partial u_\Delta(t')} \right\rangle_{u_\Delta} \tag{176}$$

$$R_{01}(t, t') = \left\langle \frac{\partial v_0(t)}{\partial u_1(t')} \right\rangle_{u_1} \tag{177}$$

being the averages respectively over

$$\mathcal{Z}_\Delta = \int \frac{d\Delta d\hat{\Delta}}{2\pi} \left\langle \exp\left( +i \int dt \hat{\Delta}(t) \left[ \Delta(t) - u_\Delta(t) - \frac{1}{\nu_1} \int dt' R_{01}(t, t')\Delta(t') \right] \right) \right\rangle_{u_\Delta \sim \mathcal{N}(0, C_0)} \tag{178}$$

and

$$\mathcal{Z}_{01} = \int \frac{dv_0 d\hat{v}_0}{2\pi} \int \frac{dv_1 d\hat{v}_1}{2\pi} \left\langle \exp\left[ +i \int dt \hat{v}_1(t) \left( v_1(t) - u_1(t) - \int dt' R_\Delta(t, t')v_0(t') \right) \right] \right\rangle_{u_1 \sim \mathcal{N}(0, \frac{1}{\nu_1} C_\Delta)}$$

$$\times \exp\left[ +i \int dt \, \hat{v}_0(t) \left( \partial_t v_0(t) + v_1(t) \right) \right]. \tag{179}$$

Taking a Fourier transform the DMFT equations simplify

$$v_0(\omega) = \frac{1}{i\omega + R_\Delta(\omega)} \left[ \beta_s - u_1(\omega) \right] \tag{180}$$

$$\Delta(\omega) = \frac{u_\Delta(\omega)}{1 + \frac{1}{\nu_1}\mathcal{H}(\omega)} \tag{181}$$

$$R_{01}(\omega) = -\frac{1}{i\omega + R_\Delta(\omega)} = -\mathcal{H}(\omega) \tag{182}$$

$$R_\Delta(\omega) = \frac{1}{1 + \frac{1}{\nu_1}\mathcal{H}(\omega)} \tag{183}$$

and the loss function can be written as

$$C_{0,0}(\omega, \omega') \equiv \left\langle v_0(\omega)v_0(\omega') \right\rangle$$
$$= \mathcal{H}(\omega)\mathcal{H}(\omega') \left[ 1 + \frac{1}{\nu_1} C_{0,0}(\omega, \omega') R_\Delta(\omega) R_\Delta(\omega') \right] \tag{184}$$

while the correlation

$$C_\Delta(\omega, \omega') \equiv \left\langle \Delta(\omega)\Delta(\omega') \right\rangle$$
$$= R_\Delta(\omega) R_\Delta(\omega') C_{0,0}(\omega, \omega'). \tag{185}$$

### C.3.1    LIMITING TIME DYNAMICS ON $\mathcal{T}_1$

If $\nu_1 \in [0, 1]$, then from the equation

$$R_\Delta = 1 - \frac{1}{\nu_1} \frac{R_\Delta}{i\omega + R_\Delta} \tag{186}$$

we find that, at limiting time $R_\Delta(0) = \frac{\nu_1}{1-\nu_1}$, and so $\frac{1}{\nu_1}C_\Delta = \frac{\nu_1}{(1-\nu_1)}$. From the definition $v_0(t) = \beta_s - v(t)$ we get

$$
\begin{aligned}
v &= \lim_{\omega \to 0} \beta_s - i\omega v_0(\omega) \\
&= \lim_{\omega \to 0} (1 - i\omega \mathcal{H}(\omega))\beta_s + i\omega \mathcal{H}(\omega)u_1 \\
&\sim \nu_1 \beta_s + \sqrt{\nu_1(1-\nu_1)}g
\end{aligned}
\tag{187}
$$

by defining $g \sim \mathcal{N}(0, I)$ as Gaussian vector uncorrelated with the source $\beta_s$. As a consequence, the kernel is

$$
vv^\top = \left[\nu_1 \beta_s + \sqrt{\nu_1(1-\nu_1)}g\right]\left[\nu_1 \beta_s + \sqrt{\nu_1(1-\nu_1)}g\right]^\top.
\tag{188}
$$

With this kernel, as we did above, we would now like to study a fine-tuned model with fixed pre-trained features and a linear readout that has to align with the downstream task $\mathcal{T}_2$ identified by a target vector $\beta_t \in \mathbb{R}^D$.

We call $v_0 = \beta_t - K^{1/2}\hat{\beta}(t)$ and get the dynamics

$$
\partial_t v_0 = -\left[\nu_1^2 C_{v_1\beta}(t)\beta_s + \nu_1\sqrt{\nu_1(1-\nu_1)}\left(C_{v_1g}(t)\beta_s + C_{v_1\beta}(t)g\right) + \nu_1(1-\nu_1)C_{v_1g}(t)g\right] + \delta(t)\beta_t
\tag{189}
$$

where

$$
\Delta(t) = \frac{1}{\sqrt{D}}Xv_0(t) \in \mathbb{R}^{P_2}
\tag{190}
$$

$$
v_1 = \frac{\sqrt{D}}{P_2}X\Delta \in \mathbb{R}^D
\tag{191}
$$

$$
C_{v_1\beta} = \frac{1}{D}v_1 \cdot \beta_s
\tag{192}
$$

$$
C_{v_1g} = \frac{1}{D}v_1 \cdot g
\tag{193}
$$

$$
\alpha_s = \frac{1}{D}\beta_t \cdot \beta_s
\tag{194}
$$

$$
\alpha_g = \frac{1}{D}\beta_t \cdot g.
\tag{195}
$$

As a consequence

$$
\partial_t C_{v_0\beta}(t) = -\nu_1\left[\nu_1 C_{v_1\beta}(t) + \sqrt{\nu_1(1-\nu_1)}C_{v_1g}(t)\right] + \alpha_s\delta(t)
\tag{196}
$$

$$
\partial_t C_{v_0g}(t) = -\sqrt{\nu_1(1-\nu_1)}\left[\nu_1 C_{v_1\beta}(t) + \sqrt{\nu_1(1-\nu_1)}C_{v_1g}(t)\right] + \alpha_g\delta(t).
\tag{197}
$$

At this point, by realizing through DMFT that

$$
v_1(t) = u_1(t) + \int dt' R_\Delta(t, t')v_0(t')
\tag{198}
$$

and by taking a Fourier transform of Eqs. 196, 197 we get

$$
i\omega C_{v_0\beta}(\omega) = -\nu_1\left[\nu_1 C_{v_0\beta}(\omega) + \sqrt{\nu_1(1-\nu_1)}C_{v_0g}(\omega)\right] + \alpha_s
\tag{199}
$$

$$
i\omega C_{v_0g}(\omega) = -\sqrt{\nu_1(1-\nu_1)}\left[\nu_1 C_{v_0\beta}(\omega) + \sqrt{\nu_1(1-\nu_1)}C_{v_1g}(\omega)\right] + \alpha_g
\tag{200}
$$

with $R_\Delta = 1$. By solving the above system at limiting time we get that

$$
C_{v_0\beta}(0) = \frac{\nu_1\alpha_s + \sqrt{\nu_1(1-\nu_1)}\alpha_g}{\nu_1}
\tag{201}
$$

$$
C_{v_0g}(0) = \frac{\sqrt{\nu_1(1-\nu_1)}\left(\nu_1\alpha_s + \sqrt{\nu_1(1-\nu_1)}\alpha_g\right)}{\nu_1^2}
\tag{202}
$$

From these, the loss function is

$$\mathcal{L} = \lim_{\omega,\omega' \to 0} i\omega i\omega' \boldsymbol{v}_0 \cdot \boldsymbol{v}_0 = 1 - \frac{(\nu_1 \alpha_s + \sqrt{\nu_1(1-\nu_1)}\alpha_g)^2}{\nu_1} \tag{203}$$

We list some interesting conclusions that can be derived in this setting.

- The loss, as well as the correlation functions, do not depend on $\nu_2$ in this setting. This is reasonable, since any dependence on the amount of $P_2$ data only comes from how well you can estimate a single scalar coefficient in this rank-1 feature, and that vanishes as the sample size $P_2$ grows.

- As $\nu_1 \to 0$, then $\mathcal{L} = 1 - \alpha_g^2$.

- In the limit where $\nu_1 = 1$ we find $\mathcal{L} = 1 - \alpha_s^2$, which is what one would expect when the learned feature after $\mathcal{T}_1$ is a rank-1 along $\boldsymbol{\beta}_s$. In this case, indeed, the best predictor explains $\alpha_s^2$ fraction of $y_t^2$'s variance, so the residual variance is exactly $1 - \alpha_s^2$.

- If $\alpha_g = 0$, then $\mathcal{L} = 1 - \nu_1 \alpha_s^2$ is a decreasing function of $\nu_1$; if $\alpha_s = 0$, then $\mathcal{L}$ is an increasing function of $\nu_1$.

### C.4 FINE-TUNING ON POLYNOMIAL TASKS

In this small section, we make comparison between the takes of our linear models of fine-tuning, and what actually happens when training a non-linear model on polynomial tasks, from an easy source to a hard target. In Fig. 9 we show that for a data-rich source fine-tuning is always beneficial, while for a data-poor source feature learning on $\mathcal{T}_1$ and related finite-sample size fluctuations can harm performance on the downstream task.

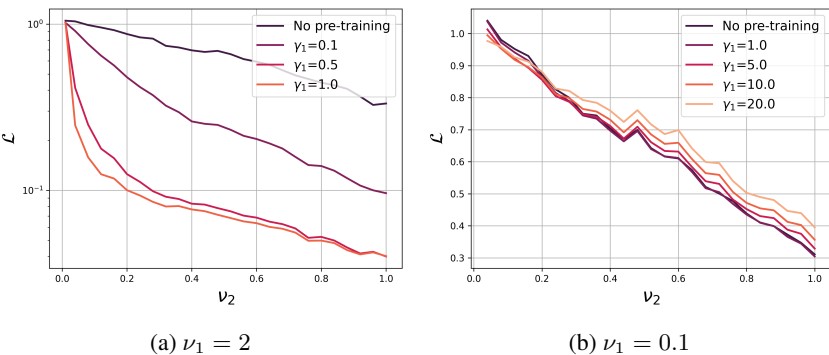

(a) $\nu_1 = 2$                          (b) $\nu_1 = 0.1$

Figure 9: Test loss vs target data $\nu_2$ for different pre-training richness levels $\gamma_1$. Source task is $\mathrm{He}_2(\boldsymbol{\beta}_s \cdot \boldsymbol{x})$, target task is $\mathrm{He}_3(\boldsymbol{\beta}_t \cdot \boldsymbol{x})$ with $\boldsymbol{\beta}_s \cdot \boldsymbol{\beta}_t = 0.8$. (a) When source task is data-rich, fine-tuning is always beneficial and the higher $\gamma_1$, the higher the gain. (b) When source task is data-poor, high feature learning on $\mathcal{T}_1$ can be harmful comparing to no-pretraining.

## D NUMERICAL DETAILS ON DMFT SOLVER

In this section, we provide more details regarding the numerical methods used for solving DMFT fixed point equations as reported in Eq. 6.

To generate the DMFT curves in Figures 3 and 4, we simulate the single-site dynamical mean-field equations as defined in Eq. 6, i.e. we capture the evolution of preactivations $h_{\mu \in \mathcal{T}_1 \cup \mathcal{T}_2}(t)$ and readout variables $z(t)$ via Monte Carlo sampling.

- We start by generating $\mathcal{S} = 50K$ Gaussian Monte Carlo samples of the pre-activation fields at initialization $\mathbf{h}(0) \in \mathbb{R}^{(P_1+P_2) \times \mathcal{S}}$, being $\{P_1, P_2\}$ the sample size of source and target tasks respectively. Given $\mathbf{K}_x = \frac{1}{D}\mathbf{x}\mathbf{x}^\top \in \mathbb{R}^{(P_1+P_2) \times (P_1+P_2)}$ the data Gram matrix

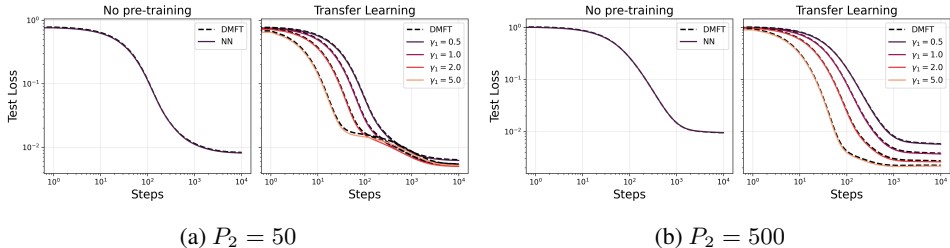

(a) $P_2 = 50$                 (b) $P_2 = 500$

Figure 10: Linear regression on anisotropic data with a power-law spectrum: comparison between training from scratch (left panels) and transfer learning (right panels). (a) Transfer learning provides an initial boost at early-time but the final performance is bottlenecked by small sample on $\mathcal{T}_2$; (b) transfer learning produces a consistent improvement in test loss across training, with richer pre-training ($\gamma_1$) corresponding to higher gains.

$(\mathcal{T}_1 \cup \mathcal{T}_2)$, this is just $\{h_{\mu,n}(0)\}_{n=1}^{\mathcal{S}} \sim \mathcal{N}(0, \mathbf{K}_x)$. In the same way, the readout fields at init are given by $\{z_n(0)_{n=1}\}^{\mathcal{S}} \sim \mathcal{N}(0, 1)$.

- From $\{\mathbf{h}(0), \mathbf{z}(0)\}$, we can evaluate the activation field and the gradient signal at initialization accordingly: $\phi(\mathbf{h}(0)), \mathbf{g}(0)$ (as defined through Eq. 6). The initial error signal on $\mathcal{T}_1$ or $\mathcal{T}_2$ is $\boldsymbol{\Delta}(0) \in \mathbb{R}^{P1 \vee P2}$ and computed as $\boldsymbol{\Delta}(0) = \mathbf{y} - \frac{1}{\gamma_{1/2}\mathcal{S}}\phi(\mathbf{h}(0))\mathbf{z}(0)$.

- At each DMFT time step we update the fields using an explicit Euler discretization of Eq. (6), approximating population averages $\langle \cdot \rangle$ with empirical averages over the $\mathcal{S}$ Monte Carlo samples.

During pretraining, the updates depend only on the residuals $\Delta_\mu$ for train samples in $\mathcal{T}_1$; during transfer learning the updates use only the residuals in $\mathcal{T}_2$. At each time step we compute the training/test losses from the DMFT residuals and estimate the learned feature kernel as $\mathbf{K}(t,t) = \frac{1}{\mathcal{S}}\phi(\mathbf{h}(t))^\top \phi(\mathbf{h}(t))$.

## E    SETTING AND RELATED WORKS FOR BAYESIAN NNS

In this section, we would like to study the effect of transfer learning for infinitely wide Bayesian neural networks. Here, we suppose that a two layer NN with parameters $\boldsymbol{\theta} = \text{Vec}\{\boldsymbol{W}, \boldsymbol{w}\}$ has to learn a target task $\mathcal{T}_2$ composed of $P_2$ input-output pairs $\{\boldsymbol{x}_\mu, y_\mu\}_{\mu=1}^{P_2}$, where the input vector is $\boldsymbol{x}_\mu \in \mathbb{R}^D$, $\{D, P_2\} = \Theta_N(1)$ are fixed, and the network width $N$ is going to infinity. The case where the solution space is sampled from a posterior that is a Gibbs distribution with generic log-likelihood $\mathcal{L}(\boldsymbol{\theta}, \mathcal{T})$ and a Gaussian prior $\frac{1}{2}||\boldsymbol{\theta}||^2$ has been studied in (Lauditi et al., 2025). Here, the purpose is to integrate the effect of transfer learning from a source task $\mathcal{T}_1$ with the effect of feature learning on $\mathcal{T}_2$.

We consider the weights $\bar{\boldsymbol{\theta}} = \text{Vec}\{\bar{\boldsymbol{W}}, \bar{\boldsymbol{w}}\}$ of a pre-trained model on $\mathcal{T}_1 = \{\bar{\boldsymbol{x}}_\mu, \bar{y}_\mu\}_{\mu=1}^{P_1}$ as quenched disorder variables for the target task $\mathcal{T}_2$, since these weights adapt only on $\mathcal{T}_1$, while the target task variables are annealed $\boldsymbol{\theta} = \text{Vec}\{\boldsymbol{W}, \boldsymbol{w}\}$. The quantity of interest we would like to compute is the free energy

$$\mathbb{E}_{\bar{\boldsymbol{W}} \sim p(\bar{\boldsymbol{\theta}}|\mathcal{T}_1)}\mathcal{F}[\bar{\boldsymbol{W}}] = -\lim_{N\to\infty}\frac{1}{N}\mathbb{E}_{\bar{\boldsymbol{W}} \sim p(\bar{\boldsymbol{\theta}}|\mathcal{T}_1)}\ln Z[\bar{\boldsymbol{W}}]$$

$$= -\lim_{N\to\infty}\frac{1}{N}\mathbb{E}_{\bar{\boldsymbol{W}} \sim p(\bar{\boldsymbol{\theta}}|\mathcal{T}_1)}\ln\left[\int d\boldsymbol{\theta}\,\exp\left(-\frac{\beta N\gamma_0^2}{2}\sum_{\mu=1}^{P_2}\mathcal{L}(\boldsymbol{\theta}, \mathcal{T}_2)\right) - \frac{1}{2}||\boldsymbol{\theta}||^2 - \frac{\delta}{2}||\boldsymbol{W} - \bar{\boldsymbol{W}}||^2\right].$$

$$(204)$$

Here, the dependency on the source weights $\bar{\boldsymbol{W}} \in \mathbb{R}^{N\times D}$ appears through an elastic coupling $\delta$ that acts as a form of regularization for the target task weights $\boldsymbol{W} \in \mathbb{R}^{N\times D}$ of $\mathcal{T}_2$. To guarantee that the

source configuration effectively solved $\mathcal{T}_1$, we take the expectation over the posterior distribution of the source weights as sampled from the Gibbs measure

$$p(\bar{\boldsymbol{\theta}}|\mathcal{T}_1) = \frac{1}{\mathcal{Z}_1} \exp\left(-\frac{\beta N \bar{\gamma}_0^2}{2} \sum_{\mu=1}^{P_1} \mathcal{L}(\bar{\boldsymbol{\theta}}, \mathcal{T}_1) - \frac{1}{2}||\bar{\boldsymbol{\theta}}||^2\right). \tag{205}$$

As clarified in the main text, both $\{\bar{\gamma}_0, \gamma_0\} = \Theta_N(1)$ in the mean-field parameterization act as richness parameters that tune the level of feature learning strength, respectively on $\mathcal{T}_1$ and $\mathcal{T}_2$ (Bordelon & Pehlevan, 2022; Bordelon et al., 2024b; Lauditi et al., 2025). This is the reason why, in our theory, representation learning remains an $\Theta_N(1)$ effect at infinite width even when $P = \Theta_N(1)$, contrary to what would happen in the theories of (Li & Sompolinsky, 2021; Pacelli et al., 2023), whose infinitely overparameterized limit $\alpha = P/N \to 0$ recovers the NNGP lazy kernel at infinite width.

The way on constraining the target weights to the source weights through an elastic coupling as in Eq. equation 204 was first proposed by (Ingrosso et al., 2025) in the context of transfer learning and then studied by (Shan et al., 2025) in the continual learning setting. This is common practice in the theory of spin glasses, where the form of Eq. equation 204 is known under the name of Franz-Parisi potential (Franz & Parisi, 1995), used to bias the posterior measure through metastable states in the energy landscape. In the context of machine learning theory, a line of works (Baldassi et al., 2015; 2016; 2019; 2021; 2022) focused on shallow architectures, made use of the Franz-Parisi potential in order to target subdominant flat regions of solutions in the loss landscape of a given task $\mathcal{T}$. Here, we stress that our theory of transfer learning described by Eq. equation 204, leads to different results than the theory of (Ingrosso et al., 2025). The authors of (Ingrosso et al., 2025) focused on a proportional limit where both the size of the training sets ($P_1$, $P_2$ in our notation) and the width $N$ go to infinity with some fixed ratios $\alpha_1 = P_1/N$ and $\alpha_2 = P_2/N$. The network parameterization they study is the standard NTK parameterization. In order to be able to study the proportional limit, they make a Gaussian Equivalence assumption for non-linear activation functions. Their theory predicts that, at finite $\alpha$, the effect of transfer learning occurs due to a renormalization effect of a *fixed* source-target kernel, accordingly to the Bayesian theories of (Li & Sompolinsky, 2021; Pacelli et al., 2023). More importantly, in the $\alpha \to 0$ overparameterized limit we are considering here, their theory predicts that TL has no effect on learning, since they recover the NNGP lazy kernel in this limit.

On the contrary, here we study the effect of mean-field ($\mu P$) parameterization to transfer learning in the overparameterized limit. As clarified by Eq. equation 204, we scale the likelihood by $N$ in order to ensure we get a non-trivial contribution from the likelihood in the infinite width limit, and we scale the network readout with $\gamma_0 N$. The form of our posterior combined with the parameterization we choose allows us to get a theory of feature learning where kernels adapt to data in a non-trivial manner even when $P = \Theta_N(1)$. In fact, as clarified in (Lauditi et al., 2025), the posterior of Eq. equation 205 do not recover the NNGP lazy kernel, and the effect of transfer learning remains non-negligible in our theory at finite $P$. Our theory do not require any Gaussian Equivalence assumptions on the pre-activation distribution. Indeed, the combined effect of feature and transfer learning leads to non-Gaussian pre-activations. We get a set of saddle point equations for the kernels of both source ($\mathcal{T}_1$) and downstream ($\mathcal{T}_2$) tasks that have to be solved self-consistently. Thus, the kernels in our theory are not fixed but adapt to data, because representation learning shapes the pre-activation distribution.

## F    THEORETICAL DERIVATION OF THE FREE ENERGY

Here, we proceed in reporting the actual computation of the free energy in Eq. equation 204. In order to compute the average over the source posterior we use the replica trick $\ln Z = \lim_{n\to 0}\frac{Z^n-1}{n}$, and we introduce a set of $n$ replicas $a \in \{n\}$ for the source weights $\{\boldsymbol{W}^a, \boldsymbol{w}^a\}$. As a consequence, we get

$$
\mathbb{E}Z^n = \int d\bar{\boldsymbol{W}}\, d\bar{\boldsymbol{w}} \prod_{a=1}^n d\boldsymbol{W}^a d\boldsymbol{w}^a df_\mu^a d\bar{f}_\mu^a \exp\left(-\frac{N\beta\gamma_0^2}{2}\sum_{\mu\in\mathcal{T}_1}[\bar{f}_\mu - \bar{y}_\mu]^2 - \frac{N\beta\gamma_0^2}{2}\sum_{a=1}^n\sum_{\mu\in\mathcal{T}_2}[f_\mu^a - y_\mu]^2\right)
$$

$$
\exp\left(-\frac12\sum_{a=1}^n|\boldsymbol{W}^a|^2 - \frac12\sum_{a=1}^n|\boldsymbol{w}^a|^2 - \frac12|\bar{\boldsymbol{w}}|^2 - \frac12|\bar{\boldsymbol{W}}|^2 - \frac{\delta}{2}\sum_{a=1}^n|\boldsymbol{W}^a - \bar{\boldsymbol{W}}|^2\right)
$$

$$
\int \prod_{a,\mu\in\mathcal{T}_1} dh_\mu^a d\hat{h}_\mu^a \prod_{\mu\in\mathcal{T}_2} d\bar{h}_\mu d\hat{\bar{h}}_\mu \exp\left(i\sum_{a=1}^n\sum_{\mu\in\mathcal{T}_2}\hat{h}_\mu^a\left(h_\mu^a - \frac{1}{\sqrt{D}}\boldsymbol{W}^a\boldsymbol{x}_\mu\right) + i\sum_{\mu\in\mathcal{T}_1}\hat{\bar{h}}_\mu\left(\bar{h}_\mu - \frac{1}{\sqrt{D}}\bar{\boldsymbol{W}}\boldsymbol{x}_\mu\right)\right)
$$

$$
\int d\hat{f}_\mu^a d\hat{\bar{f}}_\mu \exp\left(\sum_{a,\mu\in\mathcal{T}_2}\hat{f}_\mu^a\left(N\gamma_0 f_\mu^a - \boldsymbol{w}^a\cdot\phi(\boldsymbol{h}_\mu^a)\right) + \sum_{\mu\in\mathcal{T}_1}\hat{f}_\mu\left(N\gamma_0\bar{f}_\mu - \bar{\boldsymbol{w}}\cdot\phi(\bar{\boldsymbol{h}}_\mu)\right)\right)
\tag{206}
$$

*Step 1.* The first step consists in integrating out over $\boldsymbol{W}^a$ and $\boldsymbol{w}^a$. We will write these as averages over a standard normal matrices (the prior)

$$
\mathbb{E}_{\boldsymbol{W}^a\sim\mathcal{N}(0,(1+\delta)^{-1})}\exp\left(\delta\boldsymbol{W}^a\cdot\bar{\boldsymbol{W}} - \frac{i}{\sqrt{D}}\sum_a\sum_{\mu\in\mathcal{T}_2}\hat{h}_\mu^a\boldsymbol{W}^a\boldsymbol{x}_\mu\right)
$$

$$
= \exp\left(-\frac{1}{2(1+\delta)}\sum_{\mu,\nu\in\mathcal{T}_2}\hat{\boldsymbol{h}}_\mu^a\cdot\hat{\boldsymbol{h}}_\nu^a\, C_{\mu\nu} + \frac{\delta^2}{2(1+\delta)}|\bar{\boldsymbol{W}}|^2 - i\frac{\delta}{1+\delta}\sum_{\mu\in\mathcal{T}_2}\hat{\boldsymbol{h}}_\mu^a\cdot\bar{\boldsymbol{h}}_\mu\right)
$$

$$
\mathbb{E}_{\boldsymbol{w}^a\sim\mathcal{N}(0,1)}\exp\left(-\sum_a\sum_{\mu\in\mathcal{T}_2}\hat{f}_\mu^a\phi(\boldsymbol{h}_\mu^a)\cdot\boldsymbol{w}^a\right) = \exp\left(\frac{N}{2}\sum_a\sum_{\mu,\nu\in\mathcal{T}_2}\hat{f}_\mu^a\hat{f}_\nu^a\Phi_{\mu\nu}^a\right). \tag{207}
$$

We see that we must introduce the kernels and their dual variables $\{\Phi_{\mu\nu}^a, \hat{\Phi}_{\mu\nu}\}_{\mu\nu\in\mathcal{T}_2, a\in\{n\}}$ as order parameters, but these are decoupled over replica index

$$
\Phi_{\mu\nu}^a \equiv \frac{1}{N}\phi(\boldsymbol{h}_\mu^a)\cdot\phi(\boldsymbol{h}_\nu^a)
\tag{208}
$$

and enforce their definitions through some Dirac-delta functions

$$
1 = \int d\Phi_{\mu\nu}^a\,\delta\left(\Phi_{\mu\nu}^a - \frac{1}{N}\phi(\boldsymbol{h}_\mu^a)\cdot\phi(\boldsymbol{h}_\nu^a)\right) = \int \frac{d\Phi_{\mu\nu}^a\, d\hat{\Phi}_{\mu\nu}^a}{2\pi}\exp\left(i\hat{\Phi}_{\mu\nu}^a\left(\Phi_{\mu\nu}^a - \frac{1}{N}\phi(\boldsymbol{h}_\mu^a)\cdot\phi(\boldsymbol{h}_\nu^a)\right)\right).
\tag{209}
$$

*Step 2*: integrate over $\bar{\boldsymbol{W}}$ and $\bar{\boldsymbol{w}}$

$$
\mathbb{E}_{\bar{\boldsymbol{W}}}\exp\left(-\frac{\delta n}{2}|\bar{\boldsymbol{W}}|^2 + \frac{\delta^2 n}{2(1+\delta)}|\bar{\boldsymbol{W}}|^2 - \frac{i}{\sqrt{D}}\sum_{\mu\in\mathcal{T}_1\cup\mathcal{T}_2}\hat{\bar{\boldsymbol{h}}}_\mu\bar{\boldsymbol{W}}\boldsymbol{x}_\mu\right)
$$

$$
\sim_{n\to 0}\exp\left(-\frac12\sum_{\mu\nu\in\mathcal{T}_1\cup\mathcal{T}_2}C_{\mu\nu}\hat{\bar{\boldsymbol{h}}}_\mu\cdot\hat{\bar{\boldsymbol{h}}}_\nu\right)
$$

$$
\mathbb{E}_{\bar{\boldsymbol{w}}\sim\mathcal{N}(0,1)}\exp\left(-\sum_{\mu\in\mathcal{T}_1}\hat{\bar{f}}_\mu\phi(\boldsymbol{h}_\mu)\cdot\bar{\boldsymbol{w}}\right) = \exp\left(\frac{N}{2}\sum_{\mu,\nu\in\mathcal{T}_1}\hat{\bar{f}}_\mu\hat{\bar{f}}_\nu\bar{\Phi}_{\mu\nu}\right)
\tag{210}
$$

Here, similarly as we did in Eq. equation 208, we enforce the definitions of the source task kernels $\{\bar{\Phi}_{\mu\nu}, \hat{\bar{\Phi}}_{\mu\nu}\}_{\mu\nu\in\mathcal{T}_1}$, which do not carry any replica index.

*Step 3*: Factorize everything across the $N$ hidden neurons

$$
\langle Z^n \rangle \propto \int d\bar{\Phi} d\hat{\bar{\Phi}} d\bar{f}_\mu d\hat{\bar{f}}_\mu \prod_{a=1}^{n} d\Phi^a d\hat{\Phi}^a df^a d\hat{f}^a \exp\left( -\frac{\beta N \bar{\gamma}_0^2}{2} \sum_{\mu\in\mathcal{T}_1} [\bar{f}_\mu - y_\mu]^2 - \frac{\beta N \gamma_0^2}{2} \sum_a \sum_{\mu\in\mathcal{T}_2} [f_\mu^a - y_\mu]^2 \right)
$$

$$
\exp\left( N\gamma_0 \sum_{\mu a} \hat{f}_\mu^a f_\mu^a + N\bar{\gamma}_0 \sum_\mu \hat{\bar{f}}_\mu \bar{f}_\mu + \frac{N}{2} \sum_{a\mu\nu} \hat{\Phi}_{\mu\nu}^a \Phi_{\mu\nu}^a + \frac{N}{2} \sum_{\mu\nu} \bar{\Phi}_{\mu\nu} \hat{\bar{\Phi}}_{\mu\nu} \right)
$$

$$
\exp\left( \frac{N}{2} \sum_{a\mu\nu} \hat{f}_\mu^a \hat{f}_\nu^a \Phi_{\mu\nu}^a + \frac{N}{2} \sum_{\mu\nu} \hat{\bar{f}}_\mu \hat{\bar{f}}_\mu \bar{\Phi}_{\mu\nu} + N \ln \mathcal{Z}_{joint} \right)
$$

$$\tag{211}$$

where $\mathcal{Z}_{joint}$ is the joint single-site density that carries contributions from both $\mathcal{T}_1$ and $\mathcal{T}_2$. It has the form

$$
\mathcal{Z}_{joint} = \int dh_\mu^a d\hat{h}_\mu^a d\bar{h}_\mu d\hat{\bar{h}}_\mu \exp\left( -\frac{1}{2(1+\delta)} \sum_{a\mu\nu\in\mathcal{T}_2} \hat{h}_\mu^a \hat{h}_\nu^a C_{\mu\nu} - \frac{1}{2} \sum_{a\mu\nu} \phi(h_\mu^a)\phi(h_\nu^a)\hat{\Phi}_{\mu\nu}^a \right)
$$

$$
\exp\left( -\frac{1}{2} \sum_{\mu\nu\in\mathcal{T}_1\cup T_2} \hat{\bar{h}}_\mu \hat{\bar{h}}_\nu C_{\mu\nu} - \frac{1}{2} \sum_{\mu\nu} \phi(\bar{h}_\mu)\phi(\bar{h}_\nu)\hat{\bar{\Phi}}_{\mu\nu} - i\frac{\delta}{1+\delta} \sum_{a\mu} \bar{h}_\mu \hat{h}_\mu^a \right)
$$

$$
\exp\left( i \sum_{a\mu} \hat{h}_\mu^a h_\mu^a + i \sum_\mu \hat{\bar{h}}_\mu \bar{h}_\mu \right).
$$

$$\tag{212}$$

Notice that, if $\delta = 0$ in Eq. equation 212, the single site densities on $\mathcal{T}_1$ and $\mathcal{T}_2$ are perfectly decoupled as it should be, since no transfer learning effect would come into play. Instead, as soon as we keep $\delta > 0$, there is an interaction between the fields of the source task $\bar{h}$ and the dual fields of the target task $\hat{h}^a$ that will modify the $p(h^a)$ distribution as we show in the next section.

### F.1 RS ANSATZ

*Step 3*: Staring at these equations the only solution that makes sense is the Replica-Symmetric solution $\Phi^a = \Phi$ and $f^a = f$. Plugging this ansatz into the expressions and taking the $n \to 0$ limit, we get

$$
\ln \mathcal{Z}_{joint} = \ln \int d\bar{h} d\hat{\bar{h}} \exp\left( -\frac{1}{2} \sum_{\mu\nu\in\mathcal{T}_1\cup T_2} \hat{\bar{h}}_\mu \hat{\bar{h}}_\nu C_{\mu\nu} - \frac{1}{2} \sum_{\mu\nu\in\mathcal{T}_1} \phi(\bar{h}_\mu)\phi(\bar{h}_\nu)\hat{\bar{\Phi}}_{\mu\nu} + i \sum_{\mu\in\mathcal{T}_1\cup\mathcal{T}_2} \hat{\bar{h}}_\mu \bar{h}_\mu \right)
$$

$$
\times \exp\left( n \ln \mathcal{Z}_2[\bar{h}] \right)
$$

$$
= \ln \mathcal{Z}_1 + \ln\left[ 1 + n \langle \ln \mathcal{Z}_2[\bar{h}] \rangle_1 \right] \sim \ln \mathcal{Z}_1 + n \langle \ln \mathcal{Z}_2[\bar{h}] \rangle_1
$$

where $\ln \mathcal{Z}_2$ is the single site density for task $\mathcal{T}_2$

$$
\mathcal{Z}_2[\bar{h}] = \int dh_\mu d\hat{h}_\mu \exp\left( -\frac{1}{2(1+\delta)} \sum_{\mu\nu\in\mathcal{T}_2} \hat{h}_\mu \hat{h}_\nu C_{\mu\nu} - \frac{1}{2} \sum_{\mu\nu\in\mathcal{T}_2} \phi(h_\mu)\phi(h_\nu)\hat{\Phi}_{\mu\nu} \right)
$$

$$
\times \exp\left( i \sum_\mu \hat{h}_\mu h_\mu - i\frac{\delta}{1+\delta} \sum_\mu \bar{h}_\mu \hat{h}_\mu \right)
$$

$$
= \int dh_\mu \exp\left( -\frac{(1+\delta)}{2} \sum_{\mu\nu} \left( h_\mu - \frac{\delta}{1+\delta} \bar{h}_\mu \right) C_{\mu\nu}^{-1} \left( h_\nu - \frac{\delta}{1+\delta} \bar{h}_\nu \right) - \frac{1}{2} \sum_{\mu\nu\in\mathcal{T}_2} \phi(h_\mu)\phi(h_\nu)\hat{\Phi}_{\mu\nu} \right).
$$

Again, if $\delta = 0$, there would be no dependency on the source task $\mathcal{T}_1$ in Eq. equation 213. We stress that transfer learning has the effect of shifting and scaling all the moments of the distribution $p(\boldsymbol{h})$ towards $p(\bar{\boldsymbol{h}})$ as $\delta$ becomes larger and larger, while feature learning effect on Eq. equation 213 appear through the contribution of the non-Gaussian exponent proportional to the dual kernel $\hat{\boldsymbol{\Phi}}$.

## F.2 SADDLE POINT EQUATIONS

In the infinite width $N \to \infty$ limit the replicated action of Eq. equation 211 is dominated by the set of kernels $\{\bar{\boldsymbol{\Phi}}, \hat{\bar{\boldsymbol{\Phi}}}\} \in \mathcal{T}_1$ and $\{\boldsymbol{\Phi}, \hat{\boldsymbol{\Phi}}\} \in \mathcal{T}_2$ that makes the action $S$ locally stationary ($\delta S = 0$)

$$\langle Z^n \rangle = \int d\bar{\boldsymbol{\Phi}} d\hat{\bar{\boldsymbol{\Phi}}} d\bar{f} d\hat{\bar{f}} \exp\left(N S_1(\{\bar{\boldsymbol{\Phi}}, \hat{\bar{\boldsymbol{\Phi}}}\})\right) \left[\int d\boldsymbol{\Phi} d\hat{\boldsymbol{\Phi}} df d\hat{f} \exp\left(N S_2(\{\boldsymbol{\Phi}, \hat{\boldsymbol{\Phi}}\})\right)\right]^n$$

$$S_1 = \frac{1}{2}\sum_{\mu\nu} \hat{\bar{\Phi}}_{\mu\nu}\bar{\Phi}_{\mu\nu} + \frac{1}{2}\sum_{\mu\nu} \hat{\bar{f}}_\mu \hat{\bar{f}}_\nu \bar{\Phi}_{\mu\nu} + \bar{\gamma}_0 \sum_\mu \hat{\bar{f}}_\mu \bar{f}_\mu - \frac{\beta\bar{\gamma}_0^2}{2}\sum_\mu [\bar{f}_\mu - \bar{y}_\mu]^2 + \ln \mathcal{Z}_1$$

$$\mathcal{Z}_1 = \int d\bar{h}_\mu d\hat{\bar{h}}_\mu \exp\left(-\frac{1}{2}\sum_{\mu\nu} \hat{\bar{\Phi}}_{\mu\nu}\phi(\bar{h}_\mu)\phi(\bar{h}_\nu) - \frac{1}{2}\sum_{\mu\nu} \hat{\bar{h}}_\mu \hat{\bar{h}}_\nu C_{\mu\nu} + i\sum_\mu \hat{\bar{h}}_\mu \bar{h}_\mu\right)$$

$$S_2 = \gamma_0 \sum_\mu f_\mu \hat{f}_\mu + \frac{1}{2}\sum_{\mu\nu} \hat{f}_\mu \hat{f}_\nu \Phi_{\mu\nu} - \frac{\beta\gamma_0^2}{2}\sum_\mu [f_\mu - y_\mu]^2 + \frac{1}{2}\sum_{\mu\nu} \hat{\Phi}_{\mu\nu}\Phi_{\mu\nu} + \langle \ln \mathcal{Z}_2[\bar{h}]\rangle_1$$

$$\mathcal{Z}_2 = \int dh_\mu d\hat{h}_\mu \exp\left(-\frac{1}{2(1+\delta)}\hat{h}_\mu \hat{h}_\nu C_{\mu\nu} - \frac{1}{2}\sum_{\mu\nu} \hat{\Phi}_{\mu\nu}\phi(h_\mu)\phi(h_\nu) + i\sum_\mu \hat{h}_\mu(h_\mu - \delta(1+\delta)^{-1}\bar{h}_\mu).\right)$$

$$(213)$$

From these definitions, the saddle point equations give

$$\frac{\partial S}{\partial \hat{\bar{\Phi}}} = \frac{1}{2}\bar{\Phi} - \frac{1}{2}\left\langle \phi(\bar{h})\phi(\bar{h})\right\rangle_1 + \mathcal{O}(n)$$

$$\frac{\partial S}{\partial \hat{\Phi}} = \frac{1}{2}\Phi_{\mu\nu} - \frac{1}{2}\left\langle \langle \phi(h_\mu)\phi(h_\nu)\rangle_{\cdot|\bar{h}}\right\rangle_{\bar{h}} = 0$$

$$\frac{\partial S}{\partial f_\mu} = \gamma_0 \hat{f}_\mu - \beta\gamma_0^2 [f_\mu - y_\mu] = 0$$

$$\frac{\partial S}{\partial \hat{f}_\mu} = \sum_\nu \Phi_{\mu\nu} \hat{f}_\nu + \gamma_0 f_\mu = 0$$

$$\frac{\partial S}{\partial \Phi} = \hat{\Phi}_{\mu\nu} + \frac{1}{2}\hat{f}_\mu \hat{f}_\nu = 0 \qquad (214)$$

## F.3 REGRESSION TASKS

These equations are generic for any loss function $\mathcal{L}(\boldsymbol{\theta}, \mathcal{T})$. In the following, for simplicity, we will specialize to regression problems where $\mathcal{L}(\boldsymbol{\theta}, \mathcal{T}) = \frac{1}{2}\sum_{\mu=1}^P (f_\mu - y_\mu)^2$ for both source and target tasks. In this particular case, one can solve for both $\{\hat{\bar{f}}_\mu, \hat{\bar{f}}_\mu\}$ and $\{f_\mu, \hat{f}_\mu\}$ explicitly, since the squared-error loss (SE) allows to integrate out the last layer readouts. From that, one gets for the dual source and target kernels

$$\hat{\bar{\boldsymbol{\Phi}}} = -\bar{\gamma}_0^2\left(\frac{\boldsymbol{I}}{\beta} + \bar{\boldsymbol{\Phi}}\right)^{-1}\bar{\boldsymbol{y}}\bar{\boldsymbol{y}}^\top\left(\frac{\boldsymbol{I}}{\beta} + \bar{\boldsymbol{\Phi}}\right)^{-1}$$

$$\hat{\boldsymbol{\Phi}} = -\gamma_0^2\left(\frac{\boldsymbol{I}}{\beta} + \boldsymbol{\Phi}\right)^{-1}\boldsymbol{y}\boldsymbol{y}^\top\left(\frac{\boldsymbol{I}}{\beta} + \boldsymbol{\Phi}\right)^{-1}. \qquad (215)$$

Notice that the two equations are functionally equivalent, but what changes is the dependency on different task labels $\{\bar{\boldsymbol{y}}\} \in \mathcal{T}_1$ vs $\{\boldsymbol{y}\} \in \mathcal{T}_2$, different levels of feature learning strength in principle $\{\bar{\gamma}_0, \gamma_0\}$, and especially different adaptive kernels $\bar{\boldsymbol{\Phi}}$ vs $\boldsymbol{\Phi}$.

## F.4 GENERALIZATION ERROR

Knowing the form of the transfer free energy of Eq. equation 204, makes it easy to compute the test error of the target model on a new (unseen) example $(\boldsymbol{x}_0, y_0)$. For a generic loss, this is defined as

$$\epsilon_g(\boldsymbol{x}_0, y_0) = \mathbb{E}_{\bar{\boldsymbol{W}} \sim p(\bar{\boldsymbol{\theta}} | \mathcal{T}_1)} \langle \mathcal{L}(\boldsymbol{\theta}; \{\boldsymbol{x}_0, y_0\}) \rangle_{\boldsymbol{\theta} \sim p(\boldsymbol{\theta} | \mathcal{T}_2, \bar{\boldsymbol{W}})} \tag{216}$$

and can be easily computed by realizing that, if we introduce a "test-point coupling" $\epsilon$ into the transfer free energy by adding a weighted loss for the unseen sample $(\boldsymbol{x}_0, y_0)$, we get an extended free energy

$$\mathcal{F}(\epsilon) = -\lim_{N \to \infty} \frac{1}{N} \mathbb{E}_{\bar{\boldsymbol{W}} \sim p(\bar{\boldsymbol{\theta}} | \mathcal{T}_1)} \ln \int d\boldsymbol{\theta} \, \exp\left(-\frac{\beta N \gamma_0^2}{2} \left(\sum_{\mu \in \mathcal{T}_2} \mathcal{L}(\boldsymbol{\theta}; \mathcal{T}_2) + \epsilon \mathcal{L}(\boldsymbol{\theta}; \{\boldsymbol{x}_0, y_0\})\right)\right)$$
$$\times \exp\left(-\frac{1}{2}||\boldsymbol{\theta}||^2 - \frac{\delta}{2}||\boldsymbol{W} - \bar{\boldsymbol{W}}||^2\right)$$

from which the test loss can be easily computed as

$$\epsilon_g = \frac{2}{\beta \gamma_0^2} \frac{\partial \mathcal{F}(\epsilon)}{\partial \epsilon}\bigg|_{\epsilon=0}. \tag{217}$$

For regression task and SE loss, consistently with (Lauditi et al., 2025), this gives the kernel predictor

$$\epsilon_g(\boldsymbol{x}_0, y_0) = \left(y_0 - \sum_{\mu\nu} \Phi_{0\mu}\left[\Phi_{\mu\nu} + \frac{\mathbb{I}_{\mu\nu}}{\beta}\right]^{-1} y_\nu\right)^2 \tag{218}$$

being $\boldsymbol{\Phi}_{0\mathcal{T}_2}$ the train-test kernel from the saddle point equation

$$\Phi_{0\mu} = \left\langle \langle \phi(h_0)\phi(h_\mu) \rangle_{\cdot | \{\bar{h}_0, \bar{h}\}} \right\rangle_{\{\bar{h}_0, \bar{h}\}} \tag{219}$$

similarly to Eq. equation 214 for the train kernel. We explicitly derive the close form of the train-test kernel for linear networks in the following Sec. 'F.5.

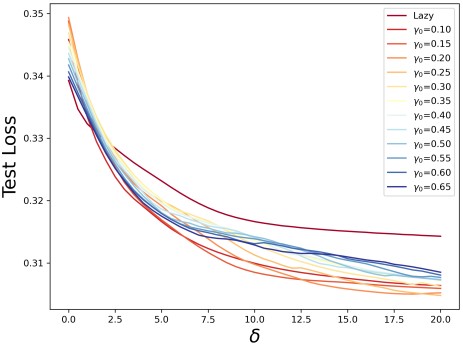

Figure 11: Langevin sampling from the energy function given in Eq. 204. Two-layer ReLU network with width $N = 20000$ as a function of $\delta$ and for different feature learning strength values $\gamma_0$. Test loss at convergence: the network is trained for $10^5$ and averaged after $t = 5 \times 10^4$ every $10^3$ steps. Lazy learning are smallest benefit from transfer learning. Optimal intermediate value of $\gamma_0$.

## F.5 LINEAR NETWORKS

If we specialize to linear networks where $\phi(h) \equiv h$ and to regression tasks, the target action can be solved explicitly. Indeed, this is given by

$$S_2 = -\frac{1}{2}\sum_{\mu\nu} \Phi_{\mu\nu}\hat{\Phi}_{\mu\nu} + \frac{\gamma_0^2}{2}\boldsymbol{y}^\top\left(\boldsymbol{\Phi} + \frac{\boldsymbol{I}}{\beta}\right)^{-1}\boldsymbol{y} - \langle \ln \mathcal{Z}_2[\bar{\boldsymbol{h}}] \rangle_1 \tag{220}$$

where the single-site remains now Gaussian even after feature learning, being

$$\mathcal{Z}_2 = \int dh_\mu d\hat{h}_\mu \exp\left(-\frac{1}{2(1+\delta)}\hat{h}_\mu\hat{h}_\nu C_{\mu\nu} - \frac{1}{2}\sum_{\mu\nu}\hat{\Phi}_{\mu\nu}h_\mu h_\nu + i\sum_\mu \hat{h}_\mu(h_\mu - \delta(1+\delta)^{-1}\bar{h}_\mu)\right).$$

(221)

Here, we can think $\hat{h}, h$ as jointly Gaussian with

$$\begin{bmatrix}\hat{h}\\h\end{bmatrix} \sim \mathcal{N}(\boldsymbol{\mu}, \boldsymbol{\Sigma})$$

$$\boldsymbol{\mu} = \begin{bmatrix}(1+\delta)^{-1}\boldsymbol{C} & -i\boldsymbol{I}\\-i\boldsymbol{I} & \hat{\boldsymbol{\Phi}}\end{bmatrix}^{-1}\begin{bmatrix}-i\delta(1+\delta)^{-1}\bar{h}\\0\end{bmatrix}, \quad \boldsymbol{\Sigma} = \begin{bmatrix}(1+\delta)^{-1}\boldsymbol{C} & -i\boldsymbol{I}\\-i\boldsymbol{I} & \hat{\boldsymbol{\Phi}}\end{bmatrix}^{-1}.$$

The mean and covariance are equal to

$$\langle h\rangle_{\cdot|h} = \delta\left[(1+\delta)\boldsymbol{C}^{-1} + \hat{\boldsymbol{\Phi}}\right]^{-1}\boldsymbol{C}^{-1}\bar{h}, \quad \text{Cov}_{\cdot|\bar{h}}(h) = \left[(1+\delta)\boldsymbol{C}^{-1} + \hat{\boldsymbol{\Phi}}\right]^{-1}.$$

(222)

We can thus compute the correlation of $h|\bar{h}$ as $\langle hh^\top\rangle = \langle h\rangle\langle h\rangle^\top + \text{Cov}(h)$

$$\langle hh^\top\rangle_{\cdot|\bar{h}} = \left[(1+\delta)\boldsymbol{C}^{-1} + \hat{\boldsymbol{\Phi}}\right]^{-1} + \delta^2\left[(1+\delta)\boldsymbol{C}^{-1} + \hat{\boldsymbol{\Phi}}\right]^{-1}\boldsymbol{C}^{-1}\bar{h}_{\mathcal{T}_2}\bar{h}_{\mathcal{T}_2}^\top\boldsymbol{C}^{-1}\left[(1+\delta)\boldsymbol{C}^{-1} + \hat{\boldsymbol{\Phi}}\right]^{-1}.$$

(223)

Now, we must perform the covariance of $\bar{h}$ using $\mathcal{Z}_1$. Note that this is technically $\bar{h}$ restricted to the second dataset $\mathcal{T}_2$. The full covariance of $\bar{h}$ for both $\mathcal{T}_1 \cup \mathcal{T}_2$ has the structure

$$\langle\bar{h}\bar{h}^\top\rangle = \left[\boldsymbol{C}_{\mathcal{T}_1\cup\mathcal{T}_2}^{-1} + \begin{bmatrix}\hat{\bar{\boldsymbol{\Phi}}} & \boldsymbol{0}\\\boldsymbol{0} & \boldsymbol{0}\end{bmatrix}\right]^{-1} = \boldsymbol{C}_{\mathcal{T}_1\cup\mathcal{T}_2}\left[\boldsymbol{I} + \begin{bmatrix}\hat{\bar{\boldsymbol{\Phi}}} & \boldsymbol{0}\\\boldsymbol{0} & \boldsymbol{0}\end{bmatrix}\boldsymbol{C}_{\mathcal{T}_1\cup\mathcal{T}_2}\right]^{-1}.$$

(224)

We are interested in the lower $(2, 2)$ block of this matrix, which gives the Schur complement

$$\langle\bar{h}_{\mathcal{T}_2}\bar{h}_{\mathcal{T}_2}^\top\rangle = \left[[\boldsymbol{C}^{-1}]_{22} - [\boldsymbol{C}^{-1}]_{21}\left([\boldsymbol{C}^{-1}]_{11} + \hat{\bar{\boldsymbol{\Phi}}}\right)^{-1}[\boldsymbol{C}^{-1}]_{12}\right]^{-1}.$$

(225)

Thus we are left with the final equations for the target kernels

$$\boldsymbol{\Phi} = \left[(1+\delta)\boldsymbol{C}_{\mathcal{T}_2}^{-1} + \hat{\boldsymbol{\Phi}}\right]^{-1}$$

$$+ \delta^2\left[(1+\delta)\boldsymbol{C}_{\mathcal{T}_2}^{-1} + \hat{\boldsymbol{\Phi}}\right]^{-1}\boldsymbol{C}_{\mathcal{T}_2}^{-1}\left[[\boldsymbol{C}^{-1}]_{22} - [\boldsymbol{C}^{-1}]_{21}\left([\boldsymbol{C}^{-1}]_{11} + \hat{\bar{\boldsymbol{\Phi}}}\right)^{-1}[\boldsymbol{C}^{-1}]_{12}\right]^{-1}\boldsymbol{C}_{\mathcal{T}_2}^{-1}\left[(1+\delta)\boldsymbol{C}_{\mathcal{T}_2}^{-1} + \hat{\boldsymbol{\Phi}}\right]^{-1}$$

(226)

$$\hat{\boldsymbol{\Phi}} = -\gamma_0^2\left(\boldsymbol{\Phi} + \beta^{-1}\boldsymbol{I}\right)^{-1}\boldsymbol{y}\boldsymbol{y}^\top\left(\boldsymbol{\Phi} + \beta^{-1}\boldsymbol{I}\right)^{-1}$$

(227)

being the action

$$S_2 = -\frac{1}{2}\text{Tr}(\boldsymbol{\Phi}\hat{\boldsymbol{\Phi}}) + \frac{\gamma_0^2}{2}\boldsymbol{y}^\top\left(\boldsymbol{\Phi} + \frac{\boldsymbol{I}}{\beta}\right)^{-1}\boldsymbol{y} + \frac{1}{2}\ln\det\left[\boldsymbol{I} + \left(\frac{\boldsymbol{C}_{\mathcal{T}_2}}{1+\delta}\right)\hat{\boldsymbol{\Phi}}\right]$$

$$-\frac{\delta^2}{2}\text{Tr}\left(\left[(\boldsymbol{C}_{\mathcal{T}_2})^{-1}\left[(1+\delta)(\boldsymbol{C}_{\mathcal{T}_2})^{-1} + \hat{\boldsymbol{\Phi}}\right]^{-1}(\boldsymbol{C}_{\mathcal{T}_2})^{-1}\right]\left[\langle\bar{h}_{\mathcal{T}_2}\bar{h}_{\mathcal{T}_2}^\top\rangle\right]\right).$$

The saddle point equations for the source kernels were firstly derived in (Lauditi et al., 2025) and are instead

$$\bar{\boldsymbol{\Phi}} = \left[\boldsymbol{C}_{\mathcal{T}_1}^{-1} + \hat{\bar{\boldsymbol{\Phi}}}\right]^{-1}$$

$$\hat{\bar{\boldsymbol{\Phi}}} = -\bar{\gamma}_0^2\left(\bar{\boldsymbol{\Phi}} + \beta^{-1}\boldsymbol{I}\right)^{-1}\bar{\boldsymbol{y}}\bar{\boldsymbol{y}}^\top\left(\bar{\boldsymbol{\Phi}} + \beta^{-1}\boldsymbol{I}\right)^{-1}.$$

(228)

### F.5.1 TRAIN-TEST ADAPTIVE KERNELS

In order to compute the test-train kernel to get the network predictor in the linear case, we need to compute $\boldsymbol{\Phi}_{0T} = \langle \boldsymbol{h}_0 \boldsymbol{h}^\top \rangle = \langle \boldsymbol{h}_0 \rangle \langle \boldsymbol{h}^\top \rangle + \mathrm{Cov}(\boldsymbol{h}_0, \boldsymbol{h}^\top)$. The covariance is computed by resorting to the single site extended to the test point with index 0

$$\mathcal{Z}_2[\bar{h}] \propto \int \prod_{\mu=0}^{P_2} dh_\mu \exp\left( -\frac{1}{2} \sum_{\mu\nu=0}^{P_2} \left( h_\mu - \frac{\eta}{1+\eta}\bar{h}_\mu \right) \left( \frac{C_{\mu\nu}}{1+\eta} \right)^{-1} \left( h_\nu - \frac{\eta}{1+\eta}\bar{h}_\nu \right) - \frac{1}{2}\sum_{\mu\nu=1}^{P_2} h_\mu h_\nu \hat{\Phi}_{\mu\nu} \right) \tag{229}$$

from which

$$\left[ \boldsymbol{\Lambda} = \left( (1+\eta)\boldsymbol{C}^{-1} + \begin{pmatrix} 0 & 0 \\ 0 & \hat{\boldsymbol{\Phi}} \end{pmatrix} \right)^{-1} \right] \tag{230}$$

and $\mathrm{Cov}(\boldsymbol{h}_0, \boldsymbol{h}^\top) = \boldsymbol{\Lambda}_{0T}$. It remains to compute

$$\begin{pmatrix} \langle \boldsymbol{h}_0 \rangle_{\cdot|\bar{h}} \\ \langle \boldsymbol{h} \rangle_{\cdot|\bar{h}} \end{pmatrix} = \eta \begin{pmatrix} \boldsymbol{\Lambda}_{00}(\boldsymbol{C}_{00}^{-1}\bar{h}_0 + \boldsymbol{C}_{0T}^{-1}\bar{h}) + \boldsymbol{\Lambda}_{0T}(\boldsymbol{C}_{T0}^{-1}\bar{h}_0 + \boldsymbol{C}_{TT}^{-1}\bar{h}) \\ \boldsymbol{\Lambda}_{T0}(\boldsymbol{C}_{00}^{-1}\bar{h}_0 + \boldsymbol{C}_{0T}^{-1}\bar{h}) + \boldsymbol{\Lambda}_{TT}(\boldsymbol{C}_{T0}^{-1}\bar{h}_0 + \boldsymbol{C}_{TT}^{-1}\bar{h}) \end{pmatrix} \tag{231}$$

where the subscript 0 refers to the test point while $T$ to the training points $P_2 \in \mathcal{T}_2$. From the above equation, we get

$$\begin{aligned}
\langle \boldsymbol{h}_0 \rangle_{\cdot|\bar{h}} \langle \boldsymbol{h}^\top \rangle_{\cdot|\bar{h}} =& \eta^2 \boldsymbol{\Lambda}_{00} \Big( \boldsymbol{C}_{00}^{-1}\bar{h}_0\bar{h}_0^\top \boldsymbol{C}_{00}^{-1} + \boldsymbol{C}_{00}^{-1}\bar{h}_0\bar{h}^\top \boldsymbol{C}_{T0}^{-1} + \boldsymbol{C}_{0T}^{-1}\bar{h}\bar{h}_0^\top \boldsymbol{C}_{00}^{-1} + \boldsymbol{C}_{0T}^{-1}\bar{h}\bar{h}^\top \boldsymbol{C}_{T0}^{-1} \Big) \boldsymbol{\Lambda}_{0T} \\
&+ \eta^2 \boldsymbol{\Lambda}_{00} \Big( \boldsymbol{C}_{00}^{-1}\bar{h}_0\bar{h}_0^\top \boldsymbol{C}_{0T}^{-1} + \boldsymbol{C}_{00}^{-1}\bar{h}_0\bar{h}^\top \boldsymbol{C}_{TT}^{-1} + \boldsymbol{C}_{0T}^{-1}\bar{h}\bar{h}_0^\top \boldsymbol{C}_{0T}^{-1} + \boldsymbol{C}_{0T}^{-1}\bar{h}\bar{h}^\top \boldsymbol{C}_{TT}^{-1} \Big) \boldsymbol{\Lambda}_{TT} \\
&+ \eta^2 \boldsymbol{\Lambda}_{0T} \Big( \boldsymbol{C}_{T0}^{-1}\bar{h}_0\bar{h}_0^\top \boldsymbol{C}_{00}^{-1} + \boldsymbol{C}_{T0}^{-1}\bar{h}_0\bar{h}^\top \boldsymbol{C}_{T0}^{-1} + \boldsymbol{C}_{TT}^{-1}\bar{h}\bar{h}_0^\top \boldsymbol{C}_{00}^{-1} + \boldsymbol{C}_{TT}^{-1}\bar{h}\bar{h}^\top \boldsymbol{C}_{T0}^{-1} \Big) \boldsymbol{\Lambda}_{0T} \\
&+ \eta^2 \boldsymbol{\Lambda}_{0T} \Big( \boldsymbol{C}_{T0}^{-1}\bar{h}_0\bar{h}_0^\top \boldsymbol{C}_{0T}^{-1} + \boldsymbol{C}_{T0}^{-1}\bar{h}_0\bar{h}^\top \boldsymbol{C}_{TT}^{-1} + \boldsymbol{C}_{TT}^{-1}\bar{h}\bar{h}_0^\top \boldsymbol{C}_{0T}^{-1} + \boldsymbol{C}_{TT}^{-1}\bar{h}\bar{h}^\top \boldsymbol{C}_{TT}^{-1} \Big) \boldsymbol{\Lambda}_{TT}.
\end{aligned} \tag{232}$$

As we did for the train kernels in the previous section, we are now interested in the lower $(2,2)$ block of each kernel matrix $\langle \bar{h}\bar{h}^\top \rangle_{\mathcal{T}_2}$ in Eq. equation 232, which would give the source kernel predictions of train and test kernels on $\mathcal{T}_2$, having learned the source task $\mathcal{T}_1$.

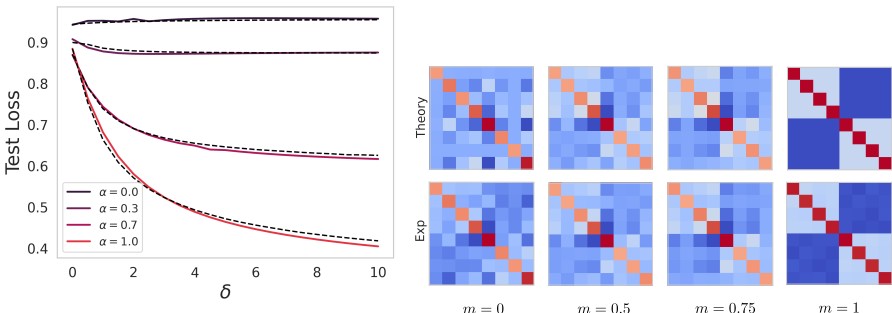

(a) Alignment and Elastic Term Improve Transfer

(b) Adaptive Feature Kernels

Figure 12: The benefit of transfer learning increases with the similarity between source and target tasks. (a) Test losses of a two-layer linear model as a function of the elastic coupling $\delta$ for different levels $\alpha$ of task-similarity. Data are generated from an isotropic Gaussian distribution $\boldsymbol{x} \sim \mathcal{N}(0, \boldsymbol{I})$. Target vector is given by a linear model $\boldsymbol{y} = \boldsymbol{w} \cdot \boldsymbol{x}$ with $||\boldsymbol{w}||_2 = 1$. Here, the target depends on the source task vector $\boldsymbol{\beta}$ (such that $||\boldsymbol{\beta}||_2 = 1$) by the relation $\boldsymbol{w} = \alpha\boldsymbol{\beta} + \sqrt{1 - \alpha^2}\boldsymbol{w}_\perp$ where $\boldsymbol{w} \cdot \boldsymbol{w}_\perp = 0$. Solid lines taken from Langevin sampling on $N = 20000$ network, black dashed lines from Bayesian theory. (b) Target kernels as a function of task similarity $m = \bar{\boldsymbol{y}} \cdot \boldsymbol{y}$.

In this setting, studying the test loss as given by Sec. F.4 as a function of $\delta$ requires to iteratively solve the saddle point equations equation 228 after having the adaptive source kernel values $\{\bar{\boldsymbol{\Phi}}, \hat{\bar{\boldsymbol{\Phi}}}\} \in \mathcal{T}_1$. Fig. 11 shows that, depending on the feature strength $\gamma_0$ value on $\mathcal{T}_2$, transfer learning advantage

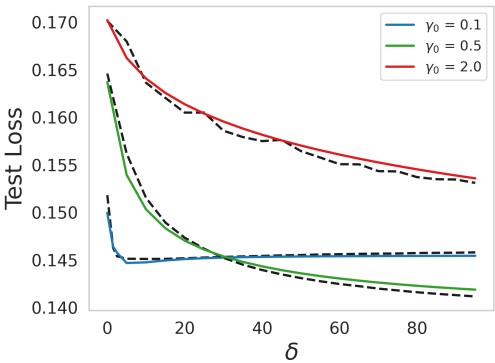

Figure 13: Test losses as a function of the elastic constraint $\eta$. Source task is a regression on two classes (0/1) of MNIST with $P_1 = 400$ labels $\bar{y} \in \{-1, 1\}^{P_1}$ and richness $\bar{\gamma}_0 = 0.5$. Target task is a regression on two classes of Fashion MNIST (2/5) with $P_2 = 50$ data points and labels $y \in \{-1, 1\}_2^P$ for different $\gamma_0$.

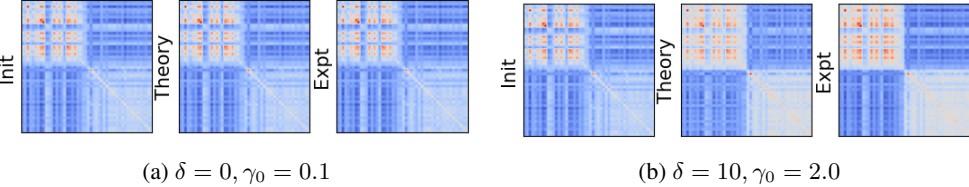

(a) $\delta = 0, \gamma_0 = 0.1$                    (b) $\delta = 10, \gamma_0 = 2.0$

Figure 14: Kernels clustered by labels $y = \{\pm 1\}^{P_2}$ ($P_2 = 50$ Fashion-MNIST data from classes 2/5) improve their task alignment with $\delta > 0$ and high $\gamma_0$. "Init" represents the Gram matrix of data, "Theory" and "Expt" refers to the adaptive feature kernels $\boldsymbol{\Phi}$.

and so the dependency of test loss to $\delta$ may vary. When $\gamma_0$ is small and the target network is almost lazy on $\mathcal{T}_2$, transfer learning has a minor effect in improving the test performance. There exists some optimal values of feature learning strength $\gamma_0$ and $\delta$ (which tunes how much the target network relies on source task features) which optimizes the network performance. In Fig. 14 we clearly show how the clustering of data points by labels pops out in the kernel appearance as soon as we both tune $\gamma_0$ and $\delta$.

### F.5.2   DECOUPLED $C_{\mathcal{T}_1 \cup \mathcal{T}_2}$

A special case we can study is the one in which data are whitened, and uncorrelated across both source and target tasks, meaning

$$C_{\mathcal{T}_1 \cup \mathcal{T}_2} = \begin{bmatrix} \boldsymbol{I} & \boldsymbol{0} \\ \boldsymbol{0} & \boldsymbol{I} \end{bmatrix} \tag{233}$$

In this case, we have

$$\left\langle \bar{h}\bar{h}^{\top} \right\rangle = \boldsymbol{I} \tag{234}$$

which simplifies the kernel saddle points on target task as

$$\boldsymbol{\Phi} = \left[ (1+\delta)\boldsymbol{I} + \hat{\boldsymbol{\Phi}} \right]^{-1} + \delta^2 \left[ (1+\delta)\boldsymbol{I} + \hat{\boldsymbol{\Phi}} \right]^{-2}$$
$$\hat{\boldsymbol{\Phi}} = -\gamma_0^2 \left( \boldsymbol{\Phi} + \beta^{-1}\boldsymbol{I} \right)^{-1} \boldsymbol{y}\boldsymbol{y}^{\top} \left( \boldsymbol{\Phi} + \beta^{-1}\boldsymbol{I} \right)^{-1}. \tag{235}$$

As mentioned in the main text, since in this case the kernel only grow in the rank-one $\boldsymbol{y}\boldsymbol{y}^{\top}$ direction, by solving for the overlaps $\boldsymbol{\Phi} = \phi \, \boldsymbol{y}\boldsymbol{y}^{\top}$ and $\hat{\boldsymbol{\Phi}} = \hat{\phi} \, \boldsymbol{y}\boldsymbol{y}^{\top}$, we get

$$\phi = (1+\delta+\hat{\phi})^{-1} + \delta^2 (1+\delta+\hat{\phi})^{-2} \tag{236}$$

and similarly that

$$\hat{\phi} = -\gamma_0^2 (\beta^{-1} + \phi)^{-2}. \tag{237}$$

In the same way, the saddle point equations for the source task $\mathcal{T}_1$ can be simplified in the source direction $\bar{\boldsymbol{y}}\bar{\boldsymbol{y}}^\top$, giving

$$\bar{\phi} = (1 + \hat{\bar{\phi}})^{-1}$$
$$\hat{\bar{\phi}} = -\bar{\gamma}_0^2 (\beta^{-1} + \bar{\phi})^{-2}. \tag{238}$$

Interestingly, here, when $\delta \to \infty$, since source and target tasks are uncorrelated, then $\phi = 1$, which means that the source kernel $\bar{\boldsymbol{\Phi}}$ is the identity along the target direction $\boldsymbol{y}$ as expected.

### F.5.3 SAME DATA ON BOTH TASKS

Another relevant case is the one where both source and target tasks share the same data and labels. If data are whitened, then

$$C_{\mathcal{T}_1 \cup \mathcal{T}_2} = \begin{bmatrix} \boldsymbol{I} & \boldsymbol{I} \\ \boldsymbol{I} & \boldsymbol{I} \end{bmatrix}, \ \langle \bar{\boldsymbol{h}}\bar{\boldsymbol{h}} \rangle = \begin{bmatrix} \boldsymbol{I} & \boldsymbol{I} \\ \boldsymbol{I} & \boldsymbol{I} \end{bmatrix} \begin{bmatrix} \boldsymbol{I} + \hat{\bar{\boldsymbol{\Phi}}} & \hat{\bar{\boldsymbol{\Phi}}} \\ \boldsymbol{0} & \boldsymbol{I} \end{bmatrix}^{-1} \tag{239}$$

which means

$$\langle \bar{\boldsymbol{h}}_2 \bar{\boldsymbol{h}}_2 \rangle = - \left( \boldsymbol{I} + \hat{\bar{\boldsymbol{\Phi}}} \right)^{-1} \hat{\bar{\boldsymbol{\Phi}}} + \boldsymbol{I} = \left( \boldsymbol{I} + \hat{\bar{\boldsymbol{\Phi}}} \right)^{-1} \tag{240}$$

giving

$$\boldsymbol{\Phi} = \left[ (1+\delta)\boldsymbol{I} + \hat{\boldsymbol{\Phi}} \right]^{-1} + \delta^2 \left[ (1+\delta)\boldsymbol{I} + \hat{\boldsymbol{\Phi}} \right]^{-1} \left( \boldsymbol{I} + \hat{\bar{\boldsymbol{\Phi}}} \right)^{-1} \left[ (1+\delta)\boldsymbol{I} + \hat{\boldsymbol{\Phi}} \right]^{-1}. \tag{241}$$

Again, we can solve for the overlaps, knowing that for $\mathcal{T}_1$

$$\bar{\phi} = (1 + \hat{\bar{\phi}})^{-1} \tag{242}$$

$$\hat{\bar{\phi}} = -\bar{\gamma}_0^2 (\beta^{-1} + \bar{\phi})^{-2}. \tag{243}$$

For $\mathcal{T}_2$ we get

$$\phi = (1 + \delta + \hat{\phi})^{-1} + \delta^2 \bar{\phi} (1 + \delta + \hat{\phi})^{-2} \tag{244}$$

$$\hat{\phi} = -\gamma_0^2 (\beta^{-1} + \phi)^{-2}. \tag{245}$$

Contrary to the previous uncorrelated case, here, when the elastic constraint $\delta \to \infty$, then $\phi = \bar{\phi}$ and the target kernel converges to the source kernel as expected.

### F.5.4 SAME DATA, DIFFERENT LABELS

Suppose again that

$$C_{\mathcal{T}_1 \cup \mathcal{T}_2} = \begin{bmatrix} \boldsymbol{I} & \boldsymbol{I} \\ \boldsymbol{I} & \boldsymbol{I} \end{bmatrix} \tag{246}$$

but that in principle, in this case,

From the saddle point equations for $\mathcal{T}_1$, we know that

$$\bar{\boldsymbol{\Phi}} = \boldsymbol{I} + (\bar{\phi} - 1) \, \boldsymbol{y}_1 \boldsymbol{y}_1^\top \tag{247}$$

and since the saddle point equations for $\mathcal{T}_2$ are

$$\boldsymbol{\Phi} = \left[ (1+\eta)\boldsymbol{I} + \hat{\boldsymbol{\Phi}} \right]^{-1} + \eta^2 \left[ (1+\eta)\boldsymbol{I} + \hat{\boldsymbol{\Phi}} \right]^{-1} \left( \boldsymbol{I} + (\bar{\phi} - 1) \, \boldsymbol{y}_1 \boldsymbol{y}_1^\top \right) \left[ (1+\eta)\boldsymbol{I} + \hat{\boldsymbol{\Phi}} \right]^{-1}$$
$$\hat{\boldsymbol{\Phi}} = -\gamma_0^2 (\boldsymbol{\Phi})^{-1} \boldsymbol{y}_2 \boldsymbol{y}_2^\top (\boldsymbol{\Phi})^{-1} \tag{248}$$

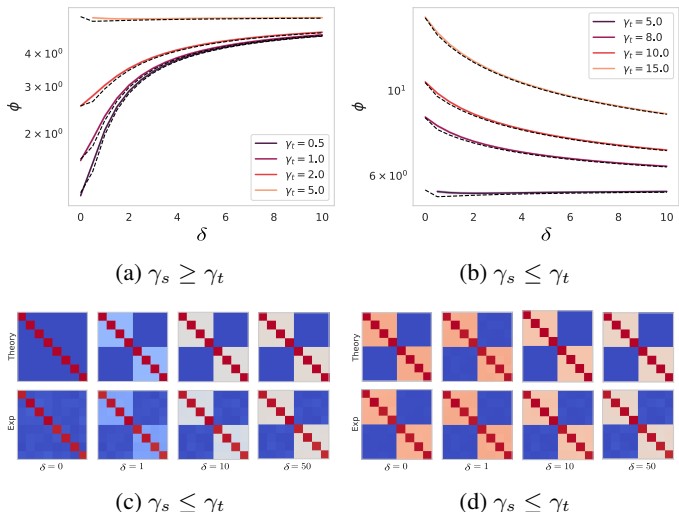

(a) $\gamma_s \geq \gamma_t$        (b) $\gamma_s \leq \gamma_t$

(c) $\gamma_s \leq \gamma_t$        (d) $\gamma_s \leq \gamma_t$

Figure 15: Transfer learning for linear networks trained on whitened data $C = I$ increases the overlap $\phi$ with the label direction $y^\top \Phi y = \phi$ if the source is richer than the target model. (a)/(b) Overlaps $\phi$ vs elastic constraint $\delta$ for a two-layer linear model trained on $P = 8$ patterns with $y = \{\pm 1\}^P$. Source network is pre-trained on the same data as the target, with a richness parameter $\gamma_s = 5.0$. Solid lines taken from Langevin dynamics on $N = 20000$ network, dashed lines from the Bayesian theory. (c)/(d) Examples of learned kernels as a function of the elastic coupling $\delta$.

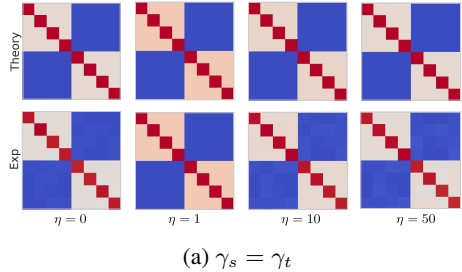

(a) $\gamma_s = \gamma_t$

Figure 16: Kernels (theory vs experiments) as a function of the elastic constraint $\delta$ with the source task ($\mathcal{T}_1$). When $\gamma_s = \gamma_s$, there exists an optimal $\delta$ value for alignment with $\mathcal{T}_2$, since in the target task you saw twice the data than in $\mathcal{T}_1$.

one realizes that the only non-trivial contributions to $\Phi$ comes from the span$\{y_1, y_2\}$, so in principle one can decompose

$$\Phi = a\,I + b\,y_1 y_1^\top + c\,(y_1 y_2^\top + y_2 y_1^\top) + d\,y_2 y_2^\top \tag{249}$$

which means

$$\Phi = aI + \begin{bmatrix} y_1 & y_2 \end{bmatrix} \begin{bmatrix} b & c \\ c & d \end{bmatrix} \begin{bmatrix} y_1^\top \\ y_2^\top \end{bmatrix} \tag{250}$$

from which

$$\Phi^{-1} = \left(aI + uCu^\top\right)^{-1} = a^{-1}I - a^{-2}u\left(C^{-1} + a^{-1}u^\top u\right)^{-1} u^\top \tag{251}$$

and

$$\Phi^{-1}y_2 = a^{-1}y_2 - a^{-2}u\left(C^{-1} + a^{-1}u^\top u\right)^{-1} \begin{bmatrix} y_1^\top y_2 \\ 1 \end{bmatrix} \tag{252}$$

being $y_1^\top y_2 = m$. It turns out, one can solve for $\{a, b, c, d\}$ self consistently and for different values of $m$.

