# OpenReview forum: "Transfer Learning in Infinite Width Feature Learning Networks"
_ICLR.cc/2026/Conference — ICLR 2026 Poster_

### Official Review · Reviewer_xbwa · 2025-10-23

**Soundness:** 3
**Presentation:** 2
**Contribution:** 3
**Rating:** 4
**Confidence:** 3

**Summary:**

This paper develops an analytical theory for transfer learning (TL) in infinitely wide neural networks trained under gradient flow. The authors extend the work on infinite-width feature learning (FL) to the TL setting, focusing on two main paradigms: (i) fine-tuning on top of source features, and (ii) a jointly rich setting where both pretraining and downstream tasks operate in the FL regime. The main contribution is quantifying the generalization benefit of pretraining as a function of the data and model "richness" parameters ($\gamma$) for both source and target tasks. The model is well-motivated by explaining empirical observations, which I believe is the right approach.

**Strengths:**

The main strength is the successful analytical quantification of the transfer learning process in the Feature Learning (FL) regime of infinite-width networks. Unlike the Neural Tangent Kernel (NTK) regime, which predicts no feature learning, this work captures how the source features influence the target task generalization. The results provide explicit, closed-form expressions for metrics like generalization error and the overlap with the label direction. The results are supported by numerical simulations. This is a significant theoretical advance over existing NTK-based TL analyses.
The distinction between fine-tuning and the "jointly rich" setting is interesting. The theory provides concrete and testable conditions, such as the relationship between source and target richness, for when pretraining is beneficial, which aligns well with the experimental results also tested on more realistic data.

**Weaknesses:**

1.	The author assumes the reader is fully familiar with their previous work and neglects to define many key terms in the main text. This makes the paper not accessible to many readers. In addition, common abbreviations and critical analysis components are not defined: DMFT (Line 275), MLP, NTK (Line 051), and the distinction between the limiting objects and their finite counterparts is often blurred. In addition, some of the quantities are not always defined, which makes it hard to read.
2.	Although the theoretical analysis lacks rigor, I believe the method and analysis are robust in certain settings. The underlying model is highly constrained but captures the phenomenon nicely. I believe the analysis relies heavily on strong assumptions about input data isotropy (e.g., isotropic Gaussian inputs), which significantly simplifies real-world datasets. This reliance on simplified, isotropic features is somewhat hidden and must be explicitly stated and discussed as a key limitation, as it fundamentally affects the general applicability of the quantitative results. I believe that the structure of the features can significantly change the results.

The paper is technically sound and achieves a novel, closed-form result for transfer learning in the feature learning regime, which is a significant theoretical contribution, together with a detailed comparison to experiments. However, the highly inaccessible presentation, relying heavily on the reader's deep knowledge of the authors' previous work, is a major drawback. That being said, I’m willing to raise my score, provided the authors improve the presentation by addressing the numerous undefined terms, confusing notation, and clarify statements as I detailed in the questions.

**Questions:**

1.	DMFT, MLP, NTK: Please define these acronyms upon first use (e.g., Line 051, 275).
2.	Result 1 ($\langle \cdot \rangle$): What measure is average $\langle \cdot \rangle$ taken over? Does it include averages over $\phi \sim N(0, 1)$ and $\chi(x) \sim GP(0,K_x)$?
3.	Result 2 ($\chi^\ell$): Please define $\chi^\ell$ and provide a clear interpretation. Also, justify why the baseline for $\mathcal{L}_2$ is $1-\nu_2$.
4.	Error Definitions: Please define $\mathcal{L}(t)$ (Line 217) and $\hat{\mathcal{L}}$ (Eq. 10) explicitly. I assume $\mathcal{L}(t)$ is the loss at time $t$. Why does $\mathcal{L}(t)$ in line 217 have a $K^{1/2}$ on only one term? Also, is $\Delta(t)$ (Line 242) the train error residuals and not the train error? Finally, it could be helpful for readability to use a distinct letter for the limiting object (e.g. $\mathcal{L}(t)$) to avoid confusion.
5.	Appendix A: What is $\Delta(x,t)$?
6.	Appendix B.1: Could you explain Eq. 27 also what is $\delta(t)$? Should it be $P_2$?
7.	Line 1068-1069: Is it clear that the response function is translation-invariant? Also what happens at the transition point?
8.	Similar to 6, could you explain Eq. (98)? I’m probably missing something, but shouldn’t there be an additional $XX^\top$ based on the definition of the Kernel?
9.  Assumption of Input Isotropy: Given the use of Statistical Physics methods, please clearly state the assumptions made about the input data distribution, particularly regarding isotropy (e.g., Gaussian inputs), and discuss how the results might qualitatively change if these assumptions were violated (e.g., for real-world, highly anisotropic datasets).
10. Generality of $\gamma$ Dependence: The main takeaway is that pretraining improves generalization if $\gamma_1 \ge \gamma_2$. Could the authors comment on whether this relationship is expected to hold qualitatively for deep, non-linear networks? For instance, in a deep non-linear network, what surrogate quantity would replace the analytical $\gamma$ to describe the "richness" of the source task/model?
11. Unbounded Feature Learning: Does the result imply that unbounded feature learning ($\gamma_2 \to \infty$) in the target task ($T_2$) does not matter for the final transfer benefit, or is this simply a case not fully explored?
12. Figure 1(c) and $\gamma_0$: What is the parameter $\gamma_0$ referenced in Figure 1(c) and Line 360?
13. In Section 3.1 (Line 377), it states that large $\gamma_2$ can be harmful. Should this be $\gamma_1$? Please clarify the dependence.
14. Optimality of $t_1, t_2$: Figure 3 is compelling. How do the results change if one varies the times $t_1, t_2$? Is there a theoretical way to determine the optimal time for transition to the new task? How do these times depend on the feature learning scales, $\gamma_1, \gamma_2$, and the overparameterization ratios, $\nu_1, \nu_2$?
15: Harmful Large FL Size: In section 3.3. You state in Fig. 4(a) that there is an improvement in the test loss for any value of $\gamma_2$, also in lines 444-445. However, it seems from the figure that for a large $\gamma_2$, it can be harmful. Could you clarify?

---

> ### Author Response · Authors · 2025-11-21
>
> #### Strengths
>
> We thank the Reviewer for appreciating of our theoretical contributions! We are glad that our novel analysis in the feature-learning regime were considered a step forward compared to the existing literature on NTK/lazy regimes.
>
> #### Weaknesses
>
> We are sorry for the resulting lack of accessibility. We have introduced each acronym upon use, and we are currently working on adding an Appendix section to discuss finite-width fluctuations. We hope that this, together with the general list of all the updates made, really improved readability.
>
> *This reliance on simplified, isotropic features is somewhat hidden and must be explicitly stated and discussed as a key limitation.*
>
> We thank the Reviewer for allowing us to clarify this important point. Notice that our Result 1 does not make any assumption on data structure, which is the reason why our theory is able to capture test loss dynamics on real datasets (see Fig. 2(b)/(c) and Fig. 4). Eq. (6) explicitly depends on the empirical data Gram matrix $\mathbf K_x$, which can be arbitrary. However, the toy models of Results 2-4 do assume isotropic data covariance, which is the reason why an average-case analysis of generalization -- through averages over the random datasets $\mathcal T_1, \mathcal T_2$ -- is possible in the first place. In the new Sec. 2 we now clearly state data and model assumtions.
>
> *Highly inaccessible presentation, relying heavily on the reader's deep knowledge of the authors' previous work, is a major drawback.*
>
> We thank the Reviewer for the constructive feedback. We have substantially revised the exposition, clarified all notation, defined key quantities at first use, and improved the presentation of the main results. We hope that this new version addresses the concerns. A point-by-point response is provided below.
>
> #### Questions
> 1. Thank you for catching this. We now define all acronyms (DMFT, MLP, NTK) upon first use.
>
> 2. Yes, the average in Eq. 5 is taken over the single-site fields $\mathbf \chi$ and $\psi$. More explicitly
>
>
> \begin{align}
> \langle \cdot \rangle  \equiv \mathbb{E}_{\chi(x),\psi}.
> \end{align}
>
> We have clarified this in the main text. We also hope that the new Appendix A makes the paper self-consistent.
>
> 3. We defined $\chi^\ell$ which are coefficients for the spike component at layer $\ell$. In our new Appendix A.1, we also have derived the test loss baseline for a linear model performing a regression task with isotropic data. We hope this clarifies why $\mathcal L_2 = 1-\nu_2$.
>
> 4. We have added definitions of these variables in the main text. The variable $\mathcal L(t)$ is the test loss and $\hat{\mathcal L}(t)$ is the training loss. The variable $\Delta(t) \in \mathbb{R}^P$ represents the residual errors $y - f$ on the training set.
>
> 5. We added a definition of this variable $\Delta(x,t) = - \frac{\partial}{\partial f(x,t)} \ell(f(x,t), y)$.
>
> 6. The variable $\delta(t)$ is a Dirac Delta function which we now state. This function can be generated, for example, as a Gaussian with vanishing variance $\delta(t) = \lim_{\sigma \to 0} \frac{1}{\sqrt{2\pi \sigma^2}} \exp( - \frac{1}{2\sigma^2} t^2  )$. This function's key properties are that it vanishes for all $t \neq 0$ but integrates to 1.
>
> 7. Good question! For the linear finetuning that we consider, the response function will be time-translation invariant due to linearity of the dynamics. However, in general DMFT response functions do not have to possess this quality.
>
> 8. This is a good point, thank you for pointing it out. In this equation, we were originally using $K$ incorrectly in this equation. We have since fixed these expressions so that $K$ is used consistently.
> 9. We thank the reviewer for this question. First, we would like to point out that Result 1 does not require any Gaussianity assumption on the data. The result holds for any input distribution. However, the solvable models of transfer in Results 2-4 do assume random isotropic data (we now explicitly state this).
> 10. We thank the reviewer for this question. The parameter $(\gamma_1,\gamma_2)$ will still govern the quality of transfer learning in nonlinear networks. For nonlinear two-layer networks we provide a description of the dependence on these parameters in Result 1. For deeper nonlinear networks, the strength of feature learning is still controlled by these parameters but characterizing them is more involved.
> 11. Our main point is that increasing $\gamma_2$ does not always improve performance, especially in settings where leveraging the featuers learned during pretraining on task $\mathcal T_1$ generates useful features and task $\mathcal T_2$ has less data. However, increasing $\gamma_2$ can help when features need to be learned from scratch and the pretrained features are suboptimal. We illustrate this idea in Figure 3 where for easy->hard using small $\gamma_2$ is better while for hard->easy using large $\gamma_2$ is better.

---

> > ### Author Response · Authors · 2025-11-21
> >
> > 12. We apologize, this should be $\gamma_1$. We have fixed this.
> > 13. Yes, thank you for catching this.
> > 14. We thank the reviewer for this good question there is an interesting effect visible from the theoretical curves and simulations where optimal early stopping times can emerge during training on the downstream task. We have not considered how to compute this optimal early stopping time yet or discovered how to mkae comparisons across $(\gamma_1,\gamma_2)$ at optimal early stopping time but consider this an important future direction.

---

### Official Review · Reviewer_RgUn · 2025-10-30

**Soundness:** 4
**Presentation:** 1
**Contribution:** 3
**Rating:** 4
**Confidence:** 3

**Summary:**

This paper develops a first-principles theory of transfer learning for infinitely wide neural networks operating in the feature-learning (mean-field/µP) regime. The analysis covers two scenarios: fine-tuning with frozen features and a "jointly rich" setting with feature adaptation on the target task. The core of the theoretical work involves deriving analytical results for increasingly simplified models (nonlinear $\rightarrow$ linear $\rightarrow$ linear two layer).The authors leverage the intuition gained from these simplified models to explain the broader phenomenology of transfer learning, identifying the conditions under which it succeeds or fails as a function of dataset sizes, task similarity, and the strength of feature learning.

**Strengths:**

This paper considers transfer learning, which is both an important problem in the field and difficult to analyze analytically. The approach of simplifying a complex system to a tractable toy model to build intuition is a valuable and necessary process. As deep learning models become increasingly complex, developing a single theoretical framework that explains their behavior in its entirety becomes improbable. For this reason, this type of work, which provides rigorous insights into a simplified but representative model, is an excellent and crucial step in the right direction. The theoretical results appear promising and provide a new lens through which to understand the interplay of factors governing transfer success.

Overall, this is a very interesting work. With improvements to the overall writing and clarity, particularly in the presentation of the main results, I would be happy to raise my score.

**Weaknesses:**

The primary weakness of the paper lies in its presentation, which, especially in Section 2, is exceptionally dense. This density hinders the paper's accessibility and, more importantly, obscures the key findings and novel contributions of the work. The following are concrete examples where the writing and structure could be improved:

1.The authors assume the audience is familiar with DMFT (to the point that the acronym is not defined in the main text), which may not be the case for many readers. The paper could greatly benefit from a short primer on DMFT in the main text or a longer appendix dedicated to a brief overview of the key principles relevant to this work.

2. The italicized "result" paragraphs in Section 2 are long and convoluted, making it difficult for the reader to distill the actual new result, understand its specific implications, and separate it from established background knowledge. For example, in Result 2, Eq. 7 is an established result on infinite networks trained on infinite data, not a new finding specific to transfer learning, and should not have been presented as such (separately) . A similar issue occurs in Result 3, but this time it is further complicated with the introduction of a noise term that is not contextualized properly with respect to previous results.

3. The results are presented sequentially in a technically dense section with minimal guiding principle connecting them beyond the change in setting. A short paragraph giving an overview of these results and how they connect could be highly beneficial. Furthermore, while the titles of the results are meant to provide intuition, the link between the plain-English title and the complex mathematics is often not at all intuitive. A clearer bridge is needed to help the reader understand how the mathematical expressions support the high-level takeaway.

**Questions:**

1. Regarding **Result 1**:
(a) The result is stated, but no derivation or methodological sketch is provided in the main text. It would be helpful to include a brief description of the key methods used to obtain this result to make the paper more self-contained.
(b) Can the authors explicitly explain what this result implies beyond the already established fact that the downstream task has a history dependence on the pre-training dynamics? The mathematical formulation is quite involved for what appears to be a known concept.

2. How are the DMFT curves in Figures 3 and 4 obtained? The text states this is from Result 1, but these are non-trivial high-order integral equations. An explanation in the main text of the numerical methods used to solve these equations and generate the curves would be very helpful for reproducibility and understanding.

3. In the phenomenology section (Section 3), it was often not clear which theoretical results were being used to support the empirical findings and simulations. For example, when discussing the CIFAR-10 experiments, it would be beneficial to explicitly state which result (e.g., Result 1, 3, or an ansatz based on them) provides the theoretical basis for the claims being made. This should be clearly emphasized throughout the section.

4. Minor point: In **Result 3**, the text states: “With this kernel, similarly to the sketch of Result 1…”. This seems to be a typo, as the preceding discussion in Result 2 is more relevant. Is it possible the authors were referring to Result 2? If not, it is unclear what sketch from Result 1 is being referenced.

---

> ### Author Response · Authors · 2025-11-21
>
> #### Strengths
> We thank the Reviewer for this encouraging feedback regarding our work. We have substantially revised the writing and clarified the presentation of the main results, and we really hope that this updated version addresses your concerns. Please find our point-by-point responses below.
> #### Weaknesses
> 1. We appreciate this suggestion. We have now added a concise primer in Appendix A that summarizes the key principles of DMFT. We have also added an explicit example for linear regression with isotropic covariance under gradient flow dynamics, and show how to get the test loss at converge (which is the baseline for our linear toy models), i.e. $\mathcal L = 1-P/D$. We hope that this makes the paper more self-contained and the technique much more clear.
>
> 2. We have revised the ‘Result’ paragraphs to improve readability. Concerning Eq. (7), we agree that this adaptive feature kernel is an established result for deep linear networks, and we now cite the relevant prior works accordingly. We also agree that Eq. (11) is the most general ansatz one could think of for kernel of a linear model when the source task $\boldsymbol{\beta}_s$ is uncorrelated with the noise direction $\mathbf g$. However, to the best of of knowledge, we are the first to derive explicitly its form (see Appendix C.2).
>
> 3. Thank you for this useful suggestion. In Section 2.3, we have now added a short guiding paragraph to help the reader following the subsequent results. In particular, we now clearly state the regimes we study to quantify how pretraining affects fine-tuning performance:
>
> i. data-rich pretraining in the population limit $P_1 \to \infty$
>
> ii. finite data pretraining
>
> iii. ultra-rich pretraining with $\gamma_1 \to \infty$.
>
> At the end of each ‘Result’ paragraph we also now include a short explanation that connects the mathematical expression to the attributed title.
>
> #### Questions
>
> 1. (a) We have tried to expand the methodological sketch after this result paragraph. We hope that this extended version makes the main more self-contained. (b) We thank the Reviewer for this question. Our contribution in Result 1 shows that in two-layer models the entire effect of pre-training is compressed into the random variables $\{h (t_1), z(t_1) \}$, and that these obey closed deterministic DMFT equations whose structure we make explicit. This allows us to identify exactly which statistics of the pre-training dynamics influence downstream performance and to isolate the roles of $\gamma_1, \gamma_2$ as well as the dependency on the dataset covariances.
>
> 2. Thank you for this question. To generate the DMFT curves in Figures 3 and 4, we simulate the single-site dynamical mean-field equations as defined in Eq. (6), i.e. we capture the evolution of preactivations $h_{\mu \in \mathcal T_1 \cup \mathcal T_2}(t)$ and readout variables $z(t)$ via Monte Carlo sampling.
>
> i. We start by generating $\mathcal{S} = 50K$ Monte Carlo samples of the pre-activation fields at initialization $\textbf{h}(0)\in \mathbb{R}^{(P_1+P_2) \times \mathcal S}$, being $\{P_1, P_2\}$ the sample size of source and target tasks respectively.
> Given
>
> \begin{align}
> \textbf{K}_x = \frac{1}{D} \textbf{x} \textbf{x}^{\top}\in \mathbb{R}^{(P_1+P_2) \times (P_1+P_2)}
> \end{align}
>
> the data Gram matrix ($\mathcal T_1 \cup \mathcal T_2$), this is just $h_{\mu,n}(0)\sim \mathcal{N}(0,\mathbf K_x)$ with $\mu\in \left[P_1+P_2\right]$ and $n \in \left[\mathcal{S}\right]$. In the same way, the readout fields at init are given by $z_n (0)\sim \mathcal N (0,1)$.
>
> ii. From $\mathbf{h}(0), \mathbf{z}(0)$, we can evaluate the activation field and the gradient signal at initialization accordingly: $\phi (\mathbf{h}(0)), \mathbf g(0)$ (as defined through Eq. 6). The initial error signal on $\mathcal{T}_1$ or $\mathcal{T}_2$ is
> $\mathbf{\Delta} (0)\in \mathbb{R}^{P1 \lor P2}$ and computed as
>
> \begin{align}
> \mathbf{\Delta} (0)= \boldsymbol{y} - \frac{1}{\gamma_{1,2}\mathcal{S}} \phi(\textbf{h}(0)) \textbf{z}(0).
> \end{align}
>
> iii. At each DMFT time step we update the fields using an explicit Euler discretization of Eq. (6), approximating population averages $\langle \cdot \rangle$ with empirical averages over Monte Carlo samples.
>
> During pretraining, the updates depend only on the residuals $\Delta_{\mu}$ for train samples in $\mathcal{T}_1$; during transfer learning only on the residuals of $\mathcal{T}_2$.
> At each time step we compute the training/test losses from the DMFT residuals and estimate the learned feature kernel as $\textbf{K}(t,t) = \frac{1}{\mathcal{S}}\phi(\textbf{h} (t))^{\top} \phi (\textbf{h}(t))$.
>
> We have added a section in Appendix fD or numerical methods. We also provide a brief description in the main text.
>
> 3. We thank the Reviewer for this suggestion. We have revised Section 3 to make the connection explicit. Figure captions now indicate which theoretical prediction each plot illustrates.
>
> 4. This was a typo, thanks. We have corrected the text accordingly.

---

### Official Review · Reviewer_FaLX · 2025-10-30

**Soundness:** 4
**Presentation:** 3
**Contribution:** 3
**Rating:** 8
**Confidence:** 4

**Summary:**

The authors theoretically analyze a wide variety of transfer learning setups in a feature-learning regime. The analysis relies on DMFT calculations; the resulting learning trajectories are typically history-dependent and expensive to numerically evaluate. In more tractable settings (shallow models, linear models) the authors analytically predict various measurables of the trained estimator, revealing conditions under which transfer learning helps or harms. The theory holds qualitatively in practical empirical settings.

**Strengths:**

This is a strong paper which uses powerful technical machinery to extract insights about learning trajectories in a transfer learning setup. It's really nice to have a predictive toy model of this phenomenon -- this is the first result (afaik) that establishes average-case predictions for this feature-learning behavior. The DMFT formalism is quite opaque (to me) but the takeaway messages are clearly communicated.

**Weaknesses:**

1) The authors don't treat the case where fine-tuning lazily updates all weights in the network (rather than just the readout). Do we expect quantitatively similar behavior in this regime?
2) The analytically tractable results rely on an isotropic data assumption which does not hold in many interesting settings. Data anisotropy can qualitatively change the behavior of learning algorithms, especially if $\Sigma_{xx}$ and $\Sigma_{xy}$ do not commute. I think it'd be really interesting to understand how the data geometry interacts with the task geometry, and how the transferability of learned features depends on this interaction! I wish the results touched on this.
3) It's unclear how to think about $\nu_1$. The data-rich limit appears to be $\nu_1\to\infty$, as stated in line 288 and Fig 1a. But the post-training NTK in Result 4 diverges in this limit, and the limit in line 313 seems to contradict the previously-defined data-rich regime.
4) I wish there was more empirics measuring task overlap on real targets :) can't have it all, I guess...

**Questions:**

1) Out of curiosity -- do you think there exist approximations that aren't as restrictive as linear models, but which might give more analytical insight than Result 1? Would it help at all to impose a lot of mutual structure on $\mathcal{T}_1$ and $\mathcal{T}_2$?
2) Fine-tuning can often be made parameter-efficient by using LoRA. Can DMFT-style results extend to the case where the updates on T2 are forced to be low-rank?
3) In result 2, do you assume that $\|\beta_{(\cdot)}\|^2=D$? That seems necessary to ensure $\alpha \in [-1, 1]$.
4) It would be interesting to extend this analysis to prescribe optimal curricula. Do you think it's possible?

---

> ### Author Response · Authors · 2025-11-21
>
> #### Strengths
> *This is a strong paper which uses powerful technical machinery to extract insights about learning trajectories in a transfer learning setup. It's really nice to have a predictive toy model of this phenomenon -- this is the first result (afaik) that establishes average-case predictions for this feature-learning behavior. The DMFT formalism is quite opaque (to me) but the takeaway messages are clearly communicated.*
>
> We thank the Reviewer for the positive assessment and for recognizing the contribution of our work. We are glad that the main takeaways come across clearly.
>
> #### Weaknesses
>
> *1. What about lazy training of all weights?*
>
> This is a good question! At infinite width, lazy fine-tuning of all the weights is equivalent to kernel gradient descent with the NTK evaluated at the pre-trained parameters $\{\mathbf{W}_{source}, \mathbf{a}_{source}\}$, while fine-tuning only the readout corresponds to kernel regression with the feature kernel (the NNGP kernel) computed at the pre-trained features. For our linear toy models the NTK is related to the NNGP through
> \begin{align}
>     \mathbf{K}^{\text{NTK}}(\mathbf{X},\mathbf{X'}) = \mathbf{K}^{\text{NNGP}}(\mathbf{X},\mathbf{X}') + \frac{||\mathbf{a}_{source}||^2}{N}\mathbf{X}\mathbf{X'}^{\top}.
> \end{align}
>
> Here, the second term $||\mathbf{a}_{source}||^2/N$ concentrates in the limit $N\to \infty$, leading to a simple rescaling of the input Gram matrix. We don't expect this correction to qualitatively change the behavior.
>
> *2. What about analysis for structured features?*
>
> We are adding some plots for structured (power law) data (see Fig. 10 in Appendix C4). We agree that developing a data-averaged theory for structured data (different from our current Result 1) could be an interesting future direction! We mentioned that in the discussion.
>
>
> *3. How to think about $\nu_1$? Some formulas don't make sense if $\nu_1 \to \infty$*
>
> Thank you for this point. We were being sloppy in the summary of Result 4. This solution holds on a branch for $\nu_1 < 1$. For $\nu_1 > 1$ there is a different solution. We fixed that in the main text for consistency with Appendix C.3.1.
>
> *4. Overlap on real targets?*
>
> Super interesting question. We agree that empirically quantifying task overlap on real datasets would be very interesting. A way to do this could be through representation-similarity metrics such as centered kernel alignment (CKA). For instance, one could train a deep network on $\mathcal T_1$, and then measure how the CKA similarity between layers changes during fine-tuning on $\mathcal T_2$. This could be a promising direction for future works!
>
>
> #### Questions
>
> *1. Out of curiosity -- do you think there exist approximations that aren't as restrictive as linear models?*
>
> We think so, specifically for two layer networks learning sparse polynomials (like Montanari, Urbani 2025 recent paper). This would require sophisticated use of similar DMFT machinery.
>
> *2. What about LORA?*
>
> Great question! We are interested in this direction, but it would likely need to be in the deeper finetuning setting (rather than the setting where we train only the last layer readout). We suspect that this could be analyzed with similar ideas and tools that we use in this paper.
>
> *3. Is $|\beta|^2 \sim D$?*
>
> Yes!
>
> *4. It would be interesting to extend this analysis to prescribe optimal curricula. Do you think it's possible?*
>
> This is a great future direction. We will mention this in the discussion.

---

### Official Review · Reviewer_djNz · 2025-10-31

**Soundness:** 3
**Presentation:** 2
**Contribution:** 4
**Rating:** 8
**Confidence:** 5

**Summary:**

Summary

The manuscript presents two different theories of transfer learning in
two-layer neuronal networks in the limit of infinite width
where feature learning is assured by muP scaling.

The first theory studies gradient flow and technically employs
dynamical mean field theory to reduce the problem of transfer learning
to an effective single neuron problem on the background of some
self-consistently determined fields.

The second approach (mostly shown in the appendix) studies a more
idealized version of transfer learning in a Bayesian setting, where an
additional "elastic term" is added to the energy function that aligns
weights between the pre- and the post-training phase.

The main part of the manuscript explores the relevant parameters that
control transfer learning and explains qualitative properties, such as
the dependence of the amount of feature learning in the pre- and
post-training phase as well as the alignment of the targets and their
relative complexities (e.g., linear vs higher order polynomial
functions).

The general theoretical results are further illustrated in more
idealized settings (linear activation functions and Gaussian i.i.d.
data) to explain fundamental mechanisms which are subsequently
demonstrated on more real-world data sets (e.g. CIFAR-10, Fashion MNIST).

Overall the manuscript makes an important contribution to the theory
of transfer learning. My main points below are with regard to the
presentation that should still be improved so the work becomes
more accessible also for a broader set of readers.


Soundness

Most of the results appear to be sound and the authors support their
theoretical calculations by comparison to numerical results.

A few points to my mind deserve more explanation:

1.)
In the discussion of Figure 2 it would be important to explain
why panel a) for the theory of the linear model differs qualitatively
from panels b) and c) for the real dataset: Panel a) shows that there
is an optimal feature learning scale \gamma_1^\star (as also stated in the text)
whereas in panel b) and c) that the test loss monotonically
decreases with gamma_1.

2.)
In Figure 3a), the authors should comment on why the asymptotic
value of the test loss is so different between the case of
no pre-training and transfer learning. Should the pre-training
not ultimately become irrelevant at large step sizes of the
second task, so that the two curves are expected to reach the
same asymptotic value?

A similar question for Figure 3b): Here the initial transient
of the test loss seems to be almost identical until the
peak but ultimately at large step sizes, transfer learning
seems to lead to worse generalization than training from scratch.
Why is this?



Presentation

There are a couple of points that the authors should try to improve.
This is to the most extent due to the very dense presentation of
the main text due to the 8 page limit. Here are points I felt
deserve a some care:

1.)
It is confusing to use subscripts _t and _s in section 2.3 to denote
source and target, as the same letters appear as time points.
The authors may want to consider some alternative notation.
Why not use _1 and _2, as also for \mathcal{T}?

2.)
with regard to Result 4:
Please explain the idea of the balance condition in words in the
main text. The appendix is more explicit here.

3.)
Caption of Figure 1:
Remind reader that \alpha is source-target alingment.
Why are you using \alpha_s and \alpha, both of which are defined
in the same way?
Same for \nu_1 \nu_2: remind reader that these are the ratios of P/D.
Panel c: What is the value of \alpha_g for the shown plot?

4.)
A remark would be good explaining why the test loss in Fig 1a
is non-zero at \nu_2 = 0, even if \alpha=1, so source and target task
are identical. Should the loss then not be identical to the test
loss on task 1 alone, so rather small?

5.)
Neither at Eq 13 nor at the discussion of the results in relation
to Fig 1 it is mentioned what is the meaning of the constants c_1 and c_2.
They are explained in the appendix, but a few words in the main text
are needed to understand their meaning, in particular here:

"The simple alignment case (αs =1, αg =0) of Eq. 13 shows that there (i) larger c2 always helps, while (ii) c1 always hurt, since it rotates the high-gain direction towards the noise."

6.)
Figure 5: misalignment of symbol \gamma_2 in caption and \gamma_0
in figure legend.

Appendix B: Here the feature learning strength is defined as \gamma_0,
in the main text as \gamma_1 or \gamma_2, respectively. This should
be made consistent to minize confusion for the reader.

7.)
Eq. (25): not clear how right hand side depends on x' (what is x' btw?);
please clarify.

8.)
Please explain between Eq. (25) and Eq. (26) what is the motivation to
study the particular loss. Also please add some additional steps to
clarify how training the parameters a and W on task T_2 relates
to the dynamics of \hat{beta}.

9.)
When introducing the auxiliary fields in eq. (28) - (30) it would be
good to motivate this by their anticipated self-averaging due to the
summations over high-dimensional index spaces.

10.)
Are the results Eq. (84) - Eq. (95) used subsequently?
It seems you make an ansatz in Eq. (96) that is directly motivated
from Eq. (83) where effective parameters c_1, c_2, c_3 are determined
ad hoc. If this is the case, I would recommend removing Eqs. (84) - (95)
for clarity.

11.)
For the results shown in appendix B it does not become clear how
the numerical results were obtained. It is stated that they are
given by Langevin training.
My guess is that they are obtained by Langevin sampling from the
energy function given in Eq. (180), including the term \propto \delta.
Is this right? If so, it would be important to be stated, for example
in the figure caption.

In particular it should be clarified to the reader that the
results for the Bayesian case cannot be obtained by pre-training
the weights on task one first and subsequently switch to task two
(as is the case for the DMFT), because the posterior only captures the
stationary distribution which I believe should be agnostic to the
initialization. This qualitative difference should be explained
more clearly.


Contribution

The main contribution of the work is to present a theoretical treatment
of transfer learning. This is an important topic and the authors
present important theoretical results that explain qualitative
observation in networks trained on real data.

**Strengths:**

Transfer learning is an important topic of practical relevance
and theoretical progress is needed. The manuscript condenses this
complex question down to nicely analytically or semi-analytically
tractable settings.

**Weaknesses:**

The conciseness of the main part makes it hard to follow the main
text. The appendix, however, supplies all details as far as I see
(apart from points marked above).

Some connections between the idealized settings and the real-world
settings are still a bit loose and should be mentioned honestly
in the text (see points under "soundness" above).

**Questions:**

Please see itemized list above.

---

> ### Author Response · Authors · 2025-11-21
>
> We thank the Reviewer for acknowledging the theoretical contributions of our work. We have revised the manuscript to improve clarity and presentation, and we hope that the new version is now accessible to a broader audience.
>
> *1. Figure 2: why is there sometimes an optimal $\gamma_1$ and sometimes not?*
>
> Figure 2(a) is for a particular source/target alignment level. We made an additional plot (see Fig. 8 in Appendix) that show that monotone improvements in $\gamma_1$ are also possible if the source and target are sufficiently aligned. This, and also the additional non-isotripic structure in the real data, may account for the difference in Fig. 2. Regarding this second point, Fig. 9 in Appendix shows that on polynomial tasks there might be an optimal $\gamma_1$ if the source task is data limited, as it happens for our linear model of Fig. 2(a).
>
> *2. Shouldn't we forget the initial condition for sufficeintly rich task 2 training?*
>
> Thank you for this question. The asymptotic predictor depends on the representation inherited from Task 1. In Eq. (6), DMFT equations show that the hidden representation $h(\mathbf{x},t)$ at any $t>t_1$ depends explicitly on the full trajectory on $\mathcal T_1$. In other words, the dynamics on $\mathcal T_2$ are not ergodic: they follow a deterministic trajectory conditioned on the state at time $t_1$.
>
> *A similar question for Figure 3b): Here the initial transient of the test loss seems to be almost identical until the peak but ultimately at large step sizes, transfer learning seems to lead to worse generalization than training from scratch. Why is this?*
>
> We thank the reviewer for noticing this. Our point with this plot was that pre-training helps primarily in the case where a model extracts useful features from an easier task before fine-tuning on a task that would be hard to learn from scratch. This is similar to the staircase effect (see Abbe et al., 2021; 2023; 2024; Yang et al., 2025). However, going the other way, feature learning on a hard task could slow down learning compared to training from scratch on an easy task (a task the untrained model has a good spectral bias for).
>
> #### Questions
>
> 1. This is a good point, we will change notation.
> 2. We have tried to improve this in the main text.
> 3. Thank you for pointing this out. We have made the presentation more consistent and precise. We now specify the alignment value with the noise direction $\alpha_g$ in each panel.
> 4. The loss at $\nu_2 = 0$ should be the loss of the trivial predictor since we are considering kernel methods with kernels from models that were pretrained on task 1. In the case of normalized target labels $|\boldsymbol{y}_t|^2 = 1$,  the corresponding loss is $\mathcal{L}(\nu_2 = 0)=1$.
> 5. Thank you for pointing this out. After Eq. (12), we specify these constants are deterministic functions of $\gamma_1$ (feature learning strength on $\mathcal T_1$) and $\nu_1 = P_1/D$. We refer to Appendix C.2 for a detailed explanation about the ansatz of Eq. (11).
> 6. We changed the figure legend accordingly.
> 7. This was a typo. In Eq.(25) the kernel is now $\mathbf{K}(\mathbf{x},\mathbf{x}')=\mathbf{x}^{\top} \Big(\mathbf{I}+\frac{\chi}{D}\boldsymbol{\beta}_s\boldsymbol{\beta}_s^{\top}\Big)\mathbf{x}'$.
> 8. We added clarifications in the Appendix.
> 9. We specified this.
> 10. Thanks for allowing us to clarify this point. The ansatz on the kernel follows from the fact that, in the asymptotic limit, the fields $\{\mathbf{h}(t),\boldsymbol{\xi}(t)\}$ admit the causal decompositions shown in Eqs. (92)–(93), with coefficients that can be in principle derived through DMFT. From these decompositions one sees that the anisotropic component of the kernel must lie in the 2-dimensional subspace spanned by the signal direction $\boldsymbol{\beta}_s$ and the **uncorrelated** noise direction  $\boldsymbol{g}$.
> 11. Thank you for pointing this out. This is totally correct! For the Bayesian setting, the numerical results in Appendix C were obtained by running noisy SGD (i.e., (S)GD + white noise) on a loss function defined by the energy as in Eq.(180). This explicitly depends on the elastic coupling $\delta$. So, importantly, this is not implemented as a sequential pretrainig on $\mathcal{T}_1$ and fine-tuning on $\mathcal{T}_2$ as in the DMFT setting. We specify this in the Appendix.
>
> #### Weaknesses
> We are currently working on a cleaner and self-contained version of the main draft. We added an introductory paragraph for the toy models results to help the reader follow the subsequent paragraphs. We have a new appendix A for a primer on DMFT, and a new Appendix D for numerical details. We extended the discussion on the limitations of the current framework and the extent to which the theoretical predictions align with empirical neural network behavior on real tasks.

---

> > ### Comment · Reviewer_djNz · 2025-11-25
> >
> > I thank the authors for carefully addressing all my concerns, which mainly considered the presentation. I believe that having addressed these points, the paper will be more accessible also to other readers, increasing the score on the presentation.
> >
> > Overall, I will keep my score and it would be nice to see this work being presented.

---

### Author Response · Authors · 2025-11-21

We thank the Reviewers for their careful reading and constructive feedback. We have made substantial revisions in response to the comments, and we kindly ask you to consider those improvements in your evaluation. Below we specify the main concerns raised regarding the presentation, and the corresponding changes made to address them.

*No introduction to DMFT.*

We added a new Appendix A (“Primer on DMFT”) to make the paper self-contained. This appendix provides:

i. a high-level introduction to the DMFT formalism, and

ii. a detailed worked example for linear regression with isotropic covariance.
In this example, we explicitly derive the test loss at convergence under gradient flow and recover the baseline (no pretraining) loss $\mathcal{L} = 1-P/D$.

*Section 2 is too dense, difficult to unpack.*

We have substantially revised Section 2 to improve clarity:

i. Result 1: we now include a concise sketch of the derivation immediately after the statement of the result.

ii. Before presenting the toy-model results, we added a short introductory subsection explaining the data-generating model, the objective, and the motivation for the three regimes considered (data rich source, finite data source and ultra-rich pre-training).

iii. For Results 2–4, we inserted plain-English summary sentences that explicitly connect each result to its title, improving readability and guiding the reader through the logic.

*Numerical details are missing.*

We have added a new Appendix D for reproducibility. This specifies  how DMFT fixed point equations are solve for Result 1.

---

### Meta-Review · Area_Chair_EQXQ · 2026-01-08

**Summary:**

This paper studies theoretical aspects of infinitely wide two‑layer neural networks in the feature learning setting. The authors consider tractable toy models to understand how pertaining affects generalization using dynamical mean-field theory. This approach in turn allows for a better understanding of feature learning efficacy, including newly yielding average-case predictions for feature learning. Reviewer scores are mixed but the sentiment leans positive that the contributions are substantial on a timely and important topic, and in particular the theoretical results provide new insights.

**Reviewer Concerns:**

Reviewers noted some disconnect between the theory and empirical section, and pointed out issues with presentation, notation, and numerical details, though my read of the author response largely addresses the more important ones among these issues.

**Reviewer Scores:**

Reviewer scores are largely consistent with the content of reviews; one reviewer mentions they would be open to changing the score up with more clarification and I believe the response was sufficiently detailed to their concerns to warrant this in consideration

---

### Decision · Program_Chairs · 2026-01-26

Accept (Poster)